# Towards Instance-Optimal Offline Reinforcement Learning with Pessimism

**Ming Yin** [1,2] and **Yu-Xiang Wang**[1]

[1]Department of Computer Science, UC Santa Barbara
[2]Department of Statistics and Applied Probability, UC Santa Barbara
ming_yin@ucsb.edu   yuxiangw@cs.ucsb.edu

## Abstract

We study the *offline reinforcement learning* (offline RL) problem, where the goal is to learn a reward-maximizing policy in an unknown *Markov Decision Process* (MDP) using the data coming from a policy $\mu$. In particular, we consider the sample complexity problems of offline RL for finite-horizon MDPs. Prior works study this problem based on different data-coverage assumptions, and their learning guarantees are expressed by the covering coefficients which lack the explicit characterization of system quantities. In this work, we analyze the *Adaptive Pessimistic Value Iteration* (APVI) algorithm and derive the suboptimality upper bound that nearly matches

$$O\left(\sum_{h=1}^{H}\sum_{s_h,a_h} d_h^{\pi^\star}(s_h,a_h)\sqrt{\frac{\mathrm{Var}_{P_{s_h,a_h}}(V_{h+1}^\star + r_h)}{d_h^\mu(s_h,a_h)}}\sqrt{\frac{1}{n}}\right). \tag{1}$$

In complementary, we also prove a *per-instance* information-theoretical lower bound under the weak assumption that $d_h^\mu(s_h,a_h) > 0$ if $d_h^{\pi^\star}(s_h,a_h) > 0$. Different from the previous minimax lower bounds, the *per-instance lower bound* (via local minimaxity) is a much stronger criterion as it applies to individual instances separately. Here $\pi^\star$ is a optimal policy, $\mu$ is the behavior policy and $d_h^\mu$ is the marginal state-action probability. We call (1) the *intrinsic offline reinforcement learning bound* since it directly implies all the existing optimal results: minimax rate under uniform data-coverage assumption, horizon-free setting, single policy concentrability, and the tight problem-dependent results. Later, we extend the result to the *assumption-free* regime (where we make no assumption on $\mu$) and obtain the assumption-free intrinsic bound. Due to its generic form, we believe the intrinsic bound could help illuminate what makes a specific problem hard and reveal the fundamental challenges in offline RL.

## 1  Introduction

In *offline reinforcement learning* (offline RL Levine et al. [2020], Lange et al. [2012]), the goal is to learn a reward-maximizing policy in an unknown environment (*Markov Decision Process* or MDP) using the historical data coming from a (fixed) behavior policy $\mu$. Unlike online RL, where the agent can keep interacting with the environment and gain new feedback by exploring unvisited state-action space, offline RL usually populates when such online interplays are expensive or even unethical. Due to its nature of without the access to interact with the MDP model (which causes the distributional mismatches), most of the literature that study the sample complexity / provable efficiency of offline RL (*e.g.* Le et al. [2019], Chen and Jiang [2019], Xie and Jiang [2020, 2021],

35th Conference on Neural Information Processing Systems (NeurIPS 2021).

Yin et al. [2021a,b], Ren et al. [2021], Rashidinejad et al. [2021], Xie et al. [2021b]) rely on making different data-coverage assumptions for making the problem learnable and provide the near-optimal worst-case performance bounds that depend on their data-coverage coefficients. Those results are valuable in general as they do not depend on the structure of the particular problem, therefore, remain valid even for pathological MDPs. But is this good enough?

In practice, the empirical performances of offline reinforcement learning (*e.g.* Gulcehre et al. [2020], Fu et al. [2020, 2021], Janner et al. [2021]) are often far better than what those non-adaptive / problem-independent bounds would indicate. Although empirical evidence can help explain why we may observe better or worse performances on different MDPs, a systematic understanding of what types of decision processes and what kinds of behavior policies are inherently easier or more challenging for offline RL is lacking. Besides, despite the fact that a non-adaptive bound can learn even the pathological examples within the assumption family, there is no guarantee for the instances outside the family. However, practical offline reinforcement learning problems are usually beyond the scope of certain data-coverage assumptions, which limits the applicability of those results. Can we make as few assumptions as possible? Or even more, what can we guarantee when no assumption is made about offline learning?

Those motivate us to derive the provably efficient bounds that are adaptive to the individual instances but only require minimal assumptions so they can be widely applied in most cases. Ideally, such bounds should characterize the system structures of the specific problems, hold even for peculiar instances that do not satisfy the standard data-coverage assumptions, and recover the worst-case guarantees when the assumptions are satisfied. As mentioned in Zanette and Brunskill [2019], a fully adaptive characterization in RL is important as it might bring considerable saving in the time spent designing domain-specific RL solutions and in training a human expert to judge and recognize the complexity of different problems.

## 1.1 Our contribution

In this work, we provide the analysis for the *adaptive pessimistic value iteration* (APVI) (Algorithm 1) with finite horizon time-inhomogeneous (non-stationary) MDPs and derive a strong adaptive bound that is near-optimal under the weak assumption $d_h^\mu(s_h, a_h) > 0$ if $d_h^{\pi^\star}(s_h, a_h) > 0$ (Theorem 4.1). Specifically, our bound (quantity (1)) explicitly depends on the marginal importance ratios (between the optimal policy $\pi^\star$ and the behavior policy $\mu$) and the per-step conditional variances. In addition, we provide an instance-dependent (local minimax) lower bound (Theorem 4.3) to certify (1) is nearly optimal at the instance level for offline learning and call it *the intrinsic offline learning bound*. The intrinsic bound has the following consequences.

- In the non-adaptive / worst-case regime (4.1-4.3), the intrinsic bound implies $\widetilde{O}(H^3/d_m\epsilon^2)$ complexity under the uniform data-coverage 2.1, $\tilde{O}(H^3SC^\star/\epsilon^2)$ complexity under the single policy concentrability assumption 2.3 and $\widetilde{O}(H/d_m\epsilon^2)$ complexity when the sum of rewards is bounded by 1. All of those are optimal in their respectively regimes [Yin et al., 2021a, Rashidinejad et al., 2021, Xie et al., 2021b, Ren et al., 2021];

- In the adaptive domain (4.4), the intrinsic bound implies the tight problem-dependent counterpart of Zanette and Brunskill [2019], yields $\tilde{O}(H^3/nd_m)$ fast convergence in the deterministic systems, has improved complexity in the partially deterministic systems and a family of highly mixing problems, and remains optimal when reducing to the tabular contextual bandits.

Beyond the above, due to the generic form of the intrinsic bound, we could come up with as many problem instances (that are of our interests) as possible and study their properties. In this sense, the intrinsic bound helps illuminate the fundamental nature of offline RL.

Furthermore, as a step towards *assumption-free* offline reinforcement learning, we build a modified AVPI and obtain an adaptive bound that could characterize the suboptimality gap in the state-action space that is agnostic to the behavior policy (Theorem 5.1). To the best of our knowledge, all of these results are the first of its kinds.

## 1.2 Related work

Finite sample analysis for offline reinforcement learning can be traced back to Szepesvári and Munos [2005], Antos et al. [2008a,b] for the *infinite horizon discounted setting* via Fitted Q-Iteration (FQI) type function approximation algorithms. [Chen and Jiang, 2019, Le et al., 2019, Xie and Jiang, 2021, 2020] follow this line of research and derive the information-theoretical bounds. Recently, Xie and Jiang [2021] considers the offline RL with only the realizability assumption, Liu et al. [2020], Chang et al. [2021] considers the offline RL without sufficient coverage and Kidambi et al. [2020], Uehara and Sun [2021] uses the model-based approach for addressing offline RL. Under those weak coverage assumption, their finite sample analysis are suboptimal (*e.g.* in terms of the effective horizon $(1 - \gamma)^{-1}$). Recently, Yin et al. [2021a,b], Ren et al. [2021] study the finite horizon case. In the linear MDP case, Jin et al. [2020] studies the pessimistic algorithm for offline policy learning under only the compliance assumption, and, concurrently, Xie et al. [2021a] proposes the general pessimistic function approximation framework with instantiation in linear MDP and Zanette et al. [2021] shows actor-critic style algorithm is near-optimal for linear Bellman complete model. In addition, Wang et al. [2021], Zanette [2021] prove some exponential lower bounds under their linear function approximation assumptions.

Among them, there are a few works that achieve the sample optimality under their respective assumptions. Under the uniform data coverage (minimal state-action probability $d_m > 0$), Yin et al. [2021a] first proves the optimal $\tilde{O}(H^3/d_m\epsilon^2)$ complexity in the time-inhomogeneous MDP. Recently, Yin et al. [2021b] designs the offline variance reduction algorithm to achieve the optimal $\tilde{O}(H^2/d_m\epsilon^2)$ rate for the time-homogeneous case. Under the setting where the total cumulative reward is bounded by 1, Ren et al. [2021] obtains the horizon-free result with $\tilde{O}(1/d_m)$. More recently, Rashidinejad et al. [2021] considers the single concentrability coefficient $C^\star := \max_{s,a} d^{\pi^\star}(s, a)/d^\mu(s, a)$ and derives the upper bound $\tilde{O}[(1 - \gamma)^{-5} SC^\star/\epsilon^2]$ in the infinite horizon setting which is recently improved by the concurrent work Xie et al. [2021b]. While those worst-case guarantees are desirable, none of them can explain the hardness of the individual problems.[1]

## 2 Preliminaries

**Episodic non-stationary (time-varying) reinforcement learning.** A finite-horizon *Markov Decision Process* (MDP) is denoted by a tuple $M = (\mathcal{S}, \mathcal{A}, P, r, H, d_1)$ [Sutton and Barto, 2018], where $\mathcal{S}$ is the finite state space and $\mathcal{A}$ is the finite action space with $S := |\mathcal{S}| < \infty, A := |\mathcal{A}| < \infty$. A non-stationary transition kernel $P_h : \mathcal{S} \times \mathcal{A} \times \mathcal{S} \mapsto [0, 1]$ maps each state action$(s_h, a_h)$ to a probability distribution $P_h(\cdot|s_h, a_h)$ and $P_h$ can be different across the time. Besides, $r : \mathcal{S} \times A \mapsto \mathbb{R}$ is the expected instantaneous reward function satisfying $0 \leq r \leq 1$. $d_1$ is the initial state distribution. $H$ is the horizon. A policy $\pi = (\pi_1, \ldots, \pi_H)$ assigns each state $s_h \in \mathcal{S}$ a probability distribution over actions according to the map $s_h \mapsto \pi_h(\cdot|s_h) \, \forall h \in [H]$. An MDP together with a policy $\pi$ induce a random trajectory $s_1, a_1, r_1, \ldots, s_H, a_H, r_H, s_{H+1}$ with $s_1 \sim d_1, a_h \sim \pi(\cdot|s_h), s_{h+1} \sim P_h(\cdot|s_h, a_), \forall h \in [H]$ and $r_h$ is a random realization given the observed $s_h, a_h$.

**$Q$-values, Bellman (optimality) equations.** The value function $V_h^\pi(\cdot) \in \mathbb{R}^S$ and Q-value function $Q_h^\pi(\cdot, \cdot) \in \mathbb{R}^{S \times A}$ for any policy $\pi$ is defined as: $V_h^\pi(s) = \mathbb{E}_\pi[\sum_{t=h}^H r_t | s_h = s], \quad Q_h^\pi(s, a) = \mathbb{E}_\pi[\sum_{t=h}^H r_t | s_h, a_h = s, a], \quad \forall s, a \in \mathcal{S}, \mathcal{A}, h \in [H]$. The performance is defined as $v^\pi := \mathbb{E}_{d_1}[V_1^\pi] = \mathbb{E}_{\pi,d_1}\left[\sum_{t=1}^H r_t\right]$, where we denote $V_h^\pi, Q_h^\pi$ as column vectors and $P_h \in \mathbb{R}^{SA \times S}$ the transition matrix, then the vector form Bellman (optimality) equations follow $\forall h \in [H]$: $Q_h^\pi = r_h + P_h V_{h+1}^\pi, \quad V_h^\pi = \mathbb{E}_{a \sim \pi_h}[Q_h^\pi], \quad Q_h^\star = r_h + P_h V_{h+1}^\star, \quad V_h^\star = \max_a Q_h^\star(\cdot, a)$. In addition, we denote the per-step marginal state-action occupancy $d_h^\pi(s, a)$ as: $d_h^\pi(s, a) := \mathbb{P}[s_h = s | s_1 \sim d_1, \pi] \cdot \pi_h(a|s)$, which is the marginal state-action probability at time $h$.

**Offline setting and the goal.** The offline RL requires the agent to find a policy $\pi$ such that the performance $v^\pi$ is maximized, given only the episodic data $\mathcal{D} = \{(s_h^\tau, a_h^\tau, r_h^\tau, s_{h+1}^\tau)\}_{\tau \in [n]}^{h \in [H]}$ rolled out from some behavior policy $\mu$. The offline nature requires we cannot change $\mu$ and in particular we

---

[1]We do mention Zanette et al. [2021] is near-optimal in their setting, but it is unclear whether it remains optimal in the standard setting where $Q^\pi \in [0, H]$, since there is an additional $H$ factor by rescaling.

do not assume the functional knowledge of $\mu$. That is to say, given the batch data $\mathcal{D}$ and a targeted accuracy $\epsilon > 0$, the offline RL seeks to find a policy $\pi_{\text{alg}}$ such that $v^\star - v^{\pi_{\text{alg}}} \leq \epsilon$.

## 2.1 Assumptions in offline RL

We revise several types of assumptions proposed by existing studies that can yield provably efficient results. Recall $d_h^\mu(s_h, a_h)$ is the marginal state-action probability and $\mu$ is the behavior policy.

**Assumption 2.1** (Uniform data coverage [Yin et al., 2021a]). *The behavior policy obeys that $d_m := \min_{h,s_h,a_h} d_h^\mu(s_h, a_h) > 0$. Here the infimum is over all the states satisfying there exists certain policy so that this state can be reached by the current MDP with this policy.*

This is the strongest assumption in offline RL as it requires $\mu$ to explore each state-action pairs with positive probability. Under 2.1, it mostly holds $1/d_m \geq SA$. This reveals offline learning is generically harder than *the generative model setting* [Agarwal et al., 2020] in the statistical sense. On the other hand, this is required for the *uniform OPE* task in Yin et al. [2021a] as it seeks to simultaneously evaluate all the policies within the policy class and it is in general a harder task than offline learning itself.

**Assumption 2.2** (Uniform concentrability Szepesvári and Munos [2005], Chen and Jiang [2019]). *For all the policies, $C_\mu := \sup_{\pi,h} ||d_h^\pi(\cdot,\cdot)/d_h^\mu(\cdot,\cdot)||_\infty < \infty$.*

This is a classical offline RL condition that is commonly assumed in the function approximation scheme (*e.g.* Fitted Q-Iteration). Qualitatively, this is a uniform data-coverage assumption that is similar to Assumption 2.1, but quantitatively, the coefficient $C_\mu$ can be smaller than $1/d_m$ due the $d_h^\pi$ term in the numerator.

**Assumption 2.3** (Liu et al. [2019]). *There exists one optimal policy $\pi^\star$, s.t. $\forall s_h, a_h \in \mathcal{S}, \mathcal{A}$, $d_h^\mu(s_h, a_h) > 0$ if $d_h^{\pi^\star}(s_h, a_h) > 0$. We further denote the trackable set as $\mathcal{C}_h := \{(s_h, a_h) : d_h^\mu(s_h, a_h) > 0\}$.*

Assumption 2.3 is (arguably) the weakest assumption needed for accurately learning the optimal value $v^\star$ and we will use 2.3 for most parts of this paper. It only requires $\mu$ to trace the state-action space of one optimal policy and can be agnostic at other locations. Rashidinejad et al. [2021], Xie et al. [2021b] considers this assumption and provide analysis is based on the single concentrability coefficient $C^\star := \max_{s,a} d^{\pi^\star}(s,a)/d^\mu(s,a)$. The dependence on $C^\star$ makes their result less adaptive since there can be lots of locations that have the ratio $d^{\pi^\star}(s,a)/d^\mu(s,a)$ much smaller than $C^\star$. Furthermore, what could we end up with when 2.3 is not met? We will provide our answers in the subsequent sections.

## 3 A warm-up case study: Vanilla Pessimistic Value Iteration

As a step towards the optimal and strong adaptive offline RL bound, we analyze *the vanilla pessimistic value iteration* (VPVI), a tabular counterpart of *pessimistic value iteration* (PEVI initiated in Jin et al. [2020]), to understand what is missing for achieving the fully adaptivity. In particular, VPVI relies on the model-based construction.

**Model-based Components.** Given data $\mathcal{D} = \{(s_h^\tau, a_h^\tau, r_h^\tau, s_{h+1}^\tau)\}_{\tau \in [n]}^{h \in [H]}$, we denote $n_{s_h, a_h} := \sum_{\tau=1}^n \mathbf{1}[s_h^\tau, a_h^\tau = s_h, a_h]$ be the total counts that visit $(s_h, a_h)$ pair at time $h$, then we use the offline plug-in estimator to construct the estimators for $P_h$ and $r_h$ as:

$$\widehat{P}_h(s'|s_h, a_h) = \frac{\sum_{\tau=1}^n \mathbf{1}[(s_{h+1}^\tau, a_h^\tau, s_h^\tau) = (s', s_h, a_h)]}{n_{s_h, a_h}}, \ \widehat{r}_h(s_h, a_h) = \frac{\sum_{\tau=1}^n \mathbf{1}[(a_h^\tau, s_h^\tau) = (s_h, a_h)] \cdot r_h^\tau}{n_{s_h, a_h}},$$

(2)

if $n_{s_h, a_h} > 0$ and $\widehat{P}_h(s'|s_h, a_h) = 1/S, \widehat{r}_h(s_h, a_h) = 0$ if $n_{s_h, a_h} = 0$. In particular, we use the word "vanilla" as it directly mirrors Jin et al. [2020] with a pessimistic penalty of order $O(H/\sqrt{n_{s_h, a_h}})$.[2] With $\widehat{P}_h, \widehat{r}_h$ in Algorithm 2 (which we defer to Appendix), VPVI guarantees the following:

---

[2]This is due to $\sqrt{\phi(s_h, a_h)^\top \Lambda_h^{-1} \phi(s_h, a_h)}$ reduces to $\sqrt{1/n_{s_h, a_h}}$ when setting $\phi(s_h, a_h) = \mathbf{1}(s_h, a_h)$ and $\lambda = 0$.

**Theorem 3.1.** *Under the Assumption 2.3, denote $\bar{d}_m := \min_{h \in [H]}\{d_h^\mu(s_h, a_h) : d_h^\mu(s_h, a_h) > 0\}$. For any $0 < \delta < 1$, there exists absolute constants $c_0, C' > 0$, such that when $n > c_0 \cdot 1/\bar{d}_m \cdot \iota$ ($\iota = \log(HSA/\delta)$), with probability $1 - \delta$, the output policy $\widehat{\pi}$ of VPVI satisfies*

$$0 \leq v^\star - v^{\widehat{\pi}} \leq C' H \sum_{h=1}^{H} \sum_{(s_h, a_h) \in \mathcal{C}_h} d_h^{\pi^\star}(s_h, a_h) \cdot \sqrt{\frac{\iota}{n \cdot d_h^\mu(s_h, a_h)}}. \tag{3}$$

The full proof can be found in Appendix C. Theorem 3.1 makes some improvements over the existing works. First, it is more adaptive than the results with uniform data-coverage Assumption 2.1 (Yin et al. [2021a], Ren et al. [2021]). In addition, by straightforward calculation (3) can be bounded by $\widetilde{O}(\sqrt{H^4 SC^\star/n})$ which improves VI-LCB [Rashidinejad et al., 2021] by a factor of $H$.[3] Besides, the analysis of VPVI also improves the direct reduction of PEVI [Jin et al., 2020] in the tabular case by a factor $SA$ since their $\beta = SAH$ when $d = SA$.

However, VPVI is not optimal as the dependence on horizon is $H^4$ which does not match the optimal worst case guarantee $H^3$ [Yin et al., 2021a] in the nonstationary setting. Also, the explicit dependence on $H$ in (3) possibly hides some key features of the specific offline RL instances. For example, no improvement can be made if the system has the deterministic transition.

---

**Algorithm 1** Adaptive (*assumption-free*) Pessimistic Value Iteration or LCBVI-Bernstein

---
1: **Input:** Offline dataset $\mathcal{D} = \{(s_h^\tau, a_h^\tau, r_h^\tau, s_{h+1}^\tau)\}_{\tau, h=1}^{n, H}$. Set $C_1 = 2, C_2 = 14$, failure probability $\delta$.
2: **Initialization:** Set $\widehat{V}_{H+1}(\cdot) \leftarrow 0$. Set $\iota = \log(HSA/\delta)$. (if assumption-free, set $M^\dagger, \widehat{M}^\dagger$ as in Section 5.)
3: **for** time $h = H, H-1, \ldots, 1$ **do**
4:     Set $\widehat{Q}_h(\cdot, \cdot) \leftarrow \widehat{r}_h(\cdot, \cdot) + (\widehat{P}_h \cdot \widehat{V}_{h+1})(\cdot, \cdot)$  (use $\widehat{r}_h^\dagger + (\widehat{P}_h^\dagger \cdot \widehat{V}_{h+1})$ if assumption-free)
5:     $\forall s_h, a_h$, set $\Gamma_h(s_h, a_h) = C_1 \sqrt{\frac{\mathrm{Var}_{\widehat{P}_{s_h, a_h}}(\widehat{r}_h + \widehat{V}_{h+1}) \cdot \iota}{n_{s_h, a_h}}} + \frac{C_2 H \cdot \iota}{n_{s_h, a_h}}$ if $n_{s_h, a_h} \geq 1$, o.w. set to $\frac{CH\iota}{1}$.
6:     (If assumption-free, use $C_1 \sqrt{\mathrm{Var}_{\widehat{P}_{s_h, a_h}^\dagger}(\widehat{r}_h^\dagger + \widehat{V}_{h+1}) \cdot \iota/n_{s_h, a_h}} + \frac{C_2 H \cdot \iota}{n_{s_h, a_h}}$ if $n_{s_h, a_h} \geq 1$, o.w. use $0$.)
7:     Set $\widehat{Q}_h^p(\cdot, \cdot) \leftarrow \widehat{Q}_h(\cdot, \cdot) - \Gamma_h(\cdot, \cdot)$. Set $\overline{Q}_h(\cdot, \cdot) \leftarrow \min\{\widehat{Q}_h^p(\cdot, \cdot), H - h + 1\}^+$.   // Pessmistic update
8:     $\forall s_h$, Select $\widehat{\pi}_h(\cdot|s_h) \leftarrow \arg\max_{\pi_h} \langle \overline{Q}_h(s_h, \cdot), \pi_h(\cdot|s_h) \rangle$. Set $\widehat{V}_h(s_h) \leftarrow \langle \overline{Q}_h(s_h, \cdot), \widehat{\pi}_h(\cdot|s_h) \rangle$.
9: **end for**
10: **Output:** $\{\widehat{\pi}_h\}$.

---

# 4  Intrinsic Offline Reinforcement Learning bound

Now we go deeper to understand what is the more intrinsic characterization for offline reinforcement learning. From the study of VPVI, penalizing the Q-function by $\widetilde{O}(H/\sqrt{n_{s_h, a_h}})$ is crude as it estimates the confidence width of $\widehat{Q}_h$ in Algorithm 2 too conservatively therefore loses the accuracy (the bound is suboptimal). This motivates us to use empirical standard deviation instead to create a more adaptive (and also less conservative) Bernstein-type confidence width as the pessimistic penalty:

$$\Gamma_h(s_h, a_h) = \widetilde{O}\left[\sqrt{\frac{\mathrm{Var}_{\widehat{P}_{s_h, a_h}}(\widehat{r}_h + \widehat{V}_{h+1})}{n_{s_h, a_h}}} + \frac{H}{n_{s_h, a_h}}\right] \text{ (if } n_{s_h, a_h} > 0); = \widetilde{O}(H) \text{ (if } n_{s_h, a_h} = 0). \tag{4}$$

and update $\widehat{Q}_h \leftarrow \widehat{Q}_h - \Gamma_h$. On one hand, $\sqrt{\mathrm{Var}_{\widehat{P}_{s_h, a_h}}(\widehat{r}_h + \widehat{V}_{h+1})/n_{s_h, a_h}}$ is a "less pessimistic" penalty than VPVI due to $\sqrt{\mathrm{Var}_{\widehat{P}}(\widehat{r}_h + \widehat{V}_{h+1})} \leq H$ and critically this design is more data-adaptive since it holds negative view towards the locations with high uncertainties and recommends the locations that we are confident about, as opposed to the online RL (which encourages exploration in the uncertain locations). Such principles are not reflected by the isotropic design in VPVI. On the other hand, it carries the extremely negative view towards fully agnostic locations $\widetilde{O}(H)$ which in turn causes the agent unlikely to choose them. We summarized the this *adaptive pessimistic value*

---
[3]To be rigorous, translating the result from the infinite horizon setting to the finite horizon setting requires explanation. We add this discussion in Appendix D.

*iteration* (APVI) into the Algorithm 1, with $\widehat{P}_h, \widehat{r}_h$ defined in (2). APVI has the following guarantee. A sketch of the analysis is presented in Section B and Appendix F includes the full proof.

**Theorem 4.1** (Intrinsic offline RL bound). *Under the Assumption 2.3, denote $\bar{d}_m := \min_{h \in [H]} \{ d_h^\mu(s_h, a_h) : d_h^\mu(s_h, a_h) > 0 \}$. For any $0 < \delta < 1$, there exists absolute constants $c_0, C' > 0$, such that when $n > c_0 \cdot 1/\bar{d}_m \cdot \iota$ ($\iota = \log(HSA/\delta)$), with probability $1 - \delta$, the output policy $\widehat{\pi}$ of APVI (Algorithm 1) satisfies ($\widetilde{O}$ hides log factor and higher order terms)*

$$0 \le v^\star - v^{\widehat{\pi}} \le C' \sum_{h=1}^{H} \sum_{(s_h, a_h) \in \mathcal{C}_h} d_h^{\pi^\star}(s_h, a_h) \cdot \sqrt{\frac{\mathrm{Var}_{P_{s_h, a_h}}(r_h + V_{h+1}^\star) \cdot \iota}{n \cdot d_h^\mu(s_h, a_h)}} + \widetilde{O}\left(\frac{H^3}{n \cdot \bar{d}_m}\right) \quad (5)$$

**Remark 4.2.** *APVI (Algorithm 1) can also be called **LCBVI-Bernstein** as it creates the offline counterpart of UCBVI in Azar et al. [2017]. However, to highlight that the resulting bound fully adapts to the specific system structure, we use the word "adaptive" instead.*

APVI makes significant improvements in a lot of aspects. First and foremost, the dominate term is fully expressed by the system quantities that admits no explicit dependence on $H, S, A$. To the best of our knowledge, this is the first offline RL bound that concretely depicts the interrelations within the problem when the problem instance is a tuple $(M, \pi^\star, \mu)$: an MDP $M$ (coupled with the optimal policy $\pi^\star$) with the data rolling from an offline logging policy $\mu$. As we will discuss later, this result indicates (nearly) all the optimal worst-case non-adaptive bounds (and clearly also the VPVI) under their respective regimes / assumptions. Thus, (5) is generic. More interestingly, Theorem 4.1 caters to the specific MDP structures and adaptively yields improved sample complexities (*e.g.* faster convergence in deterministic systems) that existing works cannot imply. Such features are crucial as it helps us to understand what type of problems are harder / easier than others, and even more, in a *quantitative* way. Last but not least, to illustrate this bound exhibits the intrinsic nature of offline RL, we prove a *per-instance dependent* information-theoretical lower bound that shares a similar formulation. The proof of Theorem 4.3 can be found in Appendix G.

**Theorem 4.3** (Instance-dependent information theoretical offline lower bound). *Denote $\mathcal{G} := \{ (\mu, M) : \exists \pi^\star \ s.t. \ d_h^\mu(s, a) > 0 \ if \ d_h^{\pi^\star}(s, a) > 0 \}$. Fix an instance $\mathcal{P} = (\mu, M) \in \mathcal{G}$. Let $\mathcal{D}$ consists of $n$ episodes and define $\xi = \sup_{h, s_h, a_h, s_{h+1}, d_h^\mu(s_h, a_h) \cdot \mathrm{Var}_{P_{s_h, a_h}}(V_{h+1}^\star) > 0} \frac{P_h(s_{h+1}|s_h, a_h)\left(V_{h+1}^\star(s_{h+1}) - \mathbb{E}_{P_{s_h, a_h}}[V_{h+1}^\star]\right)}{\sqrt{2 \cdot d_h^\mu(s_h, a_h) \cdot \mathrm{Var}_{P_{s_h, a_h}}(V_{h+1}^\star)}}$. Let $\widehat{\pi}$ to be the output of any algorithm. Define the local non-asymptotic minimax risk as*

$$\mathfrak{R}_n(\mathcal{P}) := \sup_{\mathcal{P}' \in \mathcal{G}} \inf_{\widehat{\pi}} \max_{\mathcal{Q} \in \{\mathcal{P}, \mathcal{P}'\}} \sqrt{n} \cdot \mathbb{E}_{\mathcal{Q}}\left[v^\star(\mathcal{Q}) - v^{\widehat{\pi}}\right] \quad (6)$$

*where $v^\star(\mathcal{Q})$ denotes the optimal value under the instance $\mathcal{Q}$. Then there exists universal constants $c_0, p, C > 0$, such that if $n \ge c_0 H^6 \xi^4 / \left(\sum_{h=1}^{H} \sum_{s_h, a_h} d_h^{\pi^\star}(s_h, a_h) \sqrt{\frac{\mathrm{Var}_{P_{s_h, a_h}}(V_{h+1}^\star)}{\zeta \cdot d_h^\mu(s_h, a_h)}}\right)^2$, with constant probability $p > 0$, Then we have (here $\zeta = H/\bar{d}_m$):*

$$\mathfrak{R}_n(\mathcal{P}) \ge C \cdot \sum_{h=1}^{H} \sum_{(s_h, a_h) \in \mathcal{C}_h} d_h^{\pi^\star}(s_h, a_h) \cdot \sqrt{\frac{\mathrm{Var}_{P_{s_h, a_h}}(r_h + V_{h+1}^\star)}{\zeta \cdot d_h^\mu(s_h, a_h)}}, \quad (7)$$

*where $\mathcal{P} = (\mu, M)$ and $M = (\mathcal{S}, \mathcal{A}, P, r, H, d_1)$.*

The interpretation of Theorem 4.3 is: for any instance $\mathcal{P}$, learning requires (7) (divided by $1/\sqrt{n}$) for any algorithm. Note this notion is significantly stronger than the previous minimax offline lower bounds [Yin et al., 2021a, Rashidinejad et al., 2021, Xie et al., 2021b, Jin et al., 2020] (where they only select a particular family of hard problems), therefore, their lower bounds in general do not hold for individual instances.

The quantity (1) nearly-matches the per-instance lower bound (7) (they deviate by a factor of $\zeta = H/\bar{d}_m$ due to the technical reason) and, in addition, we provide a matching minimax lower bound in Appendix H. These results certify Theorem 4.1 is not only adaptive but also near-optimal. Hence, we call the quantity $\sum_{h=1}^{H} \sum_{(s_h, a_h) \in \mathcal{C}_h} d_h^{\pi^\star}(s_h, a_h) \cdot \sqrt{\frac{\mathrm{Var}_{P_{s_h, a_h}}(r_h + V_{h+1}^\star)}{n \cdot d_h^\mu(s_h, a_h)}}$ *intrinsic offline*

*reinforcement learning bound*. In the sequel, we provide thorough discussions to explain the intrinsic bound embraces the fundamental challenges in offline RL and the strong adaptivity. The detailed technical derivations that are missing in Section 4.1-4.4 are deferred to Appendix I.

**Intrinsic Offline Learning Bound**

$$\sum_{h=1}^{H} \sum_{s_h,a_h} d_h^{\pi^*}(s_h, a_h) \sqrt{\frac{\mathrm{Var}_{P_{s_h,a_h}}(V_{h+1}^* + r_h)}{d_h^{\mu}(s_h, a_h)}} \cdot \sqrt{\frac{1}{n}}$$

**Uniform Visitation**

$$\widetilde{O}\left(\sqrt{\frac{H^3}{n \cdot d_m}}\right)$$

**Single Concentrability**

$$\widetilde{O}\left(\sqrt{\frac{H^3 SC^*}{n}}\right)$$

**Adaptive Domain**

$$\widetilde{o}\left(\sum_{h=1}^{H} \sqrt{\frac{\mathbb{Q}_h^*}{n \cdot d_m}}\right) + \widetilde{o}\left(\frac{H^3}{n \cdot d_m}\right)$$

Figure 1: A visualization on how intrinsic learning bound subsumes existing best-known results: uniform visitation, single concentrability (partial coverage) and adaptive domain.

## 4.1 Optimality under Uniform data-coverage assumption

Under the uniform exploration Assumption 2.1 with parameter $d_m := \min_{h,s_h,a_h} d_h^{\mu}(s_h, a_h) > 0$, Yin et al. [2021a] analyzes the model-based plug-in approach and obtains the optimal sample complexity $\widetilde{O}(H^3/d_m \epsilon^2)$ and shows $\Omega(H^3/d_m \epsilon^2)$ is also the lower bound. Indeed, this rate can be directly implied by the intrinsic RL bound via *Cauchy inequality* and *the Sum of Total Variance* (Lemma J.6):[4]

$$\sum_{h=1}^{H} \langle d_h^{\pi^*}(\cdot), \sqrt{\frac{\mathrm{Var}_{P_{(\cdot)}}(r_h + V_{h+1}^{\star})}{n \cdot d_h^{\mu}(\cdot)}} \rangle = \sum_{h=1}^{H} \langle \sqrt{d_h^{\pi^*}(\cdot)}, \sqrt{\frac{d_h^{\pi^*}(\cdot) \odot \mathrm{Var}_{P_{(\cdot)}}(r_h + V_{h+1}^{\star})}{n \cdot d_m}} \rangle$$

$$\leq \sum_{h=1}^{H} \left\| \sqrt{d_h^{\pi^*}(\cdot)} \right\|_2 \left\| \sqrt{\frac{d_h^{\pi^*}(\cdot) \odot \mathrm{Var}_{P_{(\cdot)}}(r_h + V_{h+1}^{\star})}{n \cdot d_m}} \right\|_2 \leq \sqrt{\frac{H \cdot \mathrm{Var}_{\pi^{\star}}(\sum_{h=1}^{H} r_h)}{n \cdot d_m}} \leq \sqrt{\frac{H^3}{n \cdot d_m}} \tag{8}$$

which translates to $\widetilde{O}(H^3/d_m \epsilon^2)$ complexity. Our result maintains the optimal worst-case guarantee when $\mu$ has the uniform data-coverage:

**Proposition 4.4.** *Under Assumption 2.1 and apply Theorem 4.1, APVI achieves the sample complexity of minimax-rate $\widetilde{O}(H^3/d_m \epsilon^2)$ (Theorem 4.1 and Theorem G.2 in Yin et al. [2021a]).*

**Remark 4.5.** *We believe if the MDP is time-invariant, then by a modified construction of $\widehat{P}, \widehat{r}$ in (2) our result will imply the minimax-rate of $\widetilde{O}(H^2/d_m \epsilon^2)$ as achieved in Yin et al. [2021b]. We include this discussion in Appendix I.*

## 4.2 Bounded sum of total rewards and the Horizon-Free case

There is another thread of studies that follow the bounded sum of total rewards assumption: *i.e.* $r_h \geq 0, \sum_{h=1}^{H} r_h \in [0, 1]$ [Krishnamurthy et al., 2016, Jiang et al., 2017, Zhang et al., 2021]. Such a setting is much weaker than the uniform bounded instantaneous reward condition, as explained in Jiang and Agarwal [2018]. In offline RL, Ren et al. [2021] derives the nearly horizon-free worst case bound $\widetilde{O}(\sqrt{1/nd_m})$ for the time-invariant MDPs, under the Assumption 2.1. As a comparison, our Theorem 4.1 achieves the following guarantee for the time-varying (non-stationary) MDPs.

**Proposition 4.6.** *Assume $r_h \geq 0$, $\sum_{h=1}^{H} r_h \leq 1$. Then in the time-varying case AVPI (Theorem 4.1) outputs a policy $\widehat{\pi}$ such that the suboptimality gap $v^{\star} - v^{\widehat{\pi}}$ is bounded by $\widetilde{O}(\sqrt{H/nd_m})$ with high probability under the Assumption 2.1.*

---

[4]Here $\odot$ denotes element-wise multiplication. Also note under 2.1, our $\bar{d}_m = d_m$.

The derivation is straightforward by using $\mathrm{Var}_{\pi^\star}(\sum_{h=1}^H r_h) \le 1$ in (8). This proposition is interesting since it indicates when the MDP is non-stationary, $\widetilde{O}(H/d_m\epsilon^2)$ is required in the worst case even under $\sum_{h=1}^H r_h \le 1$.[5] The extra $H$ factor resembles the challenge that we have $H$ transitions $(P_1, \ldots, P_H)$ to learn, as opposed to the bandit-type $1/d_m\epsilon^2$ result due to there is only one $P$ throughout (time-invariant). This reveals that one hardness in solving the MDP is in proportion to the number of different transition kernels within the MDP. Such a finding could help researchers understand the special settings like *low switching cost in transitions* [Bai et al., 2019] or *non-stationarity* [Cheung et al., 2020].

## 4.3 Optimality with Single Concentrability

In the finite horizon discounted setting, Rashidinejad et al. [2021] proposes the single policy concentrability assumption which is defined as $C^\star := \max_{h,s,a} \frac{d_h^{\pi^\star}(s,a)}{d_h^\mu(s,a)} < \infty$ in the current episodic non-stationary MDP setting. As discussed in Appendix D, their lower bound translates to $\Omega(\sqrt{\frac{H^3 S C^\star}{n}})$ and their VI-LCB algorithm yields $\widetilde{O}(\sqrt{\frac{H^5 S C^\star}{n}})$ suboptimality gap in $H$-horizon case. Since single policy concentrability is strictly weaker than its uniform version (Assumption 2.2), we only discuss this set up. In particular, we have the following implication from our Theorem 4.1 (whose derivation can be found in Appendix I):

**Proposition 4.7.** *Let $\pi^\star$ be a deterministic policy such that $C^\star := \max_{h,s,a} \frac{d_h^{\pi^\star}(s,a)}{d_h^\mu(s,a)} < \infty$. Then by Theorem 4.1, with high probability the output policy of APVI satisfies the suboptimality gap $\widetilde{O}(\sqrt{\frac{H^3 S C^\star}{n}})$ in the time-varying (non-stationary) MDPs.*

This can computed similar to (8) except we use $\frac{d_h^{\pi^\star}(s,a)}{d_h^\mu(s,a)} \le C^\star$. Our implication improves the VI-LCB by the factor $H^2$ (in terms of sample complexity) and is optimal (recover the concurrent Xie et al. [2021b]). Qualitatively, single concentrability is the same as Assumption 2.3, but the use of $C^\star$ makes the bound highly problem independent and limits the adaptivity. Problem dependent bound is a more interesting domain as it tailors to each MDP separately. We discuss it now.

## 4.4 Problem dependent domain

We define the *pre-step environmental norm* (the finite horizon counterpart of Maillard et al. [2014]) as: $\mathbb{Q}_h^\star = \max_{s_h,a_h} \mathrm{Var}_{P_{s_h,a_h}}(r_h + V_{h+1}^\star)$ for all $h \in [H]$, and relax the total sum of rewards to be bounded by any arbitrary value $\mathcal{B}$ (*i.e.* $\sum_{h=1}^H r_h \le \mathcal{B}$), then Theorem 4.1 implies:

**Proposition 4.8.** *Under Assumption 2.1, with high probability, subopmality of AVPI is bounded by*

$$\min\left\{\widetilde{O}\Big(\sum_{h=1}^H \sqrt{\frac{\mathbb{Q}_h^\star}{n\bar{d}_m}}\Big), \widetilde{O}\Big(\sqrt{\frac{H \cdot \mathcal{B}^2}{n\bar{d}_m}}\Big)\right\} + \widetilde{O}\Big(\frac{H^3}{n\bar{d}_m}\Big).$$

Such a result mirrors the online version of the tight problem-dependent bound Zanette and Brunskill [2019] but with a more general *pre-step environmental norm* for the non-stationary MDPs.[6] For the problem instances with either small $\mathcal{B}$ or small $\mathbb{Q}_h^\star$, our result yields much better performances, as discussed in the following.

**Deterministic systems.** For many practical applications of interest, the systems are equipped with low stochasticity, *e.g.* robotics, or even deterministic dynamics, *e.g.* the game of GO. In those scenarios, the agent needs less experience for each state-action therefore the learning procedure could be much faster. In particular, when the system is fully deterministic (in both transitions and rewards) then $\mathbb{Q}_h^\star = 0$ for all $h$. This enables a faster convergence rate of order $\frac{H^3}{n\bar{d}_m}$ and significantly improves

---

[5]Suppose in this case we can achieve $\widetilde{O}(1/d_m\epsilon^2)$ just like Ren et al. [2021], then by a rescaling we obtain the $\widetilde{O}(H^2/d_m\epsilon^2)$ under the usual $0 \le r_h \le 1$ assumption which violates the $\Omega(H^3/d_m\epsilon^2)$ lower bound.

[6]Zanette and Brunskill [2019] uses the maximal version by maximizing over $h$.

over the existing non-adaptive results that have order $\frac{1}{\sqrt{n}}$. The convergence rate $\frac{1}{n}$ matches Wen and Van Roy [2013] by translating their constant (in $T$) regret into the PAC bound.

**Partially deterministic systems.** Practical worlds are complicated and we could sometimes have a mixture model which contains both deterministic and stochastic steps. In those scenarios, the main complexity is decided by the number of stochastic stages: suppose there are $t$ stochastic $P_h, r_h$'s and $H - t$ deterministic $P_{h'}, r_{h'}$'s, then completing the offline learning guarantees $t \cdot \sqrt{\max Q_h^\star / n \bar{d}_m}$ suboptimality gap, which could be much smaller than $H \cdot \sqrt{\max Q_h^\star / n \bar{d}_m}$ when $t \ll H$.

**Fast mixing domains.** Consider a class of highly mixing non-stationary MDPs (a variant of Zanette and Brunskill [2018]) that satisfies the transition $P_h(\cdot | s_h, a_h) := \nu_h(\cdot)$ depends on neither the state $s_h$ nor the action $a_h$. Define $\bar{s}_t := \arg\max V_t^\star(s)$ and $\underline{s}_t := \arg\max V_t^\star(s)$. Also, denote $\mathrm{rng} V_h^\star$ to be the range of $V_h^\star$. In such cases, Bellman optimality equations have the form

$$V_h^\star(\bar{s}_h) = \max_a \left( r_h(\bar{s}_h, a) + \nu_h^\top V_{h+1}^\star \right), \quad V_h^\star(\underline{s}_h) = \max_a \left( r_h(\underline{s}_h, a) + \nu_h^\top V_{h+1}^\star \right),$$

which yields $\mathrm{rng} V_h^\star = V_h^\star(\bar{s}_h) - V_h^\star(\underline{s}_h) = \max_a r_h(\bar{s}_h, a) - \min_a r_h(\underline{s}_h, a) \le 1$, and this in turn gives $\mathbb{Q}_h^\star \le 1 + (\mathrm{rng} V_h^\star)^2 = 2$. As a result, the suboptimality is bounded by $\widetilde{O}(\sqrt{H^2/nd_m})$ in the worst case. This result reveals, although this is a family of stochastic non-stationary MDPs, but it is only as hard as the family of stationary MDPs in the minimax sense ($\Omega(H^2/d_m \epsilon^2)$).

**Tabular contextual bandits.** Our result also implies $\widetilde{O}(\sum_{x_1, a_1} d_1^{\pi^\star}(x_1, a_1) \sqrt{\frac{\mathrm{Var}(r_1)}{n \cdot d_1^\mu(x_1, a_1)}})$ gap for the *offline tabular contextual bandit* problem and improves to $\widetilde{O}(1/nd_m)$ when the reward is deterministic. In either cases, the result is optimal and this is due to: when $r_1$ is deterministic, the agent only needs one sample at every location (see Bubeck and Cesa-Bianchi [2012] for a survey).

# 5 Towards Assumption-Free Offline RL

While assumption 2.3 is (arguably) the weakest assumption for correctly learning the optimal value, for the real-world applications even this might not be guaranteed. Can we still learn something meaningful? In this section, we consider this most general setting where the behavior policy $\mu$ can be arbitrary. In this case, $\mu$ might not cover any optimal policy $\pi^\star$ (*i.e.* there might be high reward location $(s, a)$ that $\mu$ can never visit, *e.g.* in the extreme case where a clumsy doctor only uses one treatment all the time), and, irrelevant to the number of episode $n$, a constant suboptimality gap needs to be suffered. To tackle this problem, we create a fictitious augmented MDP $M^\dagger$ that can help characterize the discrepancy of the values between the original MDP $M$ and the estimated MDP $\widehat{M}^\dagger$. In particular, $M^\dagger$ is negative towards agnostic state-actions $s_h, a_h$ by setting $r_h^\dagger = 0$ and transitions to an absorbing state $s_{h+1}^\dagger$.

**Pessimistic augmented MDP.** $M^\dagger$ is defined with one extra state $s_h^\dagger$ for all $h \in \{2, \ldots, H+1\}$ with the augmented state space $\mathcal{S}^\dagger = \mathcal{S} \cup \{s_h^\dagger\}$. The transition and the reward are defined as follows:

$$P_h^\dagger(\cdot \mid s_h, a_h) = \begin{cases} P_h(\cdot \mid s_h, a_h), \ n_{s_h, a_h} > 0, \\ \delta_{s_{h+1}^\dagger}, \ s_h = s_h^\dagger \text{ or } n_{s_h, a_h} = 0. \end{cases} \quad r^\dagger(s_h, a_h) = \begin{cases} r(s_h, a_h), \ n_{s_h, a_h} > 0, \\ 0, \ s_h = s_h^\dagger \text{ or } n_{s_h, a_h} = 0. \end{cases}$$

here $\delta_s$ is the Dirac measure and we denote $V_h^{\dagger \pi}$ and $v^{\dagger \pi}$ to be the values under $M^\dagger$. $\widehat{M}^\dagger$ is the empirical counterpart of $M^\dagger$ with $\widehat{P}, \widehat{r}$ (the same as (2)) replacing $P, r$. By Algorithm 1, we have

**Theorem 5.1** (Assumption-free offline reinforcement learning). *Let us make no assumption for $\mu$ and still denote $\bar{d}_m := \min_{h \in [H]} \{ d_h^\mu(s_h, a_h) : d_h^\mu(s_h, a_h) > 0 \}$. For any $0 < \delta < 1$, there exists absolute constants $c_0, C' > 0$, such that when $n > c_0 \cdot 1/\bar{d}_m \cdot \iota$ ($\iota = \log(HSA/\delta)$), with probability $1 - \delta$, the output policy $\widehat{\pi}$ of APVI satisfies (recall $\mathcal{C}_h := \{(s_h, a_h) : d_h^\mu(s_h, a_h) > 0\}$)*

$$v^\star - v^{\widehat{\pi}} \le \sum_{h=2}^{H+1} d_h^{\dagger \pi^\star}(s_h^\dagger) + C' \sum_{h=1}^{H} \sum_{(s_h, a_h) \in \mathcal{C}_h} d_h^{\dagger \pi^\star}(s_h, a_h) \cdot \sqrt{\frac{\mathrm{Var}_{P_{s_h, a_h}^\dagger}(r_h^\dagger + V_{h+1}^{\dagger \pi^\star}) \cdot \iota}{n \cdot d_h^\mu(s_h, a_h)}} + \widetilde{O}\left( \frac{H^3}{n \bar{d}_m} \right), \quad (9)$$

*where $d_h^{\dagger \pi^\star}(s_h, a_h) \le d_h^{\pi^\star}(s_h, a_h), V_h^{\dagger \pi^\star}(s_h) \le V_h^\star(s_h)$ for all $s_h, a_h \in \mathcal{S} \times \mathcal{A}$, and for all $h \in [H]$, $d_h^{\dagger \pi^\star}(s_h^\dagger) = \sum_{t=1}^{h-1} \sum_{(s_t, a_t) \in \mathcal{S} \times \mathcal{A} \backslash \mathcal{C}_t} d_t^{\dagger \pi^\star}(s_t, a_t)$. The proof is in Appendix E.*

**Take-aways of Theorem 5.1.** In $M^\dagger$, there is no agnostic location any more since the original unknown spaces now all have *known* deterministic transitions to $s^\dagger$ in $M^\dagger$. At a price, the algorithm has to suffer the constant suboptimality $\sum_{h=2}^{H+1} d_h^{\dagger \pi^\star}(s_h^\dagger)$ due to no data in the region. The quantity $\sum_{h=2}^{H+1} d_h^{\dagger \pi^\star}(s_h^\dagger)$ helps characterize the hardness when nothing is assumed about $\mu$: it is always less than $H$ (cannot suffer more than $H$ suboptimality); under Assumption 2.1, it is 0 since $M^\dagger = M$ with high probability (by Chernoff bound) and this causes $\mathcal{S} \times \mathcal{A} \backslash \mathcal{C}_h = \emptyset$; under Assumption 2.3, it is also 0 and 5.1 reduces to Theorem 4.1 (see Appendix F).

## 5.1 Assumption Free vs Without Great Coverage (Partial Coverage)

Recently there is a surge of studies that aim at weakening the assumptions of provable offline / batch RL. Those learning bounds are derived (mostly) under the insufficient data coverage assumptions. One type of works consider the assumption *without great coverage* (or partial coverage): Chang et al. [2021], Uehara and Sun [2021] assume $\max_{s,a} d^{\pi_e}(s,a)/\mu(s,a) < \infty$ where $\pi_e$ is either an expert policy or a policy of great quality and they further compete against with this policy $\pi_e$. Those assumptions are similar to 2.3 and therefore are stronger than the assumption-free RL we considered in 5.1.

In addition, there are other studies that apply to the case where $\mu$ can be arbitrary: Liu et al. [2020] considers the behavior policy with insufficient coverage probability $\epsilon_\zeta$ (see their Definition 1), and they end up with the constant suboptimality gap $\frac{V_{\max} \epsilon_\zeta}{1-\gamma}$ (their Theorem 1), when the insufficient coverage probability $\epsilon_\zeta > 0$, this gap has order $(1-\gamma)^{-2}$, which is larger in order than the biggest possible suboptimality gap $(1-\gamma)^{-1}$ therefore unable to characterize the essential statistical gap over the region that can never be visited by the behavior policy (and this happens similarly in Kidambi et al. [2020], see their Theorem 1); Jin et al. [2020] derive the nice assumption-free result via regularization and their bound can incur $O(H^2)$ constant gap when there is at least one $(s_h, a_h)$ cannot be obtained by $\mu$ for all $h \in [H]$ (*i.e.* replacing $nd_h^\mu(s_h, a_h)$ by 1 in (3)). The concurrent work Xie et al. [2021a] provides a better characterization (and they call it *off-support error*) with roughly $\frac{1}{1-\gamma} \sum_{(s,a) \in \mathcal{S} \times \mathcal{A}} (d_\pi \backslash \nu)(s,a) [\Delta f_\pi(s,a) - (\mathcal{T}^\pi \Delta f_\pi)(s,a)]$, however, in the worst case $\Delta f_\pi(s,a) - (\mathcal{T}^\pi \Delta f_\pi)(s,a)$ might be large (which depends on the quality (assumption) of the function approximation class).

In contrast, our $\sum_{h=2}^{H+1} d_h^{\dagger \pi^\star}(s_h^\dagger)$ quantity (with $d_h^{\dagger \pi^\star}(s_h^\dagger) = \sum_{t=1}^{h-1} \sum_{(s_t, a_t) \in \mathcal{S} \times \mathcal{A} \backslash \mathcal{C}_t} d_t^{\dagger \pi^\star}(s_t, a_t) \leq 1$) describes the "must-suffer" gap in a more precise way by absorbing all the agnostic probabilities into $s^\dagger$ and it is always bounded between 0 and $H$. It reduces to 0 when $\pi^\star$ is covered. The gap is always of order $H$ (as opposed to $O(H^2)$).

# 6 Discussion and Conclusion

This work studies the offline reinforcement learning problem and contributes the intrinsic offline learning bound which is a near-optimal and strong adaptive bound that subsumes existing worst-case bounds under various assumptions. The adaptive characterization of the intrinsic bound abandons the explicit dependence on $H, S, A, C^\star, d_m$ and helps reveal the fundamental hardness of each individual instances. In this sense, it draws a clearer picture of what offline reinforcement learning looks like and serves as a step towards instance optimality in offline RL.

Nevertheless, it is still unclear whether (5) is optimal over all the instances. For example, for fully deterministic systems, our bound provides a faster convergence $H^3/n\bar{d}_m$, however, $H^3$ might be very suboptimal comparing to algorithms that are designed specifically for deterministic MDPs, since the agent only need to experience each location $(s,a)$ once to fully acquire the dynamic $P(\cdot|s,a)$ and $r(s,a)$. Recently, Xiao et al. [2021] goes beyond the minimax (worst case) optimality and studies the instance optimality behavior for the simplified batch bandit setting. One of their findings is: for "easy enough" tasks, different type of algorithms can be equally good, provably. This seems to suggest instance optimality only matters for problems that are hard to learn. How to formally define the instance optimality metric for different problems remains an open problem and how to design a single algorithm that can achieve optimality for all instances could be challenging (or even infeasible). We leave those as the future works.

**Acknowledgment**

The research is partially supported by NSF Awards #2007117 and #2003257. MY would like to thank Chenjun Xiao for bringing up a related literature [Xiao et al., 2021] and Masatoshi Uehara, Yu Bai for helpful suggestions.

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
