# Appendix

---

**Algorithm 2** Vanilla Pessimistic Value Iteration

---

1: **Input:** Offline dataset $\mathcal{D} = \{(s_h^\tau, a_h^\tau, r_h^\tau, s_{h+1}^\tau)\}_{\tau,h=1}^{n,H}$. Absolute Constant $C$, failure probability $\delta$.
2: **Initialization:** Set $\widehat{V}_{H+1}(\cdot) \leftarrow 0$.
3: **for** time $h = H, H-1, \ldots, 1$ **do**
4:     Set $\widehat{Q}_h(\cdot, \cdot) \leftarrow \widehat{r}_h(\cdot, \cdot) + (\widehat{P}_h \cdot \widehat{V}_{h+1})(\cdot, \cdot)$
5:     $\forall s_h, a_h$, set $\Gamma_h(s_h, a_h) = \frac{CH \log(HSA/\delta)}{\sqrt{n_{s_h,a_h}}}$ if $n_{s_h,a_h} \geq 1$, o.w. set to $\frac{CH \log(HSA/\delta)}{1}$.
6:     Set $\widehat{Q}_h^p(\cdot, \cdot) \leftarrow \widehat{Q}_h(\cdot, \cdot) - \Gamma_h(\cdot, \cdot)$.             // Pessmistic update
7:     Set $\overline{Q}_h(\cdot, \cdot) \leftarrow \min\{\widehat{Q}_h^p(\cdot, \cdot), H-h+1\}^+$.
8:     Select $\widehat{\pi}_h(\cdot|s_h) \leftarrow \operatorname{argmax}_{\pi_h} \langle \overline{Q}_h(s_h, \cdot), \pi_h(\cdot|s_h) \rangle, \forall s_h$.
9:     Set $\widehat{V}_h(s_h) \leftarrow \langle \overline{Q}_h(s_h, \cdot), \widehat{\pi}_h(\cdot|s_h) \rangle, \forall s_h$.
10: **end for**
11: **Output:** $\{\widehat{\pi}_h\}$.

---

# A    On the statistical limits for Offline Learning and OPE in tabular RL

| Task | Dominate Bound | Type |
|------|----------------|------|
| Offline policy learning | $\sum_{h=1}^{H} \sum_{s_h, a_h} d_h^{\pi^\star}(s_h, a_h) \sqrt{\frac{\operatorname{Var}_{P_{s_h,a_h}}(V_{h+1}^\star + r_h)}{d_h^\mu(s_h, a_h)}} \sqrt{\frac{1}{n}}$ | Instance-dependent (Theorem 4.1, 4.3) |
| OPE ($|v^\pi - \hat{v}^\pi|$) | $\sqrt{\frac{1}{n} \sum_{h=0}^{H} \sum_{s_h, a_h} \frac{d_h^\pi(s_h, a_h)^2}{d_h^\mu(s_h, a_h)} \operatorname{Var}_{P_{s_h,a_h}}(V_{h+1}^\pi + r_h)}$ | Upper bound, Cramer-Rao lower bound |

Table 1: Showing the statistical optimalities for offline policy learning ($v^\star - v^{\widehat{\pi}}$) and offline policy evaluation (OPE) ($|v^\pi - \hat{v}^\pi|$) for the non-stationary tabular MDPs. The upper bound of OPE comes from Yin and Wang [2020], Duan et al. [2020] and the Cramer-Rao lower bound comes from Jiang and Li [2016].

Table A shows the statistical optimality for *offline policy learning* and *offline policy evaluation* (OPE) in the non-stationary tabular MDPs. By Cauchy-Schwartz inequality, it can be checked that the rate between the two bounds (roughly) deviate by a factor of $H$ (in terms of sample complexity), and this reveals that offline learning is inherently harder than OPE from the statistical aspect.

# B    Proof Overview and Some Notations

Our analysis of the intrinsic learning bound in Section 4 leverage the key design feature of APVI that $\widehat{V}_{h+1}$ only depends on the transition data from time $h+1$ to $H$ while $\widehat{P}_h$ only uses transition pairs at time $h$. This enables concentration inequalities due the *conditional* independence.[7] To cater for the data-adaptive bonus (4), we need to use *Empirical* Bernstein inequality to get $(\widehat{P}_h - P_h)\widehat{V}_{h+1} \lesssim \sqrt{\operatorname{Var}_{\widehat{P}}(\widehat{V}_{h+1})/n_{s_h,a_h}}$. Especially, to recover the $\sqrt{\operatorname{Var}_P(V_{h+1}^\star)}$ structure to we use a self-bounding reduction as follows. First, $\sqrt{\operatorname{Var}_{\widehat{P}}(\widehat{V}_{h+1})} - \sqrt{\operatorname{Var}_P(\widehat{V}_{h+1})} \lesssim H/\sqrt{n\bar{d}_m}$ and $\sqrt{\operatorname{Var}_P(\widehat{V}_{h+1})} - \sqrt{\operatorname{Var}_P(V_{h+1}^\star)} \leq ||\widehat{V}_{h+1} - V_{h+1}^\star||_\infty$. Next, we use (3) as the intermediate step to crude bounding $||\widehat{V}_{h+1} - V_{h+1}^\star||_\infty \lesssim H^2/\sqrt{n\bar{d}_m}$ (where "the use of (3)" is the more intricate self-bounding Lemma E.7 in the actual proof) and this yields the desired structure of $\sqrt{\operatorname{Var}_P(V_{h+1}^\star)} + H^2/\sqrt{n\bar{d}_m}$. Lastly, we can combine this with *the extended value difference lemma* in Cai et al. [2020] to bound $V_1^\star - \widehat{V}_1$ and leverage the pessimistic design for bounding $\widehat{V}_1 - V_1^{\widehat{\pi}}$.

For the per-instance lower bound, similar to Khamaru et al. [2021a], we reduce the problem from $\mathfrak{R}_n(\mathcal{P})$ to the two point testing problem and construct a problem-dependent local instance

---

[7]This trick is also leveraged in Yin et al. [2021a], but they consider the empirical optimal value $\widehat{V}^{\widehat{\pi}^\star}$ instead.

$P'_h(s_{h+1}|s_h,a_h) = P_h(s_{h+1}|s_h,a_h) + \frac{P_h(s_{h+1}|s_h,a_h)\left(V^\star_{h+1}(s_{h+1})-\mathbb{E}_{P_{s_h,a_h}}[V^\star_{h+1}]\right)}{8\sqrt{\zeta\cdot n_{s_h,a_h}\cdot\mathrm{Var}_{P_{s_h,a_h}}(V^\star_{h+1})}}$. The design with the subtraction of "the baseline" $\mathbb{E}_{P_{s_h,a_h}}[V^\star_{h+1}]$ is the key to make sure $P'$ center around the instance $P$.

For the assumption-free offline RL, the use of *pessimistic augmented MDP* help characterize the constant gap (due to the agnostic locations) via the following conclusion (Lemma E.2):

$$v^\pi - \sum_{h=1}^{H}\sum_{t=1}^{h-1}\sum_{(s_t,a_t)\in\mathcal{S}\times\mathcal{A}\setminus\mathcal{C}_h} d_t^\pi(s_t,a_t) \le v^\pi - \sum_{h=1}^{H} d_h^{\dagger\pi}(s_h^\dagger) \le v^{\dagger\pi} \le v^\pi.$$

Especially, the mass of the absorbing state $s_h^\dagger$ have the expression

$$d_h^{\dagger\pi}(s_h^\dagger) = \sum_{t=1}^{h-1}\sum_{(s_t,a_t)\in\mathcal{S}\times\mathcal{A}\setminus\mathcal{C}_t} d_t^{\dagger\pi}(s_t,a_t)$$

which absorbs all the first time exit probabilities $d_t^{\dagger\pi}(s_t,a_t)$ under $M^\dagger$, see Section E.2 for detailed explanations.

We use the following notations throughout the entire appendix. First recall $\mathcal{C}_h = \{(s_h,a_h) : d_h^\mu(s_h,a_h) > 0\}$ and $\bar{d}_m := \min_{h\in[H],(s_h,a_h)\in\mathcal{C}_h}\{d_h^\mu(s_h,a_h)\}$. Also, $\iota = \log(HSA/\delta)$. Next, for any $V \in \mathbb{R}^S$, denote $\mathcal{T}_h(V)(s,a) := r_h(s,a) + (P_h\cdot V)(s,a)\ \forall s,a \in \mathcal{S},\mathcal{A}$ be the Bellman update operator.

## C  Proof of VPVI (Theorem 3.1)

We begin with the following helpful lemma.

**Lemma C.1.** *For any $0 < \delta < 1$, there exists an absolute constant $c_1$ such that when total episode $n > c_1 \cdot 1/\bar{d}_m \cdot \log(HSA/\delta)$, then with probability $1 - \delta$, $\forall h \in [H]$*

$$n_{s_h,a_h} \ge n \cdot d_h^\mu(s_h,a_h)/2, \quad \forall\,(s_h,a_h) \in \mathcal{C}_h.$$

*Furthermore, we denote*

$$\mathcal{E} := \{n_{s_h,a_h} \ge n \cdot d_h^\mu(s_h,a_h)/2,\ \forall\,(s_h,a_h) \in \mathcal{C}_h,\ h \in [H].\} \tag{10}$$

*then equivalently $P(\mathcal{E}) > 1 - \delta$.*

*In addition, we denote*

$$\mathcal{E}' := \{n_{s_h,a_h} \le \frac{3}{2}n \cdot d_h^\mu(s_h,a_h),\ \forall\,(s_h,a_h) \in \mathcal{C}_h,\ h \in [H].\} \tag{11}$$

*then similarly $P(\mathcal{E}') > 1 - \delta$.*

*Proof of Lemma C.1.* Define $E := \{\exists h, (s_h,a_h) \in \mathcal{C}_h \text{ s.t. } n_{s_h,a_h} < nd_h^\mu(s_h,a_h)/2\}$. Then combining the first part of multiplicative Chernoff bound (Lemma J.1 in the Appendix) and a union bound, we obtain

$$\mathbb{P}[E] \le \sum_h \sum_{(s_h,a_h)\in\mathcal{C}_h} \mathbb{P}[n_{s_h,a_h} < nd_h^\mu(s_h,a_h)/2]$$
$$\le HSA \cdot e^{-\frac{n\cdot d_m}{8}} := \delta$$

solving this for $n$ then provides the stated result.

For $\mathcal{E}'$ we can similarly use the second part of Lemma J.1 to prove. $\blacksquare$

Now in Lemma J.10, take $\pi = \pi^\star$, $\widehat{Q}_h = \overline{Q}_h$ and $\widehat{\pi} = \widehat{\pi}$ in Algorithm 2, we have

$$V_1^{\pi^\star}(s) - V_1^{\widehat{\pi}}(s) \le \sum_{h=1}^{H}\mathbb{E}_{\pi^\star}\left[\xi_h(s_h,a_h)\mid s_1 = s\right] - \sum_{h=1}^{H}\mathbb{E}_{\widehat{\pi}}\left[\xi_h(s_h,a_h)\mid s_1 = s\right] \tag{12}$$

here $\xi_h(s,a) = (\mathcal{T}_h\widehat{V}_{h+1})(s,a) - \overline{Q}_h(s,a)$. This is true since by the definition of $\widehat{\pi}$ in Algorithm 2 $\langle\overline{Q}_h(s_h,\cdot), \pi_h(\cdot|s_h) - \widehat{\pi}_h(\cdot|s_h)\rangle \le 0$ almost surely. Next we prove the asymmetric bound for $\xi_h$, which is the key lemma for the proof.

**Lemma C.2.** *Denote $\xi_h(s,a) = (\mathcal{T}_h\widehat{V}_{h+1})(s,a) - \overline{Q}_h(s,a)$, where $\widehat{V}_{h+1}$ and $\overline{Q}_h$ are the quantities in Algorithm 2 and $\mathcal{T}_h(V) := r_h + P_h \cdot V$ for any $V$. Then with probability $1 - \delta$, then for any $h, s_h, a_h$ such that $d_h^\mu(s_h, a_h) > 0$, we have ($C'$ is an absolute constant)*

$$0 \le \xi_h(s_h, a_h) = (\mathcal{T}_h\widehat{V}_{h+1})(s_h, a_h) - \overline{Q}_h(s_h, a_h) \le C' \cdot \sqrt{\frac{H^2 \log(HSA/\delta)}{n \cdot d_h^\mu(s_h, a_h)}}.$$

*Proof of Lemma C.2.* Let us first consider the case where $n_{s_h, a_h} \ge 1$ for all $(s_h, a_h) \in \mathcal{C}_h$. In this case, by Hoeffding's inequality and a union bound, w.p. $1 - \delta$, since $0 \le r_h \le 1$,

$$|\widehat{r}_h(s_h, a_h) - r_h(s_h, a_h)| \le 2\sqrt{\frac{\log(HSA/\delta)}{n_{s_h, a_h}}} \; \forall (s_h, a_h) \in \mathcal{C}_h, h \in [H]. \tag{13}$$

Next, recall $\widehat{\pi}_{h+1}$ in Algorithm 2 is computed backwardly therefore only depends on sample tuple from time $h + 1$ to $H$. Aa a result $\widehat{V}_{h+1} = \langle \overline{Q}_{h+1}, \widehat{\pi}_{h+1} \rangle$ also only depends on the sample tuple from time $h + 1$ to $H$. On the other side, by our construction $\widehat{P}_h$ only depends on the transition pairs from $h$ to $h + 1$. Therefore $\widehat{V}_{h+1}$ and $\widehat{P}_h$ are *Conditionally* independent (This trick is also use in Yin et al. [2021a]) so by Hoeffding's inequality again[8] (note $||\widehat{V}_h||_\infty \le ||\overline{Q}_h|| \le H$ by VPVI)

$$\left| \left( (\widehat{P}_h - P_h)\widehat{V}_{h+1} \right)(s_h, a_h) \right| \le 2\sqrt{\frac{H^2 \cdot \log(HSA/\delta)}{n_{s_h, a_h}}}, \;\; \forall (s_h, a_h) \in \mathcal{C}_h. \tag{14}$$

Now apply Lemma C.1, we have with high probability the event $\mathcal{E}$ (10) is true, combining this with (13), (14) and rescaling the constants we obtain with probability $1 - \delta$, for all $h \in [H]$,

$$
\begin{aligned}
|\widehat{r}_h(s_h, a_h) - r_h(s_h, a_h)| &\le C\sqrt{\frac{\log(HSA/\delta)}{6n \cdot d_h^\mu(s_h, a_h)}} \\
\left| \left( (\widehat{P}_h - P_h)\widehat{V}_{h+1} \right)(s_h, a_h) \right| &\le C\sqrt{\frac{H^2 \cdot \log(HSA/\delta)}{6n \cdot d_h^\mu(s_h, a_h)}}, \;\; \forall (s_h, a_h) \in \mathcal{C}_h.
\end{aligned} \tag{15}
$$

Now we are ready to prove the Lemma.

**Step1:** we prove $\xi_h(s_h, a_h) \ge 0$ for all $(s_h, a_h) \in \mathcal{C}_h, h \in [H]$ with probability $1 - \delta$.

We can condition on $\mathcal{E}'$ and (15) is true since our lemma is high probability version. Indeed, if $\widehat{Q}_h^p(s_h, a_h) < 0$, then $\overline{Q}_h(s_h, a_h) = 0$. In this case, $\xi_h(s_h, a_h) = (\mathcal{T}_h\widehat{V}_{h+1})(s_h, a_h) \ge 0$. If $\widehat{Q}_h^p(s_h, a_h) \ge 0$, then by definition $\overline{Q}_h(s_h, a_h) = \min\{\widehat{Q}_h^p(s_h, a_h), H - h + 1\}^+ \le \widehat{Q}_h^p(s_h, a_h)$ and this implies

$$
\begin{aligned}
\xi_h(s_h, a_h) &\ge (\mathcal{T}_h\widehat{V}_{h+1})(s_h, a_h) - \widehat{Q}_h^p(s_h, a_h) \\
&= (r_h - \widehat{r}_h)(s_h, a_h) + (P_h - \widehat{P}_h)\widehat{V}_{h+1}(s_h, a_h) + \Gamma_h(s_h, a_h) \\
&\ge -2C\sqrt{\frac{H^2 \cdot \log(HSA/\delta)}{6n \cdot d_h^\mu(s_h, a_h)}} + \Gamma_h(s_h, a_h) \\
&\ge -C\sqrt{\frac{2H^2 \cdot \log(HSA/\delta)}{3n \cdot d_h^\mu(s_h, a_h)}} + C\sqrt{\frac{H^2 \cdot \log(HSA/\delta)}{3/2 \cdot n \cdot d_h^\mu(s_h, a_h)}} = 0
\end{aligned}
$$

where the second inequality uses (15) and the third inequality uses $\mathcal{E}'$.

**Step2:** we prove $\xi_h(s_h, a_h) \le C' \cdot \sqrt{\frac{H^2 \log(HSA/\delta)}{n \cdot d_h^\mu(s_h, a_h)}}$ for all $h \in [H], (s_h, a_h) \in \mathcal{C}_h$ with probability $1 - \delta$.

First, since the construction $\widehat{V}_h \le H - h + 1$ for all $h \in [H]$, this implies

$$\widehat{Q}_h^p = \widehat{Q}_h - \Gamma_h \le \widehat{Q}_h = \widehat{r}_h + (\widehat{P}_h\widehat{V}_{h+1}) \le 1 + (H - h) = H - h + 1$$

---

[8]It is worth mentioning if sub-policy $\widehat{\pi}_{h+1:t}$ depends on the data from all time steps $1, 2, \ldots, H$, then $\widehat{V}_{h+1}$ and $\widehat{P}_h$ are no longer conditionally independent and Hoeffding's inequality cannot be applied.

which uses $\widehat{r}_h \leq 1$ almost surely and $\widehat{P}_h$ is row-stochastic. Due to this, we have the equivalent definition

$$\overline{Q}_h := \min\{\widehat{Q}_h^p, H - h + 1\}^+ = \max\{\widehat{Q}_h^p, 0\} \geq \widehat{Q}_h^p.$$

Therefore

$$
\begin{aligned}
\xi_h(s_h, a_h) =& (\mathcal{T}_h \widehat{V}_{h+1})(s_h, a_h) - \overline{Q}_h(s_h, a_h) \leq (\mathcal{T}_h \widehat{V}_{h+1})(s_h, a_h) - \widehat{Q}_h^p(s_h, a_h) \\
=& (\mathcal{T}_h \widehat{V}_{h+1})(s_h, a_h) - \widehat{Q}_h(s_h, a_h) + \Gamma_h(s_h, a_h) \\
=& (r_h - \widehat{r}_h)(s_h, a_h) + (P_h - \widehat{P}_h)\widehat{V}_{h+1}(s_h, a_h) + \Gamma_h(s_h, a_h) \\
\leq& 2C\sqrt{\frac{H^2 \cdot \log(HSA/\delta)}{6n \cdot d_h^\mu(s_h, a_h)}} + \Gamma_h(s_h, a_h) \\
\leq& C\sqrt{\frac{2H^2 \cdot \log(HSA/\delta)}{3n \cdot d_h^\mu(s_h, a_h)}} + C\sqrt{\frac{2H^2 \cdot \log(HSA/\delta)}{n \cdot d_h^\mu(s_h, a_h)}} \\
=& (\sqrt{\tfrac{2}{3}} + \sqrt{2})C\sqrt{\frac{H^2 \cdot \log(HSA/\delta)}{n \cdot d_h^\mu(s_h, a_h)}} := C'\sqrt{\frac{H^2 \cdot \log(HSA/\delta)}{n \cdot d_h^\mu(s_h, a_h)}}
\end{aligned}
$$

where the first inequality uses (15) and the second one uses $P(\mathcal{E}) \geq 1 - \delta$ (10).

Combining Step 1 and Step 2 we finish the proof. ∎

Now we can finish proving the Theorem 3.1.

*Proof of Theorem 3.1.* Indeed, applying Lemma C.2 to (12) and average over initial distribution $s_1$, we obtain with probability $1 - \delta$

$$
\begin{aligned}
v^{\pi^\star} - v^{\widehat{\pi}} \leq& \sum_{h=1}^{H} \mathbb{E}_{\pi^\star}\left[\xi_h(s_h, a_h)\right] - \sum_{h=1}^{H} \mathbb{E}_{\widehat{\pi}}\left[\xi_h(s_h, a_h)\right] \\
\leq& \sum_{h=1}^{H} \mathbb{E}_{\pi^\star}\left[\xi_h(s_h, a_h)\right] - \sum_{h=1}^{H} \mathbb{E}_{\widehat{\pi}}\left[0\right] \\
\leq& C'H \sum_{h=1}^{H} \mathbb{E}_{\pi^\star}\left[\sqrt{\frac{\log(HSA/\delta)}{n \cdot d_h^\mu(s_h, a_h)}}\right] - 0 \\
=& C'H \sum_{h=1}^{H} \sum_{(s_h, a_h) \in \mathcal{C}_h} d_h^{\pi^\star}(s_h, a_h) \cdot \sqrt{\frac{\log(HSA/\delta)}{d_h^\mu(s_h, a_h)}} \cdot \sqrt{\frac{1}{n}}
\end{aligned}
$$

Note the second inequality is valid since by Line 5 of Algorithm 2 the Q-value at locations with $n_{s_h, a_h} = 0$ are heavily penalized with $O(H)$, hence the greedy $\widehat{\pi}$ will search at locations where $n_{s_h, a_h} > 0$ (which implies $d_h^\mu(s_h, a_h) > 0$). The third inequality is valid since $d_h^{\pi^\star}(s_h, a_h) > 0$ only if $d_h^\mu(s_h, a_h) > 0$. Therefore the expectation over $\pi^\star$, instead of summing over all $(s_h, a_h) \in \mathcal{S} \times \mathcal{A}$, is a sum over $(s_h, a_h)$ s.t. $d_h^\mu(s_h, a_h) > 0$. This completes the proof. ∎

# D  Discussion: the lower bound for single policy concentrability

To be rigorous, here we provide some detailed explanations of Rashidinejad et al. [2021]. In particular, we can mirror their construction to obtain the $\Omega(\sqrt{\frac{H^3 SC^\star}{n}})$ lower bound in the non-stationary finite horizon episodic setting. Indeed, their construction relies on the family with MDPs consisting of $S/4$ replicas of sub-MDPs with states $s_0, s_1, s_\oplus, s_\ominus$. There is an additional state $s_{-1}$ and in total there are $S + 1$ states. Here $s_0, s_\oplus, s_\ominus$ all have only 1 action $a_1$ and $s_1$ has two actions $a_1, a_2$ with transition $\mathbb{P}\left(s_\oplus^j \mid s_1^j, a_1\right) = \mathbb{P}\left(s_\ominus^j \mid s_1^j, a_1\right) = 1/2$, $\mathbb{P}\left(s_\oplus^j \mid s_1^j, a_2\right) = 1/2 + v_j\delta$ and $\mathbb{P}\left(s_\ominus^j \mid s_1^j, a_2\right) = 1/2 - v_j\delta$. $v_j \in \{-1, +1\}$ is the design choice w.r.t $j$-th replica and $\delta \in [0, 1/4]$.

$s_{-1}$ transition to itself with probability 1. The rewards for all of the states are 0 except $s_\oplus^j$ has reward 1 (See their Figure 5). In such a case, if $v_j = 1$, the optimal action at $s_1^j$ is $a_2$, otherwise, the optimal one is $a_1$. We can roughly create

$$d^\star\left(s_0^j\right) = O(\frac{1}{S}), \quad d^\star\left(s_1^j\right) = O(\frac{1}{HS})$$

$$d^\star\left(s_\oplus^j\right) = \frac{\left(\frac{1}{2}\mathbf{1}\{v_j = -1\} + \left(\frac{1}{2} + \delta\right)\mathbf{1}\{v_j = 1\}\right) \cdot H}{2} \cdot d^\star\left(s_1^j\right),$$

$$d^\star\left(s_\ominus^j\right) = \frac{\left(\frac{1}{2}\mathbf{1}\{v_j = 1\} + \left(\frac{1}{2} - \delta\right)\mathbf{1}\{v_j = -1\}\right) \cdot H}{2} \cdot d^\star\left(s_1^j\right), \quad d^\star(s_{-1}) = 0$$

and the behavior policy as

$$\mu_0\left(s_0^j\right) = \frac{d^\star\left(s_0^j\right)}{C^\star}, \quad \mu_0\left(s_1^j, a_2\right) = \frac{d^\star\left(s_1^j\right)}{C^\star}, \quad \mu_0\left(s_1^j, a_1\right) = d^\star\left(s_1^j\right) \cdot \left(1 - \frac{1}{C^\star}\right)$$

$$\mu_0\left(s_\oplus^j\right) = O(\frac{H}{C^\star} \cdot d^\star\left(s_1^j\right)), \quad \mu_0\left(s_\ominus^j\right) = O(\frac{H}{C^\star}) \cdot d^\star\left(s_1^j\right)$$

$$\mu_0(s_{-1}) = 1 - \sum_j \left(\mu_0\left(s_0^j\right) + \mu_0\left(s_1^j\right) + \mu_0\left(s_\oplus^j\right) + \mu_0\left(s_\ominus^j\right)\right)$$

By Fano's inequality, we can obtain: as long as $O(\frac{n\delta^2}{HSC^\star}) \leq 1$, then it holds $\inf_{\hat\pi} \sup_P \mathbb{E}\left[|v^\star - v^{\widehat\pi}|\right] \gtrsim H\delta$. One can set $\delta = O(\sqrt{\frac{HSC^\star}{n}})$ to obtain the result.

# E  Proof of Assumption-Free Offline Reinforcement Learning (Theorem 5.1)

Due to the assumption-free setting, the behavior policy $\mu$ is on longer guaranteed to trace any optimal policy $\pi^\star$. Therefore, in order to characterize the gap for the state-action agnostic space, we design the *pessimistic augmented MDP* $M^\dagger$ to reformulate the system so that the stat-actions that are agnostic to the behavior policy are subsumed into new state $s^\dagger$. Indeed, it comes from its optimistic counterpart which has a long history (*e.g.* RMAX exploration Brafman and Tennenholtz [2002], Jung and Stone [2010]). Recently, Liu et al. [2019], Kidambi et al. [2020], Buckman et al. [2021] leverage this idea for continuous offline policy optimization, but their use either does not follow the assumption-free regime (see Assumption 1 of Liu et al. [2019]) or is more empirically orientated [Buckman et al., 2021, Kidambi et al., 2020]. We find this helps to characterize the statistical gap when no assumption is made in offline RL, which provides a formal understanding of the hardness in distributional mismatches.

## E.1  Pessimistic augmented MDP

Let us define $M^\dagger$ use one extra state $s_h^\dagger$ for all $h \in \{2, \ldots, H\}$ with augmented state space $\mathcal{S}^\dagger = \mathcal{S} \cup \{s_h^\dagger\}$ and the transition and reward is defined as follows: (recall $\mathcal{C}_h := \{(s_h, a_h) : d_h^\mu(s_h, a_h) > 0\}$)

$$P_h^\dagger(\cdot \mid s_h, a_h) = \begin{cases} P_h(\cdot \mid s_h, a_h) & s_h, a_h \in \mathcal{C}_h, \\ \delta_{s_{h+1}^\dagger} & s_h = s_h^\dagger \text{ or } s_h, a_h \notin \mathcal{C}_h, \end{cases} \quad r^\dagger(s_h, a_h) = \begin{cases} r(s_h, a_h) & s_h, a_h \in \mathcal{C}_h \\ 0 & s_h = s_h^\dagger \text{ or } s_h, a_h \notin \mathcal{C}_h \end{cases}$$

and we further define for any $\pi$

$$V_h^{\dagger\pi}(s) = \mathbb{E}_\pi^\dagger\left[\sum_{t=h}^H r_t^\dagger \middle| s_h = s\right], v^{\dagger\pi} = \mathbb{E}_\pi^\dagger\left[\sum_{t=1}^H r_t^\dagger\right] \forall h \in [H]. \tag{16}$$

Furthermore, denote $\mathcal{K}_h := \{(s_h, a_h) : n_{s_h, a_h} > 0\}$, we also create a fictitious version $\widetilde{M}^\dagger$ with:

$$\widetilde{P}_h^\dagger(\cdot \mid s_h, a_h) = \begin{cases} P_h(\cdot \mid s_h, a_h) & s_h, a_h \in \mathcal{K}_h, \\ \delta_{s_{h+1}^\dagger} & s_h = s_h^\dagger \text{ or } s_h, a_h \notin \mathcal{K}_h, \end{cases} \quad \widetilde{r}^\dagger(s_h, a_h) = \begin{cases} r(s_h, a_h) & s_h, a_h \in \mathcal{K}_h \\ 0 & s_h = s_h^\dagger \text{ or } s_h, a_h \notin \mathcal{C}_h \end{cases}$$
$$\tag{17}$$

and the value functions under $\widetilde{M}^\dagger$ is similarly defined. Note in Section 5, we call (17) $M^\dagger$. However, it does not really matter since $\widetilde{M}^\dagger = M^\dagger$ with high probability, as stated in the following.

**Lemma E.1.** *For any $0 < \delta < 1$, there exists absolute constant $c$ s.t. when $n \geq c \cdot 1/\bar{d}_m \cdot \log(HSA/\delta)$,*

$$\mathbb{P}(\widetilde{M}^\dagger = M^\dagger) \geq 1 - \delta.$$

*Proof.* Note $\{\widetilde{M}^\dagger \neq M^\dagger\} \subset \{\exists \ d_h^\mu(s_h, a_h) > 0 \ and \ n_{s_h, a_h} = 0\}$. Similar to Lemma C.1, this happens with probability less than $\delta$ under the condition of $n$. ∎

We have the following theorem to characterize the difference between the augmented MDP $M^\dagger$ and the original MDP $M$.

**Theorem E.2.** *Denote $M^\dagger = \{\mathcal{S}, \mathcal{A}, H, r^\dagger, P^\dagger, d_1\}$ and for any $\pi$ denote $V_h^{\dagger\pi}$ be the value under $M^\dagger$. Then*

$$v^\pi - \sum_{h=2}^{H+1} \sum_{t=1}^{h-1} \sum_{(s_t, a_t) \in \mathcal{S} \times \mathcal{A} \setminus \mathcal{C}_h} d_t^\pi(s_t, a_t) \leq v^\pi - \sum_{h=2}^{H+1} d_h^{\dagger\pi}(s_h^\dagger) \leq v^{\dagger\pi} \leq v^\pi \tag{18}$$

Before proving Theorem E.2, we first prove the following helper Lemmas E.3, E.4.

**Lemma E.3.** $\forall h \in [H], (s_h, a_h) \in \mathcal{S} \times \mathcal{A}$, $d_h^\pi(s_h, a_h) \geq d_h^{\dagger\pi}(s_h, a_h)$.

*Proof of Lemma E.3.* There are two cases for $(s_h, a_h) \in \mathcal{S} \times \mathcal{A}$: either $(s_h, a_h) \in \mathcal{C}_h$ or $(s_h, a_h) \notin \mathcal{C}_h$.

**Step1:** by the definition of $P_h^\dagger$, it directly holds: for all $s_{h+1} \in \mathcal{S}$ and $(s_h, a_h) \in \mathcal{S} \times \mathcal{A}$, $P_h^\dagger(s_{h+1}|s_h, a_h) \leq P_h(s_{h+1}|s_h, a_h)$.

**Step2:** we prove the argument by induction. It is clear when $h = 1$ $d_1^\pi(s_1, a_1) = d_1^{\dagger\pi}(s_1, a_1)$ (since there is no $s_1^\dagger$). Then for any $(s_h, a_h) \in \mathcal{S} \times \mathcal{A}$,

$$d_{h+1}^\pi(s_{h+1}, a_{h+1}) = \sum_{s_h, a_h \in \mathcal{S} \times \mathcal{A}} P^\pi(s_{h+1}, a_{h+1}|s_h, a_h) d_h^\pi(s_h, a_h)$$

$$= \sum_{s_h, a_h \in \mathcal{S} \times \mathcal{A}} \pi(a_{h+1}|s_{h+1}) P_h^\pi(s_{h+1}|s_h, a_h) d_h^\pi(s_h, a_h)$$

$$\geq \sum_{s_h, a_h \in \mathcal{S} \times \mathcal{A}} \pi(a_{h+1}|s_{h+1}) P_h^{\dagger\pi}(s_{h+1}|s_h, a_h) d_h^\pi(s_h, a_h)$$

$$\geq \sum_{s_h, a_h \in \mathcal{S} \times \mathcal{A}} \pi(a_{h+1}|s_{h+1}) P_h^{\dagger\pi}(s_{h+1}|s_h, a_h) d_h^{\dagger\pi}(s_h, a_h)$$

$$= \sum_{s_h, a_h \in \mathcal{S} \times \mathcal{A}, s_h = s_h^\dagger} \pi(a_{h+1}|s_{h+1}) P_h^{\dagger\pi}(s_{h+1}|s_h, a_h) d_h^{\dagger\pi}(s_h, a_h) = d_{h+1}^{\dagger\pi}(s_{h+1}, a_{h+1}).$$

where the first inequality uses Step1, the second inequality uses induction assumption and the second to last equal sign uses $P_h^{\dagger\pi}(s_{h+1}|s_h^\dagger, a_h) = 0$ for $s_{h+1} \in \mathcal{S}$. By induction we conclude the proof for this lemma.

∎

Next we prove the second lemma that measures $d_h^{\dagger\pi}(s_h^\dagger)$.

**Lemma E.4.** *For all $h \in [2, H+1]$, $d_h^{\dagger\pi}(s_h^\dagger) = \sum_{t=1}^{h-1} \sum_{(s_t, a_t) \in \mathcal{S} \times \mathcal{A} \setminus \mathcal{C}_t} d_t^{\dagger\pi}(s_t, a_t)$.*

*Proof of Lemma E.4.* Indeed,

$$d_{h+1}^{\dagger\pi}(s_{h+1}^{\dagger}) = \sum_{a_{h+1}} d_{h+1}^{\dagger\pi}(s_{h+1}^{\dagger}, a_{h+1})$$

$$= \sum_{a_{h+1}} \sum_{(s_h,a_h)\notin\mathcal{C}_h, s_h=s_h^{\dagger}} P^{\dagger}(s_{h+1}^{\dagger}, a_{h+1} \mid s_h, a_h) d_h^{\dagger\pi}(s_h, a_h)$$

$$= \sum_{a_{h+1}} \left( \sum_{(s_h,a_h)\notin\mathcal{C}_h} P^{\dagger}(s_{h+1}^{\dagger}, a_{h+1} \mid s_h, a_h) d_h^{\dagger\pi}(s_h, a_h) + \sum_{a_h} P^{\dagger}(s_{h+1}^{\dagger}, a_{h+1} \mid s_h^{\dagger}, a_h) d_h^{\dagger\pi}(s_h^{\dagger}, a_h) \right)$$

$$= \sum_{a_{h+1}} \left( \sum_{(s_h,a_h)\notin\mathcal{C}_h} P^{\dagger}(s_{h+1}^{\dagger}, a_{h+1} \mid s_h, a_h) d_h^{\dagger\pi}(s_h, a_h) + \sum_{a_h} \pi(a_{h+1} \mid s_{h+1}^{\dagger}) d_h^{\dagger\pi}(s_h^{\dagger}, a_h) \right)$$

$$= \sum_{a_{h+1}} \left( \sum_{(s_h,a_h)\notin\mathcal{C}_h} P^{\dagger}(s_{h+1}^{\dagger}, a_{h+1} \mid s_h, a_h) d_h^{\dagger\pi}(s_h, a_h) \right) + d_h^{\dagger\pi}(s_h^{\dagger})$$

$$= \sum_{a_{h+1}} \left( \sum_{(s_h,a_h)\notin\mathcal{C}_h} \pi(a_{h+1} \mid s_{h+1}^{\dagger}) d_h^{\dagger\pi}(s_h, a_h) \right) + d_h^{\dagger\pi}(s_h^{\dagger}) = \sum_{(s_h,a_h)\notin\mathcal{C}_h} d_h^{\dagger\pi}(s_h, a_h) + d_h^{\dagger\pi}(s_h^{\dagger}).$$

Apply the above recursively we obtain the result. ∎

Now we are ready to prove Theorem E.2.

*Proof of Theorem E.2.* **Step1:** we first show $v^{\dagger\pi} \leq v^{\pi}$.

Consider the stopping time $T = \inf\{t : s.t. (s_t, a_t) \notin \mathcal{C}_h\} \wedge H$. Then $1 \leq T \leq H$.

$$v^{\pi} = E_{\pi}\left[\sum_{h=1}^{H} r(s_h, a_h)\right] = E_{\pi}\left[\sum_{h=1}^{T-1} r(s_h, a_h) + \sum_{h=T}^{H} r(s_h, a_h)\right]$$

$$= E_{\pi}^{\dagger}\left[\sum_{h=1}^{T-1} r(s_h, a_h)\right] + E_{\pi}\left[\sum_{h=T}^{H} r(s_h, a_h)\right] \geq E_{\pi}^{\dagger}\left[\sum_{h=1}^{T-1} r(s_h, a_h)\right] + E_{\pi}\left[\sum_{h=T}^{H} 0\right]$$

$$= E_{\pi}^{\dagger}\left[\sum_{h=1}^{T-1} r(s_h, a_h)\right] + E_{\pi}^{\dagger}\left[\sum_{h=T}^{H} 0\right] = E_{\pi}^{\dagger}\left[\sum_{h=1}^{T-1} r(s_h, a_h)\right] + E_{\pi}^{\dagger}\left[\sum_{h=T}^{H} r(s_h, a_h)\right] = v^{\dagger\pi},$$

where the third and the fourth equal signs use the distribution of $T$ is identical under either $M$ or $M^{\dagger}$ by construction. The fifth equal sign uses the definition of pessimistic reward.

**Step2:** Next we show

$$v^{\pi} \leq v^{\dagger\pi} + \sum_{h=2}^{H+1} d_h^{\dagger\pi}(s_h^{\dagger}) \leq v^{\dagger\pi} + \sum_{h=2}^{H+1} \sum_{t=1}^{h-1} \sum_{(s_t,a_t)\in\mathcal{S}\times\mathcal{A}\backslash\mathcal{C}_t} d_t^{\pi}(s_t, a_t). \tag{19}$$

Indeed,

$$v^\pi = \sum_{h=1}^H \sum_{(s_h,a_h)\in\mathcal{S}\times\mathcal{A}} d_h^\pi(s_h,a_h) r(s_h,a_h)$$

$$=\sum_{h=1}^H \sum_{(s_h,a_h)\in\mathcal{S}\times\mathcal{A}} \left(d_h^\pi(s_h,a_h) - d_h^{\dagger\pi}(s_h,a_h)\right) r(s_h,a_h) + \sum_{h=1}^H \sum_{(s_h,a_h)\in\mathcal{S}\times\mathcal{A}} d_h^{\dagger\pi}(s_h,a_h) r(s_h,a_h)$$

$$\leq\sum_{h=1}^H \sum_{(s_h,a_h)\in\mathcal{S}\times\mathcal{A}} \left(d_h^\pi(s_h,a_h) - d_h^{\dagger\pi}(s_h,a_h)\right) \cdot 1 + \sum_{h=1}^H \sum_{(s_h,a_h)\in\mathcal{S}\times\mathcal{A}} d_h^{\dagger\pi}(s_h,a_h) r(s_h,a_h)$$

$$=\sum_{h=1}^H \left(1 - \sum_{(s_h,a_h)\in\mathcal{S}\times\mathcal{A}} d_h^{\dagger\pi}(s_h,a_h)\right) + \sum_{h=1}^H \sum_{(s_h,a_h)\in\mathcal{S}\times\mathcal{A}} d_h^{\dagger\pi}(s_h,a_h) r(s_h,a_h)$$

$$=\sum_{h=2}^H d_h^{\dagger\pi}(s_h^\dagger) + \sum_{h=1}^H \sum_{(s_h,a_h)\in\mathcal{S}\times\mathcal{A}} d_h^{\dagger\pi}(s_h,a_h) r(s_h,a_h)$$

$$=\sum_{h=2}^H d_h^{\dagger\pi}(s_h^\dagger) + \sum_{h=1}^H \sum_{(s_h,a_h)\in\mathcal{S}\times\mathcal{A}} d_h^{\dagger\pi}(s_h,a_h) \left(r(s_h,a_h) - r^\dagger(s_h,a_h)\right) + \sum_{h=1}^H \sum_{(s_h,a_h)\in\mathcal{S}\times\mathcal{A}} d_h^{\dagger\pi}(s_h,a_h) r^\dagger(s_h,a_h)$$

$$=\sum_{h=2}^H d_h^{\dagger\pi}(s_h^\dagger) + \sum_{h=1}^H \sum_{(s_h,a_h)\notin\mathcal{C}_h} d_h^{\dagger\pi}(s_h,a_h) \left(r(s_h,a_h) - r^\dagger(s_h,a_h)\right) + \sum_{h=1}^H \sum_{(s_h,a_h)\in\mathcal{S}\times\mathcal{A}} d_h^{\dagger\pi}(s_h,a_h) r^\dagger(s_h,a_h)$$

$$=\sum_{h=2}^H d_h^{\dagger\pi}(s_h^\dagger) + \sum_{h=1}^H \sum_{(s_h,a_h)\notin\mathcal{C}_h} d_h^{\dagger\pi}(s_h,a_h) \left(r(s_h,a_h) - r^\dagger(s_h,a_h)\right) + v^{\dagger\pi}$$

$$\leq\sum_{h=2}^H d_h^{\dagger\pi}(s_h^\dagger) + \sum_{h=1}^H \sum_{(s_h,a_h)\notin\mathcal{C}_h} d_h^{\dagger\pi}(s_h,a_h) \cdot 1 + v^{\dagger\pi} = \sum_{h=2}^{H+1} d_h^{\dagger\pi}(s_h^\dagger) + v^{\dagger\pi}$$

The first inequality is due to Lemma E.3. The fourth equal sign uses $d_1^\dagger(s_1^\dagger) = 0$. The sixth equal sign is due to $r(s_h,a_h) = r^\dagger(s_h,a_h)$ when $(s_h,a_h) \in \mathcal{C}_h$. The seventh equal sign is due to $r^\dagger(s_h^\dagger, a_h) = 0$. The last equal sign uses Lemma E.4. The right inequality in (19) uses Lemma E.3. Step 1 and Step 2 conclude the proof of Theorem E.2. ∎

### E.1.1 Strong adaptive assumption-free bound

Now we are ready to launch the *assumption-free* AVPI (Algorithm 1) with the following model-based construction $\widehat{M}^\dagger$ (recall $\mathcal{K}_h := \{(s_h,a_h) : n_{s_h,a_h} > 0\}$):

$$\widehat{P}_h^\dagger(\cdot \mid s_h, a_h) = \begin{cases} \widehat{P}_h(\cdot \mid s_h, a_h) & s_h, a_h \in \mathcal{K}_h, \\ \delta_{s_{h+1}^\dagger} & s_h = s_h^\dagger \text{ or } s_h, a_h \notin \mathcal{K}_h, \end{cases} \quad \widehat{r}^\dagger(s_h, a_h) = \begin{cases} \widehat{r}(s_h, a_h) & s_h, a_h \in \mathcal{S}\times\mathcal{A} \\ 0 & s_h = s_h^\dagger \text{ or } s_h, a_h \notin \mathcal{C}_h \end{cases}$$

where $\widehat{P}, \widehat{r}$ is defined as

$$\widehat{P}_h(s'|s_h,a_h) = \frac{\sum_{\tau=1}^n \mathbf{1}[(s_{h+1}^\tau, a_h^\tau, s_h^\tau) = (s', s_h, a_h)]}{n_{s_h,a_h}}, \quad \widehat{r}_h(s_h,a_h) = \frac{\sum_{\tau=1}^n \mathbf{1}[(a_h^\tau, s_h^\tau) = (s_h, a_h)] \cdot r_h^\tau}{n_{s_h,a_h}},$$

(20)

The benefit of using $\widetilde{M}^\dagger$ (17) is that in $\widetilde{M}^\dagger$ there is no agnostic location even no assumption is made. The $\widehat{M}^\dagger$ creates a empirical estimate for $\widetilde{M}^\dagger$. In this case, the pessimistic bonus is designed as

$$\Gamma_h(s_h,a_h) = 2\sqrt{\frac{\mathrm{Var}_{\widehat{P}_{s_h,a_h}^\dagger}(\widehat{r}_h^\dagger + \widehat{V}_{h+1}) \cdot \iota}{n_{s_h,a_h}}} + \frac{14H \cdot \iota}{3n_{s_h,a_h}}$$

if $n_{s_h,a_h} \in \mathcal{K}_h$ and 0 otherwise (here $\widehat{V}_{h+1}$ is computed backwardly from the next time step in Algorithm 1). Now let us start the proof. First of all, let us assume $\widetilde{M}^\dagger = M^\dagger$ for the moment so we

can get rid of the tilde expression for notation convenience. We will formally recover the result for $M^\dagger$ at the end by Lemma E.1.

In particular, while we always use $\pi^\star$ to denote the optimal policy in the *Original* MDP, we augment it in the $M^\dagger(\widetilde{M}^\dagger)$ arbitrarily and abuse the notation as:

$$\pi^\star(\cdot|s_h) = \begin{cases} \pi^\star(\cdot|s_h) & s_h \in \mathcal{S} \\ arbitrary\ distribution & s_h = s_h^\dagger \end{cases} \tag{21}$$

and always use $\widehat{\pi}$ to denote the output of Algorithm 1. We rely on the following lemma that characterize the suboptimality gap.

**Lemma E.5.** *Recall $\pi^\star$ in* (21) *and define* $(\mathcal{T}_h^\dagger V)(\cdot,\cdot) := r_h^\dagger(\cdot,\cdot) + (P_h^\dagger V)(\cdot,\cdot)$ *for any $V \in \mathbb{R}^{S+1}$. Note $\widehat{\pi}, \overline{Q}_h, \widehat{V}_h$ are defined in Algorithm 1 and denote $\xi_h^\dagger(s,a) = (\mathcal{T}_h^\dagger \widehat{V}_{h+1})(s,a) - \overline{Q}_h(s,a)$.*

$$V_1^{\dagger\pi^\star}(s) - V_1^{\dagger\widehat{\pi}}(s) \le \sum_{h=1}^H \mathbb{E}_{\pi^\star}^\dagger\left[\xi_h^\dagger(s_h,a_h) \mid s_1 = s\right] - \sum_{h=1}^H \mathbb{E}_{\widehat{\pi}}^\dagger\left[\xi_h^\dagger(s_h,a_h) \mid s_1 = s\right]. \tag{22}$$

*where $V_1^{\dagger\pi}$ is defined in* (16). *Furthermore,* (22) *holds for all $V_h^{\dagger\pi^\star}(s) - V_h^{\dagger\widehat{\pi}}(s)$.*

*Proof of Lemma E.5.* Apply Lemma J.10 with $\mathcal{T}_h = \mathcal{T}_h^\dagger$, $\pi = \pi^\star$, $\widehat{Q}_h = \overline{Q}_h$ and $\widehat{\pi} = \widehat{\pi}$ in Algorithm 1, we can obtain the result since by the definition of $\widehat{\pi}$ in Algorithm 1 $\langle \overline{Q}_h(s_h,\cdot), \pi_h(\cdot|s_h) - \widehat{\pi}_h(\cdot|s_h)\rangle \le 0$ almost surely for any $\pi$. The proof for $V_h^{\dagger\pi^\star}(s) - V_h^{\dagger\widehat{\pi}}(s)$ is identical. ∎

Next we prove the adaptive asymmetric bound for $\xi_h^\dagger$, which is the key for recover the structure of intrinsic bound.

**Lemma E.6.** *Denote $\xi_h^\dagger(s,a) = (\mathcal{T}_h^\dagger \widehat{V}_{h+1})(s,a) - \overline{Q}_h(s,a)$, where $\widehat{V}_{h+1}$ and $\overline{Q}_h$ are the quantities in Algorithm 1 and $\mathcal{T}_h^\dagger(V) := r_h^\dagger + P_h^\dagger \cdot V$ for any $V \in \mathbb{R}^{S+1}$. Then with probability $1 - \delta$, then for any $h, s_h, a_h$ such that $n_{s_h,a_h} > 0$, we have*

$$0 \le \xi_h^\dagger(s_h,a_h) = (\mathcal{T}_h^\dagger \widehat{V}_{h+1})(s_h,a_h) - \overline{Q}_h(s_h,a_h)$$

$$\le 4\sqrt{\frac{\text{Var}_{\widehat{P}_{s_h,a_h}^\dagger}(\widehat{r}_h^\dagger + \widehat{V}_{h+1}) \cdot \log(HSA/\delta)}{n_{s_h,a_h}}} + \frac{28H \cdot \log(HSA/\delta)}{3n_{s_h,a_h}}$$

*Proof of Lemma E.6.* Recall we are under $M^\dagger (\widehat{M}^\dagger)$. For all $(s_h,a_h) \in \mathcal{K}_h$, by Empirical Bernstein inequality (Lemma J.4) and a union bound[9], w.p. $1 - \delta$, since $0 \le r_h^\dagger \le 1$,

$$|\widehat{r}_h^\dagger(s_h,a_h) - r_h^\dagger(s_h,a_h)| \le \sqrt{\frac{2\text{Var}_{\widehat{P}^\dagger}(\widehat{r}_h^\dagger)\log(HSA/\delta)}{n_{s_h,a_h}}} + \frac{7\log(HSA/\delta)}{3n_{s_h,a_h}} \ \forall (s_h,a_h) \in \mathcal{K}_h, h \in [H]. \tag{23}$$

Next, recall $\widehat{\pi}_{h+1}$ in Algorithm 1 is computed backwardly therefore only depends on sample tuple from time $h + 1$ to $H$. Aa a result $\widehat{V}_{h+1} = \langle \overline{Q}_{h+1}, \widehat{\pi}_{h+1}\rangle$ also only depends on the sample tuple from time $h + 1$ to $H$. On the other side, by our construction $\widehat{P}_h^\dagger$ only depends on the transition pairs from $h$ to $h + 1$. Therefore $\widehat{V}_{h+1}$ and $\widehat{P}_h^\dagger$ are *Conditionally* independent (This trick is also use in Yin et al. [2021a]) so by Empirical Bernstein inequality again[10] and a union bound (note $||\widehat{V}_h||_\infty \le ||\overline{Q}_h|| \le H$ by APVI) for all $(s_h,a_h) \in \mathcal{K}_h$, w.p. $1 - \delta$,

$$\left|\left((\widehat{P}_h^\dagger - P_h^\dagger)\widehat{V}_{h+1}\right)(s_h,a_h)\right| \le \sqrt{\frac{2\text{Var}_{\widehat{P}_{s_h,a_h}^\dagger}(\widehat{V}_{h+1}) \cdot \log(HSA/\delta)}{n_{s_h,a_h}}} + \frac{7H \cdot \log(HSA/\delta)}{3n_{s_h,a_h}}. \tag{24}$$

---

[9]Here note even though $|\mathcal{S}^\dagger| = S + 1$, for state $s_h^\dagger$ we always have $n_{s_h^\dagger,a_h} = 0$ for any $a_h$. Therefore apply the union bound only provides $HSA$ in th log term instead of $H(S+1)A$.

[10]It is worth mentioning if sub-policy $\widehat{\pi}_{h+1:t}$ depends on the data from all time steps $1, 2, \ldots, H$, then $\widehat{V}_{h+1}$ and $\widehat{P}_h$ are no longer conditionally independent and Hoeffding's inequality cannot be applied.

Now we are ready to prove the Lemma.

**Step1:** we prove $\xi_h(s_h, a_h) \geq 0$ for all $(s_h, a_h) \in \mathcal{K}_h$, $h \in [H]$ with probability $1 - \delta$.

Indeed, if $\widehat{Q}_h^p(s_h, a_h) < 0$, then $\overline{Q}_h(s_h, a_h) = 0$. In this case, $\xi_h(s_h, a_h) = (\mathcal{T}_h \widehat{V}_{h+1})(s_h, a_h) \geq 0$ (note $\widehat{V}_h \geq 0$ by the definition). If $\widehat{Q}_h^p(s_h, a_h) \geq 0$, then by definition $\overline{Q}_h(s_h, a_h) = \min\{\widehat{Q}_h^p(s_h, a_h), H - h + 1\}^+ \leq \widehat{Q}_h^p(s_h, a_h)$ and this implies

$$
\begin{aligned}
&\xi_h^\dagger(s_h, a_h) \geq (\mathcal{T}_h^\dagger \widehat{V}_{h+1})(s_h, a_h) - \widehat{Q}_h^p(s_h, a_h) \\
&= (r_h^\dagger - \widehat{r}_h^\dagger)(s_h, a_h) + (P_h^\dagger - \widehat{P}_h^\dagger)\widehat{V}_{h+1}(s_h, a_h) + \Gamma_h(s_h, a_h) \\
&\geq -2\sqrt{\frac{\mathrm{Var}_{\widehat{P}_{s_h, a_h}^\dagger}(\widehat{r}_h^\dagger + \widehat{V}_{h+1}) \cdot \log(HSA/\delta)}{n_{s_h, a_h}}} - \frac{14H \cdot \log(HSA/\delta)}{3n_{s_h, a_h}} + \Gamma_h(s_h, a_h) = 0
\end{aligned}
$$

where the inequality uses (23), (24) and $\sqrt{a} + \sqrt{b} \leq \sqrt{2(a + b)}$ and $r_h$ and $s_{h+1}$ are conditionally independent given $s_h, a_h$. The last equal sign uses Line 6 of Algorithm 1.

**Step2:** we prove $\xi_h^\dagger(s_h, a_h) \leq 4\sqrt{\frac{\mathrm{Var}_{\widehat{P}_{s_h, a_h}^\dagger}(\widehat{r}_h^\dagger + \widehat{V}_{h+1}) \cdot \log(HSA/\delta)}{n_{s_h, a_h}}} + \frac{28H \cdot \log(HSA/\delta)}{3n_{s_h, a_h}}$ for all $h \in [H], (s_h, a_h) \in \mathcal{K}_h$ with probability $1 - \delta$.

First, since by construction $\widehat{V}_h \leq H - h + 1$ for all $h \in [H]$, this implies

$$
\widehat{Q}_h^p = \widehat{Q}_h - \Gamma_h \leq \widehat{Q}_h = \widehat{r}_h^\dagger + (\widehat{P}_h^\dagger \widehat{V}_{h+1}) \leq 1 + (H - h) = H - h + 1
$$

which uses $\widehat{r}_h^\dagger \leq 1$ almost surely and $\widehat{P}_h^\dagger$ is row-stochastic. Due to this, we have the equivalent definition

$$
\overline{Q}_h := \min\{\widehat{Q}_h^p, H - h + 1\}^+ = \max\{\widehat{Q}_h^p, 0\} \geq \widehat{Q}_h^p.
$$

Therefore

$$
\begin{aligned}
&\xi_h^\dagger(s_h, a_h) = (\mathcal{T}_h^\dagger \widehat{V}_{h+1})(s_h, a_h) - \overline{Q}_h(s_h, a_h) \leq (\mathcal{T}_h^\dagger \widehat{V}_{h+1})(s_h, a_h) - \widehat{Q}_h^p(s_h, a_h) \\
&= (\mathcal{T}_h^\dagger \widehat{V}_{h+1})(s_h, a_h) - \widehat{Q}_h(s_h, a_h) + \Gamma_h(s_h, a_h) \\
&= (r_h^\dagger - \widehat{r}_h^\dagger)(s_h, a_h) + (P_h^\dagger - \widehat{P}_h^\dagger)\widehat{V}_{h+1}(s_h, a_h) + \Gamma_h(s_h, a_h) \\
&\leq 2\sqrt{\frac{\mathrm{Var}_{\widehat{P}_{s_h, a_h}^\dagger}(\widehat{r}_h^\dagger + \widehat{V}_{h+1}) \cdot \log(HSA/\delta)}{n_{s_h, a_h}}} + \frac{14H \cdot \log(HSA/\delta)}{3n_{s_h, a_h}} + \Gamma_h(s_h, a_h) \\
&= 4\sqrt{\frac{\mathrm{Var}_{\widehat{P}_{s_h, a_h}^\dagger}(\widehat{r}_h^\dagger + \widehat{V}_{h+1}) \cdot \log(HSA/\delta)}{n_{s_h, a_h}}} + \frac{28H \cdot \log(HSA/\delta)}{3n_{s_h, a_h}}.
\end{aligned}
$$

Combining Step 1 and Step 2 we finish the proof. ∎

### E.1.2 Proof of Theorem 5.1

Now we are ready to prove the Theorem 5.1.

First of all, by Lemma E.5 and Lemma E.6, for all $t \in [H]$, $s \in \mathcal{S}$ (excluding $s^\dagger$) w.p. $1 - \delta$

$$V_t^{\dagger \pi^\star}(s) - V_t^{\dagger \widehat{\pi}}(s) \le \sum_{h=t}^H \mathbb{E}_{\pi^\star}^\dagger \left[ \xi_h^\dagger(s_h, a_h) \mid s_t = s \right] - \sum_{h=t}^H \mathbb{E}_{\widehat{\pi}}^\dagger \left[ \xi_h^\dagger(s_h, a_h) \mid s_t = s \right]$$

$$\le \sum_{h=t}^H \mathbb{E}_{\pi^\star}^\dagger \left[ \xi_h^\dagger(s_h, a_h) \mid s_t = s \right] - 0$$

$$\le \sum_{h=t}^H \mathbb{E}_{\pi^\star}^\dagger \left[ 4\sqrt{\frac{\mathrm{Var}_{\widehat{P}_{s_h,a_h}^\dagger}(\widehat{r}_h^\dagger + \widehat{V}_{h+1}) \cdot \iota}{n_{s_h,a_h}}} + \frac{28H \cdot \iota}{3n_{s_h,a_h}} \mid s_t = s \right] \tag{25}$$

$$\le \sum_{h=t}^H \mathbb{E}_{\pi^\star}^\dagger \left[ 4\sqrt{\frac{2\mathrm{Var}_{\widehat{P}_{s_h,a_h}^\dagger}(\widehat{r}_h^\dagger + \widehat{V}_{h+1}) \cdot \iota}{nd_h^\mu(s_h, a_h)}} + \frac{56H \cdot \iota}{3nd_h^\mu(s_h, a_h)} \mid s_t = s \right]$$

here recall the expectation is only taken over $s_h, a_h$. Note by the Pessimistic MDP $\widetilde{M}^\dagger$ ($\widehat{M}^\dagger$), for all $(s_h, a_h) \notin \mathcal{K}_h$ and $s_h^\dagger$, the pessimistic reward leads to $Q^{\dagger \pi}(s_h, a_h), V^{\dagger \pi}(s_h^\dagger) = 0$ for any $\pi$, therefore Lemma E.6 can be applied. Moreover, the last inequality is by Lemma C.1.

**Lemma E.7** (self-bounding). *We prove, for all $t \in [H]$, w.p. $1 - \delta$, for all $s \in \mathcal{S}$ (excluding $s^\dagger$),*

$$\left| V_t^{\dagger \pi^\star}(s) - \widehat{V}_t(s) \right| \le \frac{8\sqrt{2\iota}H^2}{\sqrt{n \cdot \bar{d}_m}} + \frac{112H^2 \cdot \iota}{3n \cdot \bar{d}_m}.$$

*where $\bar{d}_m$ is defined in Theorem 5.1.*

**Remark E.8.** *The self-bounding lemma essentially provides a crude high probability bound for $|V_t^{\dagger \pi^\star} - \widehat{V}_t|$ (or $|V_t^{\dagger \pi^\star} - V_t^{\dagger \widehat{\pi}}|$) with suboptimal order $\widetilde{O}(\frac{H^2}{\sqrt{n\bar{d}_m}})$ and we can use it to further bound the higher order term in the main result.*

*Proof of Lemma E.7.* Indeed, by (25), since $\mathrm{Var}_{\widehat{P}_{s_h,a_h}^\dagger}(\widehat{r}_h^\dagger + \widehat{V}_{h+1}) \le H^2$, we have w.p. $1 - \delta$,

$$\left| V_t^{\dagger \pi^\star}(s) - V_t^{\dagger \widehat{\pi}}(s) \right| \le \frac{4\sqrt{2\iota}H^2}{\sqrt{n \cdot \bar{d}_m}} + \frac{56H^2 \cdot \iota}{3n \cdot \bar{d}_m} \tag{26}$$

for all $t \in [H]$. Next, when apply Lemma J.10 to Lemma E.5, by (46) and (47) we essentially obtain

$$V_t^{\dagger \pi^\star}(s) - \widehat{V}_t(s) = \sum_{h=t}^H \mathbb{E}_{\pi^\star}^\dagger \left[ \xi_h^\dagger(s_h, a_h) \mid s_t = s \right] + \sum_{h=t}^H \mathbb{E}_{\pi^\star}^\dagger \left[ \langle \widehat{Q}_h(s_h, \cdot), \pi_h^\star(\cdot|s_h) - \widehat{\pi}_h(\cdot|s_h) \rangle \mid s_t = s \right]$$

$$\le \frac{4\sqrt{2\iota}H^2}{\sqrt{n \cdot \bar{d}_m}} + \frac{56H^2 \cdot \iota}{3n \cdot \bar{d}_m} + 0$$

and

$$\widehat{V}_t(s) - V_t^{\dagger \widehat{\pi}}(s) = - \sum_{h=t}^H \mathbb{E}_{\widehat{\pi}}^\dagger \left[ \xi_h^\dagger(s_h, a_h) \mid s_t = s \right] \ge 0.$$

Combing those two with (26) we obtain the result.

∎

**Lemma E.9.** *For all $(a_h, a_h) \in \mathcal{K}_h$ and any $||V||_\infty \le H$, w.p. $1 - \delta$,*

$$\sqrt{\mathrm{Var}_{\widehat{P}_{s_h,a_h}^\dagger}(V)} \le 6H\sqrt{\frac{\iota}{n \cdot d_h^\mu(s_h, a_h)}} + \sqrt{\mathrm{Var}_{P_{s_h,a_h}^\dagger}(V)}.$$

*Proof.* This is a direct application of Lemma J.8 with a union bound. Specifically, we apply $\frac{n-1}{n} \le 1$.

∎

Now by Lemma E.7 and Lemma E.9, for all $(s_h, a_h) \in \mathcal{K}_h$, w.p. $1 - \delta$,

$$\sqrt{\text{Var}_{\widehat{P}^{\dagger}_{s_h, a_h}}(\widehat{r}^{\dagger}_h + \widehat{V}_{h+1})} \leq \sqrt{\text{Var}_{P^{\dagger}_{s_h, a_h}}(\widehat{r}^{\dagger}_h + \widehat{V}_{h+1})} + 6H\sqrt{\frac{\iota}{n \cdot d^{\mu}_h(s_h, a_h)}}$$

$$\leq \sqrt{\text{Var}_{P^{\dagger}_{s_h, a_h}}(r^{\dagger}_h + V^{\dagger \pi^{\star}}_{h+1})} + \left\|(\widehat{r}^{\dagger}_h + \widehat{V}_{h+1}) - (r^{\dagger}_h + V^{\dagger \pi^{\star}}_{h+1})\right\|_{\infty, s \in \mathcal{S}} + 6H\sqrt{\frac{\iota}{n \cdot d^{\mu}_h(s_h, a_h)}}$$

$$\leq \sqrt{\text{Var}_{P^{\dagger}_{s_h, a_h}}(r^{\dagger}_h + V^{\dagger \pi^{\star}}_{h+1})} + \frac{10\sqrt{2\iota}H^2}{\sqrt{n \cdot \bar{\bar{d}}_m}} + \frac{112H^2 \cdot \iota}{3n \cdot \bar{\bar{d}}_m} + 6H\sqrt{\frac{\iota}{n \cdot d^{\mu}_h(s_h, a_h)}}$$

Therefore plug this into (25), and average over $s_1$, we finally get, w.p. $1 - \delta$,

$$v^{\dagger \pi^{\star}} - v^{\dagger \widehat{\pi}} \leq \sum_{h=1}^{H} \mathbb{E}^{\dagger}_{\pi^{\star}} \left[ 4\sqrt{\frac{2\text{Var}_{\widehat{P}^{\dagger}_{s_h, a_h}}(\widehat{r}^{\dagger}_h + \widehat{V}_{h+1}) \cdot \iota}{nd^{\mu}_h(s_h, a_h)}} + \frac{56H \cdot \iota}{3nd^{\mu}_h(s_h, a_h)} \mid s_1 = s \right]$$

$$\leq C' \sum_{h=1}^{H} \mathbb{E}^{\dagger}_{\pi^{\star}} \left[ \sqrt{\frac{\text{Var}_{P^{\dagger}_{s_h, a_h}}(r^{\dagger}_h + V^{\dagger \pi^{\star}}_{h+1}) \cdot \iota}{nd^{\mu}_h(s_h, a_h)}} \right] + \widetilde{O}(\frac{H^3}{n \cdot \bar{\bar{d}}_m})$$

$$= C' \sum_{h=1}^{H} \sum_{(s_h, a_h) \in \mathcal{K}_h} d^{\dagger \pi^{\star}}(s_h, a_h) \sqrt{\frac{\text{Var}_{P^{\dagger}_{s_h, a_h}}(r^{\dagger}_h + V^{\dagger \pi^{\star}}_{h+1}) \cdot \iota}{nd^{\mu}_h(s_h, a_h)}} + \widetilde{O}(\frac{H^3}{n \cdot \bar{\bar{d}}_m})$$

here $\widetilde{O}$ absorbs log factor and even higher orders.

Note throughout the section we assume $\widetilde{M}^{\dagger} = M^{\dagger}$. Now be Lemma E.1, we can replace the $\mathcal{K}_h$ in above by $\mathcal{C}_h$ so the result holds in high probability.

Lastly, we end up with w.p. $1 - \delta$

$$0 \leq v^{\pi^{\star}} - v^{\widehat{\pi}} \leq \sum_{h=2}^{H+1} d^{\dagger \pi^{\star}}_h(s^{\dagger}_h) + v^{\dagger \pi^{\star}} - v^{\widehat{\pi}} \leq \sum_{h=2}^{H+1} d^{\dagger \pi^{\star}}_h(s^{\dagger}_h) + v^{\dagger \pi^{\star}} - v^{\dagger \widehat{\pi}}$$

$$\leq \sum_{h=2}^{H+1} d^{\dagger \pi^{\star}}_h(s^{\dagger}_h) + C' \sum_{h=1}^{H} \sum_{(s_h, a_h) \in \mathcal{C}_h} d^{\dagger \pi^{\star}}_h(s_h, a_h) \sqrt{\frac{\text{Var}_{P^{\dagger}_{s_h, a_h}}(r^{\dagger}_h + V^{\dagger \pi^{\star}}_{h+1}) \cdot \iota}{nd^{\mu}_h(s_h, a_h)}} + \widetilde{O}(\frac{H^3}{n \cdot \bar{\bar{d}}_m})$$

(27)

where the first inequality uses Lemma E.2 with $\pi = \pi^{\star}$ and the second one uses Lemma E.2 with $\pi = \widehat{\pi}$. This concludes the proof of Theorem 5.1. The rest of the results are coming from Lemma E.3, E.4.

**Remark E.10.** *We mention the summation of the main term in (27) does not include $s^{\dagger}_h$ since $V^{\dagger \pi}_h(s^{\dagger}_h) = 0$ for any $\pi$ due to the pessimistic MDP design. In particular, this state contributes nothing to neither $v^{\dagger \pi^{\star}}$ nor $v^{\dagger \widehat{\pi}}$.*

## E.2 Interpretation of Theorem 5.1

The constant (in $n$) gap, which is incurred by the behavior agnostic space $\bigcup_{h=1}^{H}\{(s_h, a_h) : d^{\mu}_h(s_h, a_h) = 0\}$, is bounded by

$$\sum_{h=2}^{H+1} d^{\dagger \pi^{\star}}_h(s^{\dagger}_h) = \sum_{h=2}^{H+1} \sum_{t=1}^{h-1} \sum_{(s_t, a_t) \in \mathcal{S} \times \mathcal{A} \backslash \mathcal{C}_t} d^{\dagger \pi^{\star}}_t(s_t, a_t) \leq \sum_{h=2}^{H+1} \sum_{t=1}^{h-1} \sum_{(s_t, a_t) \in \mathcal{S} \times \mathcal{A} \backslash \mathcal{C}_t} d^{\pi^{\star}}_t(s_t, a_t),$$

Note for quantity $d^{\dagger \pi^{\star}}_t(s_t, a_t)$ (where $(s_t, a_t) \in \mathcal{S} \times \mathcal{A} \backslash \mathcal{C}_t$), it is equivalently defined as

$$d^{\dagger \pi^{\star}}_t(s_t, a_t) = \mathbb{P}_{M^{\dagger}}[S_t, A_t = s_t, a_t | (S_{t-1}, A_{t-1}) \in \mathcal{C}_{t-1}, \ldots, (S_1, A_1) \in \mathcal{C}_1]$$

is probability for the first time the trajectory exits the reachable regions and enters $(s_t, a_t) \notin \mathcal{C}_t$. Therefore, $d_t^{\dagger \pi^\star}(s_t, a_t)$ is much smaller than $d_t^{\pi^\star}(s_t, a_t)$ for $s_t, a_t \notin \mathcal{C}_h$ (since $d_t^{\pi^\star}(s_t, a_t)$ includes the probability that trajectory $s_t, a_t$). Such a feature is reflected by the quantity that express the gap using the mass of the absorbing state: $\sum_{h=2}^{H+1} d_h^{\dagger \pi^\star}(s_h^\dagger)(= \sum_{h=2}^{H+1} \sum_{t=1}^{h-1} \sum_{(s_t, a_t) \in \mathcal{S} \times \mathcal{A} \backslash \mathcal{C}_t} d_t^{\dagger \pi^\star}(s_t, a_t))$. Especially, this gap can vary between 0 and $H$, depending on the exploratory ability of $\mu$. Also, different from AVPI, the *assumption-free* AVPI set 0 penalty at locations where $n_{s_t, a_t} = 0$. The interpretation is: the locations with $n_{s_t, a_t} = 0$ in $M^\dagger$ are the fully aware locations (with deterministic transition to $s^\dagger$ and reward 0 by design) therefore we are certain about the behaviors in those places.

## F  Proof of Theorem 4.1

Indeed, Theorem 4.1 can be implied by Theorem 5.1 as a special case.

*Proof of Theorem 4.1.* Under Assumption 2.3, $d_h^{\pi^\star}(s_h, a_h) = 0$ if $d_h^\mu(s_h, a_h) = 0$. In this case,

$$0 \le \sum_{h=2}^{H+1} d_h^{\dagger \pi^\star}(s_h^\dagger) = \sum_{h=2}^{H+1} \sum_{t=1}^{h-1} \sum_{(s_t, a_t) \in \mathcal{S} \times \mathcal{A} \backslash \mathcal{C}_t} d_t^{\dagger \pi^\star}(s_t, a_t) \le \sum_{h=2}^{H+1} \sum_{t=1}^{h-1} \sum_{(s_t, a_t) \in \mathcal{S} \times \mathcal{A} \backslash \mathcal{C}_t} d_t^{\pi^\star}(s_t, a_t)$$

$$= \sum_{h=2}^{H+1} \sum_{t=1}^{h-1} \sum_{(s_t, a_t): d_t^\mu(s_t, a_t) = 0} d_t^{\pi^\star}(s_t, a_t) = 0$$

due to Lemma E.3,E.4. Therefore, the gap $\sum_{h=1}^{H} d_h^{\dagger \pi^\star}(s_h^\dagger)$ vanishes when Assumption 2.3 is true. Also, in this case $M^\dagger$ can be replaced by a $M'$, where $M'$ is the sub-MDP induced by $\mu$. *i.e.*, $M' = \bigcup_{h=1}^{H} \mathcal{S}_h \times \mathcal{A}_h$ with $\mathcal{S}_h \times \mathcal{A}_h = \mathcal{C}_h$.[11] The transitions and the rewards remain the same in $M^\dagger$.

Since there is certain $\pi^\star$ that is fully covered by $\mu$, for such $\pi^\star$ we have $V_h^{\pi^\star}|_M = V_h^{\pi^\star}|_{M'}$ for all $h \in [H]$. Also, in $M'$, $\mu$ can explore all the locations, therefore the probability transition to $s_h^\dagger$ is 0. Hence, all the $d^\dagger, P^\dagger, r^\dagger, V^\dagger$ in Theorem 4.1 are replaced by its original version.

∎

**Remark F.1.** *Note even though the proof can essentially leverage the reduction of the proving procedure of Theorem 5.1, for clear presentation of the algorithm design we still include the locations with no observation and set the severe penalty $\tilde{O}(H)$. This is different from its assumption-free version with 0 penalty (also see Section E.2 for related discussions).*

## G  Proof of Theorem 4.3: Instance-dependent Lower Bound

Global minimax lower bound holds uniformly over large classes of models but lacks the characterization of individual instances. The more appropriate characterization of instance dependence is the (non-asymptotic) local minimax bound, which is originated from the local minimax framework Cai and Low [2004] and recently used in Khamaru et al. [2021a,b]. The proof essentially relies on the reduction to the testing between two value instances with respect to the Hellinger distance. Specifically, the choice of the alternative instance should characterize the MDP problem we are considering and we fix the MDP problem (together with the behavior policy $\mu$) as: $\mathcal{P} := (\mu, M)$ where $M = (\mathcal{S}, \mathcal{A}, P, r, H, d_1)$. Recall the local risk is defined as:

$$\mathfrak{R}_n(\mathcal{P}) := \sup_{\mathcal{P}' \in \mathcal{G}} \inf_{\widehat{\pi}} \max_{\mathcal{Q} \in \{\mathcal{P}, \mathcal{P}'\}} \sqrt{n} \cdot \mathbb{E}_{\mathcal{Q}}\left[v^\star(\mathcal{Q}) - v^{\widehat{\pi}}\right]$$

and $\mathcal{G} := \{(\mu, M) : \exists \pi^\star \text{ s.t. } d_h^\mu(s, a) > 0 \text{ if } d_h^{\pi^\star}(s, a) > 0\}$.

For the ease of exposition, we use the notation $\mathcal{P} = (\mu, P_{1:H}, r)$ instead of $(\mu, M)$. We start by considering the following two classes of alternatives instances:

$$\mathcal{S}_1 = \{\mathcal{P}' = (\mu', P'_{1:H}, r') \mid \mu' = \mu, r' = r, \mathcal{P}' \in \mathcal{G}\}, \quad \mathcal{S}_2 = \{\mathcal{P}' = (\mu', P'_{1:H}, r') \mid \mu' = \mu, P'_{1:H} = P_{1:H}, \mathcal{P}' \in \mathcal{G}\}.$$
(28)

---

[11]In this sub-MDP, each state might have different number of actions!

and define the restricted local risks w.r.t. $\mathcal{S}_i$:

$$\mathfrak{R}_n(\mathcal{P}, \mathcal{S}_i) := \sup_{\mathcal{P}' \in \mathcal{S}_i} \inf_{\hat{\pi}} \max_{\mathcal{Q} \in \{\mathcal{P}, \mathcal{P}'\}} \sqrt{n} \cdot \mathbb{E}_{\mathcal{Q}} \left[ v^\star(\mathcal{Q}) - v^{\hat{\pi}} \right], \quad i = 1, 2. \tag{29}$$

Then it suffices to prove the following lemma:

**Lemma G.1.** *There exists an universal constant $C > 0$ such that:*

$$\mathfrak{R}_n(\mathcal{P}, \mathcal{S}_1) \geq C \cdot \sum_{h=1}^{H} \sum_{(s_h, a_h) \in \mathcal{C}_h} d_h^{\pi^\star}(s_h, a_h) \cdot \sqrt{\frac{\mathrm{Var}_{P_{s_h, a_h}}(V_{h+1}^\star)}{\zeta \cdot d_h^\mu(s_h, a_h)}},$$

$$\mathfrak{R}_n(\mathcal{P}, \mathcal{S}_2) \geq C \cdot \sum_{h=1}^{H} \sum_{(s_h, a_h) \in \mathcal{C}_h} d_h^{\pi^\star}(s_h, a_h) \cdot \sqrt{\frac{\mathrm{Var}_{s_h, a_h}(r_h)}{\zeta \cdot d_h^\mu(s_h, a_h)}}.$$

Given Lemma G.1, we can directly prove Theorem 4.3 as follows.

*Proof of Theorem 4.3.* Given Lemma G.1, we directly have

$$
\begin{aligned}
\mathfrak{R}_n(\mathcal{P}) &\geq \max\{\mathfrak{R}_n(\mathcal{P}, \mathcal{S}_1), \mathfrak{R}_n(\mathcal{P}, \mathcal{S}_2)\} \\
&\geq \frac{1}{2} \left( \mathfrak{R}_n(\mathcal{P}, \mathcal{S}_1) + \mathfrak{R}_n(\mathcal{P}, \mathcal{S}_2) \right) \\
&\geq \frac{C}{2} \cdot \sum_{h=1}^{H} \sum_{(s_h, a_h) \in \mathcal{C}_h} d_h^{\pi^\star}(s_h, a_h) \cdot \left( \sqrt{\frac{\mathrm{Var}_{P_{s_h, a_h}}(V_{h+1}^\star)}{\zeta \cdot d_h^\mu(s_h, a_h)}} + \sqrt{\frac{\mathrm{Var}_{s_h, a_h}(r_h)}{\zeta \cdot d_h^\mu(s_h, a_h)}} \right) \\
&\geq \frac{C}{2} \cdot \sum_{h=1}^{H} \sum_{(s_h, a_h) \in \mathcal{C}_h} d_h^{\pi^\star}(s_h, a_h) \cdot \sqrt{\frac{\mathrm{Var}_{P_{s_h, a_h}}(r_h + V_{h+1}^\star)}{\zeta \cdot d_h^\mu(s_h, a_h)}}
\end{aligned}
$$

where the last inequality uses $\sqrt{a} + \sqrt{b} \geq \sqrt{a+b}$ for all $a, b \geq 0$ and $\mathrm{Var}_{P_{s_h, a_h}}(V_{h+1}^\star) + \mathrm{Var}_{s_h, a_h}(r_h) = \mathrm{Var}_{P_{s_h, a_h}}(r_h + V_{h+1}^\star)$ since $V_{h+1}^\star$ and $r_h$ are conditionally independent given $s_h, a_h$. $\blacksquare$

For the rest of the section, we prove Lemma G.1.

## G.1 Reduction to two-point optimal-value estimations

We first need the following lemma, which converts local learning risk to the following $\mathfrak{M}_n$ via the reduction from estimation to testing.

**Lemma G.2.** *Define*

$$\mathfrak{M}_n(\mathcal{P}, \mathcal{S}_1) := \sup_{\mathcal{P}' \in \mathcal{S}_1} \left\{ \sqrt{n} \cdot |v^\star(\mathcal{P}) - v^\star(\mathcal{P}')| \, \big| \, d_{Hel}(P^n, P'^n)^2 \leq 0.4 \right\}$$

$$\mathfrak{M}_n(\mathcal{P}, \mathcal{S}_2) := \sup_{\mathcal{P}' \in \mathcal{S}_2} \left\{ \sqrt{n} \cdot |v^\star(\mathcal{P}) - v^\star(\mathcal{P}')| \, \big| \, d_{Hel}(p_r^n, p_r'^n) \leq 0.4 \right\}.$$

*then we have*

$$\mathfrak{R}_n(\mathcal{P}, \mathcal{S}_i) \geq \frac{1}{50} \mathfrak{M}_n(\mathcal{P}, \mathcal{S}_i), \quad i = 1, 2.$$

*Proof of Lemma G.2.* Indeed, denote $\mathcal{P}^n$ to be a product measure induced by $n$ trajectories from $\mathcal{P}$, then for any output $\hat{\pi}$ by the averaged risk we have:

$$
\begin{aligned}
\max_{\mathcal{Q} \in \{\mathcal{P}, \mathcal{P}'\}} \mathbb{E}_{\mathcal{Q}} \left[ |v^\star(\mathcal{Q}) - v^{\hat{\pi}}| \right] &\geq \frac{1}{2} \left( \mathbb{E}_{\mathcal{P}^n} \left[ |v^\star(\mathcal{P}) - v^{\hat{\pi}}| \right] + \mathbb{E}_{\mathcal{P}'^n} \left[ |v^\star(\mathcal{P}') - v^{\hat{\pi}}| \right] \right) \\
&\geq \frac{1}{2} \delta \left[ \mathcal{P}^n \left( |v^\star(\mathcal{P}) - v^{\hat{\pi}}| \geq \delta \right) + \mathcal{P}'^n \left( |v^\star(\mathcal{P}') - v^{\hat{\pi}}| \geq \delta \right) \right],
\end{aligned}
$$

where the last inequality is by Markov inequality. Now choose $\delta = \frac{1}{2} \cdot |v^\star(\mathcal{P}) - v^\star(\mathcal{P}')|$, we have $|v^\star(\mathcal{P}) - v^{\hat\pi}| \le \delta$ implies $|v^\star(\mathcal{P}') - v^{\hat\pi}| \ge \delta$, therefore above

$$
\begin{aligned}
&= \frac{1}{2}\delta \cdot \left[1 - \mathcal{P}^n \left(|v^\star(\mathcal{P}) - v^{\hat\pi}| < \delta\right) + \mathcal{P}'^n \left(|v^\star(\mathcal{P}') - v^{\hat\pi}| \ge \delta\right)\right] \\
&\ge \frac{1}{2}\delta \cdot \left[1 - \mathcal{P}^n \left(|v^\star(\mathcal{P}') - v^{\hat\pi}| \ge \delta\right) + \mathcal{P}'^n \left(|v^\star(\mathcal{P}') - v^{\hat\pi}| \ge \delta\right)\right] \qquad (30)\\
&\ge \frac{1}{2}\delta \cdot [1 - \|\mathcal{P}^n - \mathcal{P}'^n\|_{\mathrm{TV}}] \ge \frac{1}{2}\delta \cdot \left[1 - \sqrt{2} \cdot d_{Hel}(\mathcal{P}^n, \mathcal{P}'^n)\right]
\end{aligned}
$$

Then plug in the condition for $d_{Hel}(\mathcal{P}^n, \mathcal{P}'^n)$ we obtain the result. The proof for the second result is similar.

∎

## G.2 Instance-dependent lower bound

Now we complete the proof by the following lemma. Combing Lemma G.3 and Lemma G.2, we finish the proof of Lemma G.1.

**Lemma G.3.** *There exists an universal constant $C > 0$ such that:*

$$
\mathfrak{M}_n(\mathcal{P}, \mathcal{S}_1) \ge C \cdot \sum_{h=1}^{H} \sum_{(s_h, a_h) \in \mathcal{C}_h} d_h^{\pi^\star}(s_h, a_h) \cdot \sqrt{\frac{\mathrm{Var}_{P_{s_h, a_h}}(V_{h+1}^\star)}{\zeta \cdot d_h^\mu(s_h, a_h)}},
$$

$$
\mathfrak{M}_n(\mathcal{P}, \mathcal{S}_2) \ge C \cdot \sum_{h=1}^{H} \sum_{(s_h, a_h) \in \mathcal{C}_h} d_h^{\pi^\star}(s_h, a_h) \cdot \sqrt{\frac{\mathrm{Var}_{s_h, a_h}(r_h)}{\zeta \cdot d_h^\mu(s_h, a_h)}}.
$$

*Proof of Lemma G.3.* So far we haven't leveraged the specific structure of instance $\mathcal{P} = (\mu, P_{1:H}, r)$. Now we define $P'$ as follows $(\forall h, s_h, a_h)$:[12]

$$
P_h'(s_{h+1}|s_h, a_h) = P_h(s_{h+1}|s_h, a_h) + \frac{P_h(s_{h+1}|s_h, a_h)\left(V_{h+1}^\star(s_{h+1}) - \mathbb{E}_{P_{s_h, a_h}}[V_{h+1}^\star]\right)}{8\sqrt{\zeta \cdot n_{s_h, a_h} \cdot \mathrm{Var}_{P_{s_h, a_h}}(V_{h+1}^\star)}} \qquad (31)
$$

where $\zeta = H/\bar{d}_m$ and the alternative instance as $\mathcal{P}' = (\mu, P_{1:H}', r)$. Denote $\bar{Q}^\star$ to be the optimal $Q$-values under $\mathcal{P}'$ and $Q^\star$ the optimal $Q$-values under $\mathcal{P}$. $\bar{\pi}^\star$ is the optimal policy under $\mathcal{P}'$ and $\pi^\star$ is the optimal policy under $\mathcal{P}$. The proof has two steps.

**Step1:** we show

$$
2 \cdot d_{Hel}(\mathcal{P}^n, \mathcal{P}'^n)^2 \le 0.8
$$

Define $\tau = (s_1, a_1, s_2, a_2, \ldots, s_H, a_H) \sim P(s_1, a_1, s_2, a_2, \ldots, s_H, a_H)$ to be the trajectories, then

$$
\begin{aligned}
d_{Hel}(\mathcal{P}^n, \mathcal{P}'^n)^2 &= 1 - \int_{\tau^n} \sqrt{f_{P'^n}(\tau^n) \cdot f_{P^n}(\tau^n)} d\tau^n = 1 - \prod_{i=1}^{n} \int_\tau \sqrt{f_{P'}(\tau) \cdot f_P(\tau)} d\tau \\
&= 1 - \prod_{i=1}^{n} \int_{s_1} \sqrt{d_1^{P'}(s_1) d_1^P(s_1)} \left(\int_{a_1} \sqrt{\mu_{P'}(a_1|s_1) \mu_P(a_1|s_1)} \left(\int_{s_2} \sqrt{P_1'(s_2|s_1, a_1) P_1(s_2|s_1, a_1)} \ldots ds_2\right) da_1\right) ds_1 \\
&\le 1 - \prod_{i=1}^{n} \prod_{h=1}^{H} \min_{s,a} \left(\int_{s'} \sqrt{P_1'(s'|s, a) P_1(s'|s, a)} \ldots ds'\right) = 1 - \prod_{i=1}^{n} \prod_{h=1}^{H} (1 - \max_{s,a} d_{Hel}(\mathcal{P}_{h,s,a}, \mathcal{P}_{h,s,a}')^2) \\
&\le 1 - \prod_{i=1}^{n} \prod_{h=1}^{H} \left(1 - \frac{1}{2nH}\right) = 1 - \left(1 - \frac{1}{2nH}\right)^{nH} \le 1 - \frac{1}{\sqrt{e}} \le 0.4,
\end{aligned}
$$

(32)

---

[12]In below, it suffices to only consider the instance where $n_{s_h, a_h} \cdot \mathrm{Var}_{P_{s_h, a_h}}(V_{h+1}^\star) > 0$ since, 1. when $\mathrm{Var}_{P_{s_h, a_h}}(V_{h+1}^\star) = 0$, the numerator is also 0 therefore by convention by we can define ratio to be 0; 2. if $n_{s_h, a_h} = 0$, then with high probability $d_h^\mu(s_h, a_h) = 0$, in this case the transition $P(\cdot|s_h, a_h)$ does not matter since $d_h^{\pi^\star}(s_h, a_h) = 0$ by theorem condition.

where the second inequality uses independence of trajectories and the third equation comes from the conditional probability rule. The first inequality comes from $\int_a \sqrt{\mu_{P'}(a|s)\mu_P(a|s)}da = \int_a \mu_P(a|s)da = 1$ and the second inequality comes from item 2 of Lemma J.12 via Definition J.11. This verifies $P'$ satisfies the condition of $\mathfrak{M}_n(\mathcal{P}, \mathcal{S}_1)$.

**Step2:** we show for this instance $\mathcal{P}'$ we have

$$\sqrt{n}|v^\star(\mathcal{P}) - v^\star(\mathcal{P}')| \geq C \cdot \sum_{h=1}^{H} \sum_{(s_h,a_h) \in \mathcal{C}_h} d_h^{\pi^\star}(s_h, a_h) \cdot \sqrt{\frac{\mathrm{Var}_{P_{s_h,a_h}}(V_{h+1}^\star)}{\zeta \cdot d_h^\mu(s_h, a_h)}}.$$

Define $\xi = \sup_{h,s_h,a_h,s_{h+1},d_h^\mu(s_h,a_h)\cdot\mathrm{Var}_{P_{s_h,a_h}}(V_{h+1}^\star)>0} \frac{P_h(s_{h+1}|s_h,a_h)\left(V_{h+1}^\star(s_{h+1})-\mathbb{E}_{P_{s_h,a_h}}[V_{h+1}^\star]\right)}{4\sqrt{\zeta\cdot d_h^\mu(s_h,a_h)\cdot\mathrm{Var}_{P_{s_h,a_h}}(V_{h+1}^\star)}},$

$$Q_1^\star - \bar{Q}_1^\star = \left(r_1 + P_1^{\pi^\star} Q_2^\star\right) - \left(r_1 + P_1'^{\bar{\pi}^\star} \bar{Q}_2^\star\right)$$

$$\leq \left(r_1 + P_1^{\pi^\star} Q_2^\star\right) - \left(r_1 + P_1'^{\pi^\star} \bar{Q}_2^\star\right)$$

$$= P_1^{\pi^\star} \left(Q_2^\star - \bar{Q}_2^\star\right) + \left(P_1^{\pi^\star} - P_1'^{\pi^\star}\right) \bar{Q}_2^\star$$

$$\leq P_1^{\pi^\star} P_2^{\pi^\star} \left(Q_3^\star - \bar{Q}_3^\star\right) + P_1^{\pi^\star} \left(P_2^{\pi^\star} - P_2'^{\pi^\star}\right) \bar{Q}_3^\star + \left(P_1^{\pi^\star} - P_1'^{\pi^\star}\right) \bar{Q}_2^\star$$

$$\leq \cdots$$

$$\leq \sum_{h=1}^{H} P_{1:h-1}^{\pi^\star} \left(P_h^{\pi^\star} - P_h'^{\pi^\star}\right) \bar{Q}_{h+1}^\star = \sum_{h=1}^{H} P_{1:h-1}^{\pi^\star} \left(P_h - P_h'\right) \bar{Q}_{h+1}^\star(\cdot, \pi^\star(\cdot))$$

$$\leq H \cdot \sup_{h,s_h,a_h,s_{h+1}} |P_h - P_h'|(s_{h+1}|s_h, a_h) \cdot \left\|\bar{Q}_{h+1}^\star\right\|_\infty$$

$$\leq H^2 \cdot \sup_{h,s_h,a_h,s_{h+1}} \frac{P_h(s_{h+1}|s_h, a_h)\left(V_{h+1}^\star(s_{h+1}) - \mathbb{E}_{P_{s_h,a_h}}[V_{h+1}^\star]\right)}{8\sqrt{\zeta \cdot n_{s_h,a_h} \cdot \mathrm{Var}_{P_{s_h,a_h}}(V_{h+1}^\star)}}$$

$$\leq H^2 \cdot \sup_{h,s_h,a_h,s_{h+1}} \frac{P_h(s_{h+1}|s_h, a_h)\left(V_{h+1}^\star(s_{h+1}) - \mathbb{E}_{P_{s_h,a_h}}[V_{h+1}^\star]\right)}{8\sqrt{\zeta \cdot n d_h^\mu(s_h,a_h) \cdot \mathrm{Var}_{P_{s_h,a_h}}(V_{h+1}^\star)}} = H^2 \xi \sqrt{\frac{1}{n}}$$

where the first inequality is by $\bar{\pi}^\star$ is the optimal policy for $\bar{Q}^\star$ and second inequality is by recursively applying the **element-wisely** inequality for $Q_2^\star - \bar{Q}_2^\star$ and the fact that $P_1, P_2$ has non-negative coordinates. The last inequality is by Lemma J.1. By a similar calculation, we also have

$$\bar{Q}_1^\star - Q_1^\star = \left(r_1 + P_1'^{\bar{\pi}^\star} \bar{Q}_2^\star\right) - \left(r_1 + P_1^{\pi^\star} Q_2^\star\right)$$

$$\leq \cdots$$

$$\leq \sum_{h=1}^{H} P_{1:h-1}'^{\bar{\pi}^\star} \left(P_h' - P_h\right) Q_{h+1}^\star(\cdot, \bar{\pi}^\star(\cdot)) \leq H^2 \xi \sqrt{\frac{1}{n}}$$

and combing the above two we obtain $\left\|\bar{Q}_1^\star - Q_1^\star\right\|_\infty \leq H^2 \xi \sqrt{\frac{1}{n}}$, and (by similar computation and the union bound by Lemma J.1) further holds true for all $\bar{Q}_h^\star, Q_h^\star$'s, with high probability

$$\max_h \left\|\bar{Q}_1^\star - Q_1^\star\right\|_\infty \leq H^2 \xi \sqrt{\frac{1}{n}} \tag{33}$$

Now by the calculation again,

$$
\begin{aligned}
\bar{Q}_1^\star - Q_1^\star &= r_1 + P_1'^{\bar{\pi}^\star} \bar{Q}_2^\star - \left(r_1 + P_1^{\pi^\star} Q_2^\star\right) \\
&\geq r_1 + P_1'^{\pi^\star} \bar{Q}_2^\star - \left(r_1 + P_1^{\pi^\star} Q_2^\star\right) \\
&= P_1'^{\pi^\star} \left(\bar{Q}_2^\star - Q_2^\star\right) + \left(P_1'^{\pi^\star} - P_1^{\pi^\star}\right) Q_2^\star \\
&= \left(P_1'^{\pi^\star} - P_1^{\pi^\star}\right)\left(\bar{Q}_2^\star - Q_2^\star\right) + P_1^{\pi^\star}\left(\bar{Q}_2^\star - Q_2^\star\right) + \left(P_1'^{\pi^\star} - P_1^{\pi^\star}\right) Q_2^\star \\
&\geq \left(P_1'^{\pi^\star} - P_1^{\pi^\star}\right)\left(\bar{Q}_2^\star - Q_2^\star\right) + P_1^{\pi^\star}\left(P_2'^{\pi^\star} - P_2^{\pi^\star}\right)\left(\bar{Q}_3^\star - Q_3^\star\right) \\
&\quad + \left(P_1'^{\pi^\star} - P_1^{\pi^\star}\right) Q_2^\star + P_1^{\pi^\star}\left(P_2'^{\pi^\star} - P_2^{\pi^\star}\right) Q_3^\star \\
&\quad + P_1^{\pi^\star} P_2^{\pi^\star}\left(\bar{Q}_3^\star - Q_3^\star\right) \\
&\geq \ldots \\
&\geq \sum_{h=1}^{H} P_{1:h-1}^{\pi^\star}\left(P_h'^{\pi^\star} - P_h^{\pi^\star}\right)\left(\bar{Q}_{h+1}^\star - Q_{h+1}^\star\right) + \sum_{h=1}^{H} P_{1:h-1}^{\pi^\star}\left(P_h'^{\pi^\star} - P_h^{\pi^\star}\right) Q_{h+1}^\star \\
&= \sum_{h=1}^{H} P_{1:h-1}^{\pi^\star}\left(P_h'^{\pi^\star} - P_h^{\pi^\star}\right)\left(\bar{Q}_{h+1}^\star - Q_{h+1}^\star\right) + \sum_{h=1}^{H} P_{1:h-1}^{\pi^\star}\left(P_h' - P_h\right) V_{h+1}^\star
\end{aligned}
$$

(34)

where the second inequality recursively applies $\bar{Q}_2^\star - Q_2^\star \geq P_2'^{\pi^\star}\left(\bar{Q}_3^\star - Q_3^\star\right) + \left(P_2'^{\pi^\star} - P_2^{\pi^\star}\right) Q_3^\star$ and the above is equivalent to

$$
\sum_{h=1}^{H} P_{1:h-1}^{\pi^\star}\left(P_h^{\pi^\star} - P_h'^{\pi^\star}\right)\left(\bar{Q}_{h+1}^\star - Q_{h+1}^\star\right) + \bar{Q}_1^\star - Q_1^\star \geq \sum_{h=1}^{H} P_{1:h-1}^{\pi^\star}\left(P_h' - P_h\right) V_{h+1}^\star.
$$

Now by Lemma J.12 item 3, $\sum_{h=1}^{H} P_{1:h-1}^{\pi^\star}\left(P_h' - P_h\right) V_{h+1}^\star \geq 0$, therefore multiply the initial distribution on both sides and take the absolute value on the left hand side to get

$$
\sum_{h=1}^{H} d_{1:h-1}^{\pi^\star}\left|P_h^{\pi^\star} - P_h'^{\pi^\star}\right|\left|\bar{Q}_{h+1}^\star - Q_{h+1}^\star\right| + |\bar{v}^\star - v^\star| \geq \sum_{h=1}^{H} d_{1:h-1}^{\pi^\star}\left(P_h' - P_h\right) V_{h+1}^\star
$$

(35)

On one hand,

$$
\sum_{h=1}^{H} d_{1:h-1}^{\pi^\star}\left|P_h^{\pi^\star} - P_h'^{\pi^\star}\right|\left|\bar{Q}_{h+1}^\star - Q_{h+1}^\star\right| \leq H \cdot \xi \sqrt{\frac{1}{n}} \sup_h \left\|\bar{Q}_h^\star - Q_h^\star\right\|_\infty \leq H^3 \xi^2 \frac{1}{n},
$$

One the other hand,

$$
\begin{aligned}
\sum_{h=1}^{H} d_{1:h-1}^{\pi^\star}\left(P_h' - P_h\right) V_{h+1}^\star &= \sum_{h=1}^{H} \sum_{s_h,a_h} d_h^{\pi^\star}(s_h, a_h) \sum_{s_{h+1}}\left(P_h' - P_h\right)(s_{h+1}|s_h, a_h) V_{h+1}^\star(s_{h+1}) \\
&= \sum_{h=1}^{H} \sum_{s_h,a_h} d_h^{\pi^\star}(s_h, a_h) \sum_{s_{h+1}}\left(P_h' - P_h\right)(s_{h+1}|s_h, a_h)\left(V_{h+1}^\star(s_{h+1}) - \mathbb{E}_{P_{s_h,a_h}}[V_{h+1}^\star]\right) \\
&= \sum_{h=1}^{H} \sum_{s_h,a_h} d_h^{\pi^\star}(s_h, a_h) \sum_{s_{h+1}} \frac{P_h(s_{h+1}|s_h, a_h)\left(V_{h+1}^\star(s_{h+1}) - \mathbb{E}_{P_{s_h,a_h}}[V_{h+1}^\star]\right)^2}{8\sqrt{\zeta \cdot n_{s_h,a_h} \cdot \mathrm{Var}_{P_{s_h,a_h}}(V_{h+1}^\star)}} \\
&= \sum_{h=1}^{H} \sum_{s_h,a_h} d_h^{\pi^\star}(s_h, a_h) \sqrt{\frac{\mathrm{Var}_{P_{s_h,a_h}}(V_{h+1}^\star)}{8^2 \zeta \cdot n_{s_h,a_h}}} \geq \sum_{h=1}^{H} \sum_{s_h,a_h} d_h^{\pi^\star}(s_h, a_h) \sqrt{\frac{\mathrm{Var}_{P_{s_h,a_h}}(V_{h+1}^\star)}{96\zeta \cdot n d_h^\mu(s_h, a_h)}}
\end{aligned}
$$

The last step is by Lemma J.1. Combing those two into (35), we finally obtain

$$|\bar{v}^\star - v^\star| \geq \sum_{h=1}^{H} \sum_{s_h,a_h} d_h^{\pi^\star}(s_h,a_h) \sqrt{\frac{\mathrm{Var}_{P_{s_h,a_h}}(V_{h+1}^\star)}{96\zeta \cdot n d_h^\mu(s_h,a_h)}} - H^3\xi^2 \frac{1}{n}$$

$$\geq \frac{1}{2} \sum_{h=1}^{H} \sum_{s_h,a_h} d_h^{\pi^\star}(s_h,a_h) \sqrt{\frac{\mathrm{Var}_{P_{s_h,a_h}}(V_{h+1}^\star)}{96\zeta \cdot n d_h^\mu(s_h,a_h)}} \tag{36}$$

as long as $n \geq \dfrac{4H^6\xi^4}{\left(\sum_{h=1}^{H} \sum_{s_h,a_h} d_h^{\pi^\star}(s_h,a_h) \sqrt{\frac{\mathrm{Var}_{P_{s_h,a_h}}(V_{h+1}^\star)}{96\zeta d_h^\mu(s_h,a_h)}}\right)^2}.$

By (36), we finish the proof of Step2.

**The result for the reward can be similarly derived in the following sense.** First, define the perturbed mean reward as:

$$r_h'(s_h,a_h) = r_h(s_h,a_h) + \frac{\sigma_r}{2\sqrt{\zeta \cdot n_{s_h,a_h}}}, \tag{37}$$

where $\sigma_r > 0$ is a parameter and the realization of reward is sampled from normal $r|_{s_h,a_h} \sim \mathcal{N}(r_h(s_h,a_h), \sigma_r^2)$ and $r'|_{s_h,a_h} \sim \mathcal{N}(r_h'(s_h,a_h), \sigma_r^2)$. In this scenario, similar to (32), we have

$$d_{Hel}(\mathcal{P}^n, \mathcal{P}'^n)^2 = 1 - \int_{\tau^n} \sqrt{f_{P'^n}(\tau^n) \cdot f_{P^n}(\tau^n)} d\tau^n$$

$$\leq 1 - \prod_{i=1}^{n} \prod_{h=1}^{H} (1 - \max_{s,a} d_{Hel}(r_{h,s,a}, r_{h,s,a}')^2)$$

$$\leq 1 - \prod_{i=1}^{n} \prod_{h=1}^{H} (1 - \max_{s,a} D_{KL}(r_{h,s,a}, r_{h,s,a}')^2) \tag{38}$$

$$= 1 - \prod_{i=1}^{n} \prod_{h=1}^{H} \left(1 - \max_{s,a} \frac{|r_h'(s,a) - r_h(s,a)|^2}{2\sigma_r^2}\right) = 1 - \prod_{i=1}^{n} \prod_{h=1}^{H} \left(1 - \max_{s_h,a_h} \frac{1}{8\zeta \cdot n_{s_h,a_h}}\right)$$

$$\leq 1 - \prod_{i=1}^{n} \prod_{h=1}^{H} \left(1 - \frac{1}{2nH}\right) = 1 - \left(1 - \frac{1}{2nH}\right)^{nH} \leq 1 - \frac{1}{\sqrt{e}} \leq 0.4,$$

and similar to (34)

$$\bar{Q}_1^\star - Q_1^\star = r_1' + P_1^{\bar{\pi}^\star} \bar{Q}_2^\star - \left(r_1 + P_1^{\pi^\star} Q_2^\star\right)$$

$$\geq r_1' + P_1^{\pi^\star} \bar{Q}_2^\star - \left(r_1 + P_1^{\pi^\star} Q_2^\star\right)$$

$$= P_1^{\pi^\star}\left(\bar{Q}_2^\star - Q_2^\star\right) + (r_1' - r_1) \tag{39}$$

$$\geq \dots$$

$$\geq \sum_{h=1}^{H} P_{1:h-1}^{\pi^\star}\left(r_h' - r_h\right) = \sum_{h=1}^{H} P_{1:h-1}^{\pi^\star} \sqrt{\frac{\sigma_r^2}{2\zeta \cdot n_{s_h,a_h}}}$$

the last step is to average over $d_1$ and use Lemma J.1. ∎

# H   Minimax lower bound

**Theorem H.1** (Adaptive minimax lower bound). *Recall for each individual instance $(\mu, M)$, $\bar{d}_m := \min_{h \in [H]}\{d_h^\mu(s_h,a_h) : d_h^\mu(s_h,a_h) > 0\}$ and $\mathcal{D}$ consists of $n$ episodes. Now consider a class of problem family $\mathcal{G} := \{(\mu, M) : \exists \pi^\star \text{ s.t. } d_h^\mu(s,a) > 0 \text{ if } d_h^{\pi^\star}(s,a) > 0\}$. Let $\hat{\pi}$ be the output of any algorithm on $\mathcal{D}$. Then there exists universal constants $c_0, p, C > 0$, such that if*

$n \geq c_0 \cdot 1/\bar{d}_m \cdot \log(HSA/p)$, *with constant probability $p > 0$,*

$$\inf_{\widehat{\pi}} \sup_{(\mu,M)\in\mathcal{G}} \frac{\mathbb{E}_{\mu,M}\left[v^\star - v^{\widehat{\pi}}\right]}{\sum_{h=1}^H \sum_{(s_h,a_h)\in\mathcal{C}_h} d_h^{\pi^\star}(s_h,a_h) \cdot \sqrt{\frac{\mathrm{Var}_{P_{s_h,a_h}}(r_h+V_{h+1}^\star)}{n\cdot d_h^\mu(s_h,a_h)}}} \geq C. \tag{40}$$

For completeness, we provide the proof of minimax lower bound. The proof uses the hard instance construction in Jin et al. [2020].

## H.1 Proof of Theorem H.1

**Construction of the hard MDP instances.** We define a family of MDPs where each MDP instance within in the family has three states $s_1, s_+, s_-$ and $A$ actions $a_1, \ldots, a_A$. The initial state is always $s_1$. The transition kernel at time 1 will transition $s_1$ to either $s_+$ or $s_-$ depending on three probabilities $p_1, p_2, p_3(= \min\{p_1, p_2\})$ and the transition kernel at time $h \geq 2$ will deterministically transition back to itself, *i.e.*

$$P_1(s_+|s_1, a_1) = p_1, \quad P_1(s_1|s_1, a_1) = 1 - p_1,$$
$$P_1(s_+|s_1, a_2) = p_2, \quad P_1(s_1|s_1, a_2) = 1 - p_2,$$
$$P_1(s_+|s_1, a_j) = p_3, \quad P_1(s_1|s_1, a_j) = 1 - p_3, \quad \forall j \geq 3,$$
$$P_h(s_+|s_+, a) = P_h(s_-|s_-, a) = 1, \quad \forall h \geq 2, a \in \mathcal{A}.$$

The state $s_+$ always receives reward 1 regardless of the action and the state $s_1, s_-$ will always have reward 0. We denote such an instance as $M(p_1, p_2, p_3)$.

**Bounding the suboptimality gap.** By construction, the optimal policy at time $h = 1$ will be $\pi_1^\star(s_1) = a_1$ if $p_1 > p_2$ and $\pi_1^\star(s_1) = a_2$ if $p_1 < p_2$. For optimal policy for $h \geq 2$ is arbitrary. Therefore, for those instances (denote $j^* = \arg\max_{j\in\{1,2\}} p_j$)

$$v^\star = 0 + P_1(s_+|s_1, \pi_1^\star(s_1)) \cdot 1 + \sum_{h=3}^H P_1(s_+|s_1, \pi_1^\star(s_1)) \cdot \ldots \cdot P_{h-1}(s_+|s_+, \pi_{h-1}^\star(s_+)) \cdot 1 = p_{j^\star} \cdot (H-1)$$

and

$$v^{\widehat{\pi}} = 0 + \left[\sum_{j=1}^A p_j \cdot \widehat{\pi}_1(a_j|s_1)\right] \cdot 1 + \sum_{h=3}^H \left[\sum_{j=1}^A p_j \cdot \widehat{\pi}_1(a_j|s_1)\right] \cdot \ldots \cdot P_{h-1}(s_+|s_+, \widehat{\pi}_{h-1}(s_+)) \cdot 1$$

$$= \left[\sum_{j=1}^A p_j \cdot \widehat{\pi}_1(a_j|s_1)\right] \cdot (H-1).$$

Therefore the suboptimality gap

$$v^\star - v^{\widehat{\pi}} = p_{j^\star} \cdot (H-1) - \sum_{j=1}^A p_j \cdot \widehat{\pi}_1(a_j|s_1) \cdot (H-1).$$

Let us further denote $p^\star = p_{j^\star}$ and denote the rest of the probabilities as $p$ (*i.e.* if $p_1 > p_2$, then $p_1 = p^\star$, $p_2 = \ldots = p_A = p$; if $p_1 < p_2$, then $p_2 = p^\star$, $p_1 = p_3 = \ldots = p_A = p$), then

$$v^\star - v^{\widehat{\pi}} = p_{j^\star} \cdot (H-1) - \sum_{j=1}^A p_j \cdot \widehat{\pi}_1(a_j|s_1) \cdot (H-1)$$

$$= (H-1) \cdot \left(p^\star - p^\star\widehat{\pi}_1(a_1^\star|s_1) - \sum_{j=2}^A p_j \cdot \widehat{\pi}_1(a_j|s_1)\right)$$

$$= (H-1) \cdot (1 - \widehat{\pi}_1(a_1^\star|s_1))(p^\star - p).$$

Let $\mathcal{D} = \{(s_h^\tau, a_h^\tau, r_h^\tau, s_{h+1}^\tau)\}_{\tau \in [n]}^{h \in [H]}$ is coming from $\mu$ where $\mu$ satisfies $(\mu, M)$ belongs to $\mathcal{G} :=$ $\{(\mu, M) : \exists \pi^\star \text{ s.t. } d_h^\mu(s, a) > 0 \text{ if } d_h^{\pi^\star}(s, a) > 0\}$. Define $n_j = \sum_{\tau=1}^n \mathbf{1}[a_1^\tau = a_j]$. Consider two MDPs $\mathcal{M}_1 = M(p^\star, p, p)$ and $M_2 = M(p, p^\star, p)$, then

$$
\begin{aligned}
&\sup_{l \in \{1,2\}} \sqrt{n_l} \cdot \mathbb{E}_{\mu, \mathcal{M}_l} \left[ v^\star - v^{\widehat{\pi}} \right] \\
&\geq \frac{\sqrt{n_1 n_2}}{\sqrt{n_1} + \sqrt{n_2}} \cdot \left( \mathbb{E}_{\mu, \mathcal{M}_1} \left[ v^\star - v^{\widehat{\pi}} \right] + \mathbb{E}_{\mu, \mathcal{M}_2} \left[ v^\star - v^{\widehat{\pi}} \right] \right) \\
&= \frac{\sqrt{n_1 n_2}}{\sqrt{n_1} + \sqrt{n_2}} \cdot (p^\star - p) \cdot (H - 1) \cdot (\mathbb{E}_{\mu, \mathcal{M}_1}[1 - \widehat{\pi}_1(a_1|s_1)] + \mathbb{E}_{\mu, \mathcal{M}_2}[1 - \widehat{\pi}_1(a_2|s_1)])
\end{aligned}
\tag{41}
$$

where the first inequality uses $\max\{x, y\} \geq a \cdot x + (1-a)y$ for any $a \in [0,1]$, $x, y \geq 0$. Importantly, we choose the above $\mu$ to satisfy $\mu_1(a_1|s_1) > 0$, $\mu_1(a_2|s_1) > 0$. In this scenario, it satisfies $d_h^\mu(s, a) > 0$ if $d_h^{\pi^\star}(s, a) > 0$ (since $\mu$ can reach both $a_1$ and $a_2$, hence $(\mu, M) \in \mathcal{G}$ for $M$ to be either $\mathcal{M}_1$ or $\mathcal{M}_2$ and by the condition $n \geq \widetilde{O}(1/\bar{d}_m)$, $n_1, n_2 > 0$ with high probability (depends on $p$) by Chernoff bound hence the above inequality apply.

Now define the (randomized) test function:

$$
\psi(\widehat{\pi}) = \mathbf{1}\{a \neq a_1\}, \quad \text{where } a \sim \widehat{\pi}_1(\cdot|s_1),
$$

then

$$
\begin{aligned}
&\mathbb{E}_{\mu, \mathcal{M}_1}[1 - \widehat{\pi}_1(a_1|s_1)] + \mathbb{E}_{\mu, \mathcal{M}_2}[1 - \widehat{\pi}_1(a_2|s_1)] = \mathbb{E}_{\mu, \mathcal{M}_1}[\mathbf{1}\{\psi(\widehat{\pi}) = 1\}] + \mathbb{E}_{\mu, \mathcal{M}_2}[\mathbf{1}\{\psi(\widehat{\pi}) = 0\}] \\
&\geq 1 - \text{TV}(\mathbb{P}_{\mathcal{D}\sim(\mu,\mathcal{M}_1)}, \mathbb{P}_{\mathcal{D}\sim(\mu,\mathcal{M}_2)}) \geq 1 - \sqrt{\text{KL}(\mathbb{P}_{\mathcal{D}\sim(\mu,\mathcal{M}_1)}||\mathbb{P}_{\mathcal{D}\sim(\mu,\mathcal{M}_2)})/2}.
\end{aligned}
\tag{42}
$$

Now we apply the following lemma in Jin et al. [2020].

**Lemma H.2** ((C.17) in Jin et al. [2020]). *Let $n_1, n_2 \geq 4$ and $\min\{n_1/n_2, n_2/n_1\} > c$ for some absolute constant c. Then there exists some $p \neq p^\star \in [1/4, 3/4]$ such that $p^\star - p = \dfrac{\sqrt{3}}{4\sqrt{2(n_1+n_2)}}$ and*

$$
\text{KL}(\mathbb{P}_{\mathcal{D}\sim(\mu,\mathcal{M}_1)}||\mathbb{P}_{\mathcal{D}\sim(\mu,\mathcal{M}_2)}) \leq 1/2.
$$

Note the condition of Lemma H.2 is satisfied with high probability by the condition $n \geq \widetilde{O}(1/\bar{d}_m)$ and the design that $\mu_1(a_1|s_1) > 0$, $\mu_1(a_2|s_1) > 0$. Hence, by (41), (42) and Lemma H.2 we have

$$
\sup_{l \in \{1,2\}} \sqrt{n_l} \cdot \mathbb{E}_{\mu, \mathcal{M}_l} \left[ v^\star - v^{\widehat{\pi}} \right] \geq C'(H - 1),
\tag{43}
$$

where $C' = \sqrt{3}/(\sqrt{c} + 1)\sqrt{8c(c + 1)} > 0$.

### H.1.1 The intrinsic quantity in the hard instances

Note under the family $M(p_1, p_2, p_3)$, the optimal values

$$
V_2^\star(s_+) = H - 1, \; V_2^\star(s_-) = 0,
$$

also, since $P_h$ is deterministic for $h \geq 2$, therefore

$$
\text{Var}_{P_h}(r_h + V_{h+1}^\star) = 0, \quad \forall h \geq 2.
$$

For convenience, let us assume the MDP is $M(p^\star, p, p)$. Then in this case

$$\sum_{h=1}^{H} \sum_{(s_h, a_h) \in \mathcal{C}_h} d_h^{\pi^\star}(s_h, a_h) \cdot \sqrt{\frac{\mathrm{Var}_{P_{s_h, a_h}}(r_h + V_{h+1}^\star)}{n \cdot d_h^\mu(s_h, a_h)}}$$

$$= \sum_{(s_1, a_1) \in \mathcal{C}_1} d_1^{\pi^\star}(s_1, a_1) \cdot \sqrt{\frac{\mathrm{Var}_{P_{s_1, a_1}}(r_1 + V_2^\star)}{n \cdot d_1^\mu(s_1, a_1)}}$$

$$= d_1^{\pi^\star}(s_1, a_1) \cdot \sqrt{\frac{\mathrm{Var}_{P_{s_1, a_1}}(r_1 + V_2^\star)}{n \cdot d_1^\mu(s_1, a_1)}} + d_1^{\pi^\star}(s_1, a_2) \cdot \sqrt{\frac{\mathrm{Var}_{P_{s_1, a_2}}(r_1 + V_2^\star)}{n \cdot d_1^\mu(s_1, a_2)}} \tag{44}$$

$$= d_1^{\pi^\star}(s_1, a_1) \cdot \sqrt{\frac{\mathrm{Var}_{P_{s_1, a_1}}(r_1 + V_2^\star)}{n \cdot d_1^\mu(s_1, a_1)}} + 0 \cdot \sqrt{\frac{\mathrm{Var}_{P_{s_1, a_2}}(r_1 + V_2^\star)}{n \cdot d_1^\mu(s_1, a_2)}}$$

$$= 1 \cdot \sqrt{\frac{\mathrm{Var}_{P_{s_1, a_1}}(r_1 + V_2^\star)}{n \cdot d_1^\mu(s_1, a_1)}} \leq \sqrt{\frac{3\mathrm{Var}_{P_{s_1, a_1}}(r_1 + V_2^\star)}{2n_{s_1, a_1}}}$$

$$= \sqrt{\frac{3\mathrm{Var}_{P_{s_1, a_1}}(V_2^\star)}{2n_{s_1, a_1}}} = \sqrt{\frac{3p^\star(1 - p^\star)(H-1)^2}{2n_{s_1, a_1}}} \leq \frac{H-1}{\sqrt{8n_{s_1, a_1}/3}} = \frac{H-1}{\sqrt{8n_1/3}}$$

where the first inequality uses the Chernoff bound.

**Finish the proof.** By (43) and (44), we have with constant probability, for any arbitrary algorithm $\widehat{\pi}$,

$$\sup_{l \in \{1,2\}} \frac{\mathbb{E}_{\mu, \mathcal{M}_l}[v^\star - v^{\widehat{\pi}}]}{\sum_{h=1}^{H} \sum_{(s_h, a_h) \in \mathcal{C}_h} d_h^{\pi^\star}(s_h, a_h) \cdot \sqrt{\frac{\mathrm{Var}_{P_{s_h, a_h}}(r_h + V_{h+1}^\star)}{n \cdot d_h^\mu(s_h, a_h)}}}$$

$$\geq \sup_{l \in \{1,2\}} \frac{\mathbb{E}_{\mu, \mathcal{M}_l}[v^\star - v^{\widehat{\pi}}]}{\frac{H-1}{\sqrt{8n_l/3}}}$$

$$= \sup_{l \in \{1,2\}} \sqrt{\frac{8}{3}} \cdot \frac{\sqrt{n_l} \cdot \mathbb{E}_{\mu, \mathcal{M}_l}[v^\star - v^{\widehat{\pi}}]}{H-1} \geq \sqrt{\frac{8}{3}} C' := C,$$

this concludes the proof.

**Remark H.3.** *In the proofing procedure H.1.1, we can actually get rid of the hard instance construction of Jin et al. [2020] by setting the hard instances at any time step $t$, concretely:*

- *From time $1$ to $t-1$, there is only one absorbing state $s_a$ with reward $0$;*

- *At time $t$, $s_a$ can transition to either $s_+$ or $s_-$ follow the same transition as above; from $t+1$ to $H$, $s_+$ and $s_-$ are absorbing states;*

- *$s_+$ has reward $1$ and $s_-$ has reward $0$.*

*Those instances still validate the intrinsic bound is required due to the fact that there is at least one stochastic transition kernel. This finding is interesting as it reveals the intrinsic bound is only "hard" for offline reinforcement learning when there are stochasticity in the dynamic. Under the deterministic family, those hard instances fail and we enter the faster convergence regime.*

# I  Discussions and missing derivations in Section 4

We omit the $\widetilde{O}$ notation in the derivations for the simplicity.

## I.1 Derivation in Section 4.1

When the uniform data-coverage is satisfied,

$$
\begin{aligned}
v^\star - v^{\widehat{\pi}} &\lesssim \sum_{h=1}^{H} \sum_{(s_h,a_h)\in\mathcal{C}_h} d_h^{\pi^\star}(s_h,a_h) \cdot \sqrt{\frac{\mathrm{Var}_{P_{s_h,a_h}}(r_h + V_{h+1}^\star)}{n \cdot d_h^\mu(s_h,a_h)}} \\
&\leq \sqrt{\frac{1}{nd_m}} \sum_{h=1}^{H} \sum_{(s_h,a_h)\in\mathcal{C}_h} d_h^{\pi^\star}(s_h,a_h) \cdot \sqrt{\mathrm{Var}_{P_{s_h,a_h}}(r_h + V_{h+1}^\star)} \\
&\leq \sqrt{\frac{1}{nd_m}} \sum_{h=1}^{H} \sum_{(s_h,a_h)\in\mathcal{S}\times\mathcal{A}} d_h^{\pi^\star}(s_h,a_h) \cdot \sqrt{\mathrm{Var}_{P_{s_h,a_h}}(r_h + V_{h+1}^\star)} \\
&= \sqrt{\frac{1}{nd_m}} \sum_{h=1}^{H} \sum_{(s_h,a_h)\in\mathcal{S}\times\mathcal{A}} \sqrt{d_h^{\pi^\star}(s_h,a_h)} \cdot \sqrt{d_h^{\pi^\star}(s_h,a_h)\mathrm{Var}_{P_{s_h,a_h}}(r_h + V_{h+1}^\star)} \\
&\leq \sqrt{\frac{1}{nd_m}} \sum_{h=1}^{H} \sqrt{\sum_{(s_h,a_h)\in\mathcal{S}\times\mathcal{A}} d_h^{\pi^\star}(s_h,a_h)} \cdot \sqrt{\sum_{(s_h,a_h)\in\mathcal{S}\times\mathcal{A}} d_h^{\pi^\star}(s_h,a_h)\mathrm{Var}_{P_{s_h,a_h}}(r_h + V_{h+1}^\star)} \\
&= \sqrt{\frac{1}{nd_m}} \sum_{h=1}^{H} \sqrt{\sum_{(s_h,a_h)\in\mathcal{S}\times\mathcal{A}} d_h^{\pi^\star}(s_h,a_h)\mathrm{Var}_{P_{s_h,a_h}}(r_h + V_{h+1}^\star)} \\
&\leq \sqrt{\frac{1}{nd_m}} \sqrt{\sum_{h=1}^{H} 1} \cdot \sqrt{\sum_{h=1}^{H} \sum_{(s_h,a_h)\in\mathcal{S}\times\mathcal{A}} d_h^{\pi^\star}(s_h,a_h)\mathrm{Var}_{P_{s_h,a_h}}(r_h + V_{h+1}^\star)} \\
&\leq \sqrt{\frac{1}{nd_m}} \sqrt{H} \cdot \sqrt{\mathrm{Var}_\pi\left[\sum_{t=1}^{H} r_t\right]} \leq \sqrt{\frac{H^3}{nd_m}},
\end{aligned}
$$

where we use the Cauchy inequality and Lemma J.6.

## I.2 Uniform data-coverage in the time-invariant setting (Remark 4.5)

In the time-invariant setting, $P$ is identical, therefore given data $\mathcal{D} = \{(s_h^\tau, a_h^\tau, r_h^\tau, s_{h+1}^\tau)\}_{\tau\in[n]}^{h\in[H]}$, we should modify $n_{s,a} := \sum_{h=1}^{H}\sum_{\tau=1}^{n} \mathbf{1}[s_h^\tau, a_h^\tau = s_h, a_h]$ and

$$
\widehat{P}(s'|s,a) = \frac{\sum_{h=1}^{H}\sum_{\tau=1}^{n} \mathbf{1}[(s_{h+1}^\tau, a_h^\tau, s_h^\tau) = (s', s, a)]}{n_{s,a}}, \quad \widehat{r}(s,a) = \frac{\sum_{h=1}^{H}\sum_{\tau=1}^{n} \mathbf{1}[(a_h^\tau, s_h^\tau) = (s, a)] \cdot r_h^\tau}{n_{s,a}},
$$

if $n_{s_h,a_h} > 0$ and $\widehat{P}(s'|s,a) = 1/S, \widehat{r}(s,a) = 0$ if $n_{s,a} = 0$. Define $\bar{d}^\mu(s,a) = \frac{1}{H}\sum_{h=1}^{H} d_h^\mu(s,a)$, then since in this case

$$
\mathbb{E}[n_{s,a}] = \sum_{h=1}^{H}\sum_{\tau=1}^{n} d_h^\mu(s_h,a_h) = nH\bar{d}^\mu(s,a),
$$

A similar algorithm should yield

$$
\sqrt{\frac{1}{nHd_m}} \sqrt{H} \cdot \sqrt{\mathrm{Var}_\pi\left[\sum_{t=1}^{H} r_t\right]} \leq \sqrt{\frac{H^2}{nd_m}}.
$$

Formalizing this result depends on decoupling the dependence between $\widehat{P}$ and $\widehat{V}_h$, which could be more tricky (see Yin and Wang [2021], Ren et al. [2021] for two treatments under the uniform data coverage assumption). We leave this as the future work.

## I.3 Derivation in Section 4.2

This follows from the derivation of Section 4.1 by bounding

$$v^\star - v^{\widehat{\pi}} \lesssim \sqrt{\frac{1}{nd_m}} \sqrt{H} \cdot \sqrt{\mathrm{Var}_\pi\left[\sum_{t=1}^{H} r_t\right]} \le \sqrt{\frac{H}{nd_m}}.$$

## I.4 Derivation in Section 4.3

Using the single concentrability coefficient $C^\star$, when $\pi^\star$ is deterministic,

$$v^\star - v^{\widehat{\pi}} \lesssim \sum_{h=1}^{H} \sum_{(s_h,a_h)\in\mathcal{C}_h} d_h^{\pi^\star}(s_h,a_h) \cdot \sqrt{\frac{\mathrm{Var}_{P_{s_h,a_h}}(r_h+V_{h+1}^\star)}{n \cdot d_h^\mu(s_h,a_h)}} \le \sqrt{\frac{C^\star}{n}} \sum_{h=1}^{H} \sum_{(s_h,a_h)\in\mathcal{C}_h} \sqrt{d_h^{\pi^\star}(s_h,a_h)\cdot\mathrm{Var}_{P_{s_h,a_h}}(r_h+V_{h+1}^\star)}$$

$$\le \sqrt{\frac{C^\star}{n}} \sum_{h=1}^{H} \sum_{(s_h,a_h)\in\mathcal{S}\times\mathcal{A}} \sqrt{d_h^{\pi^\star}(s_h,a_h)\cdot\mathrm{Var}_{P_{s_h,a_h}}(r_h+V_{h+1}^\star)}$$

$$= \sqrt{\frac{C^\star}{n}} \sum_{h=1}^{H} \sum_{s_h\in\mathcal{S}} \sqrt{d_h^{\pi^\star}(s_h,\pi_h^\star(s_h))\cdot\mathrm{Var}_{P_{s_h,\pi_h^\star(s_h)}}(r_h+V_{h+1}^\star)}$$

$$\le \sqrt{\frac{C^\star}{n}} \sum_{h=1}^{H} \sqrt{\sum_{s_h\in\mathcal{S}} 1} \sqrt{\sum_{s_h\in\mathcal{S}} d_h^{\pi^\star}(s_h,\pi_h^\star(s_h))\cdot\mathrm{Var}_{P_{s_h,\pi_h^\star(s_h)}}(r_h+V_{h+1}^\star)}$$

$$\le \sqrt{\frac{SC^\star}{n}} \sum_{h=1}^{H} \sqrt{\sum_{s_h\in\mathcal{S}} d_h^{\pi^\star}(s_h,\pi_h^\star(s_h))\cdot\mathrm{Var}_{P_{s_h,\pi_h^\star(s_h)}}(r_h+V_{h+1}^\star)} \le \sqrt{\frac{SC^\star}{n}} \sqrt{H} \cdot \sqrt{\mathrm{Var}_\pi\left[\sum_{t=1}^{H} r_t\right]} \le \sqrt{\frac{H^3 SC^\star}{n}}.$$

where we use the Cauchy inequality and Lemma J.6. Also, from the discussion in Section D, we know this is minimax rate optimal.

## I.5 Derivation in Section 4.4

The derivation of Proposition 4.8 is similar to the previous cases except we use the bounds $\mathrm{Var}_{P_h}(V_{h+1}^\star) \le \mathbb{Q}_h^\star$ and $\sum_{h=1}^{H} r_h \le \mathcal{B}$. The derivations for the deterministic system or the partially deterministic system are straightforward. For the fast mixing example, we leverage the fact that for any random variable $X$, $|X - \mathbb{E}[X]| \le \mathrm{rng}(X)$, hence $\mathbb{Q}^\star \le 1 + (\mathrm{rng}V^\star)^2 \le 2$.

Last but not least, we mention the *per-step environmental norm* $\mathbb{Q}_h^\star := \max_{s_h,a_h} \mathrm{Var}_{P_{s_h,a_h}}(V_{h+1}^\star)$ is more general than its maximal version in Zanette and Brunskill [2019] with $\mathbb{Q}^\star := \max_{s_h,a_h,h} \mathrm{Var}_{P_{s_h,a_h}}(V_{h+1}^\star)$. Improvement can be made for the $\mathbb{Q}_h^\star$ version, *e.g.* for the partially deterministic systems, $t\sqrt{\mathbb{Q}^\star/n\bar{d}_m}$ vs $H\sqrt{\mathbb{Q}^\star/n\bar{d}_m}$. Even though Zanette and Brunskill [2019] considers the time-invariant setting, *i.e.* $P$ is identical, the quantity $\mathbb{Q}_h^\star := \max_{s,a} \mathrm{Var}_{P_{s,a}}(V_{h+1}^\star)$ can still be much smaller than $\mathbb{Q}^\star$, *e.g.* when the range of $V_t^\star, \ldots, V_H^\star$ is relatively small and the range of $V_1^\star, \ldots, V_{t-1}^\star$ is relatively large.

In this sense, beyond the current adaptive regret $\sqrt{\mathbb{Q}^\star SAT}$ [Zanette and Brunskill, 2019], the more adaptive regret should have a form like either

$$\sqrt{\frac{\sum_{h=1}^{H} \mathbb{Q}_h^\star SAT}{H}} \quad \text{or} \quad \sum_{h=1}^{H} \frac{\sqrt{\mathbb{Q}_h^\star SAT}}{H}.$$

This remains an open question in online RL.

## J  Assisting lemmas

**Lemma J.1** (Multiplicative Chernoff bound Chernoff et al. [1952])**.** *Let $X$ be a Binomial random variable with parameter $p, n$. For any $1 \ge \theta > 0$, we have that*

$$\mathbb{P}[X < (1-\theta)pn] < e^{-\frac{\theta^2 pn}{2}}. \quad \text{and} \quad \mathbb{P}[X \ge (1+\theta)pn] < e^{-\frac{\theta^2 pn}{3}}$$

**Lemma J.2** (Hoeffding's Inequality Sridharan [2002]). *Let $x_1, ..., x_n$ be independent bounded random variables such that $\mathbb{E}[x_i] = 0$ and $|x_i| \leq \xi_i$ with probability 1. Then for any $\epsilon > 0$ we have*

$$\mathbb{P}\left(\frac{1}{n}\sum_{i=1}^{n} x_i \geq \epsilon\right) \leq e^{-\frac{2n^2\epsilon^2}{\Sigma_{i=1}^n \xi_i^2}}.$$

**Lemma J.3** (Bernstein's Inequality). *Let $x_1, ..., x_n$ be independent bounded random variables such that $\mathbb{E}[x_i] = 0$ and $|x_i| \leq \xi$ with probability 1. Let $\sigma^2 = \frac{1}{n}\sum_{i=1}^{n} \text{Var}[x_i]$, then with probability $1 - \delta$ we have*

$$\frac{1}{n}\sum_{i=1}^{n} x_i \leq \sqrt{\frac{2\sigma^2 \cdot \log(1/\delta)}{n}} + \frac{2\xi}{3n}\log(1/\delta)$$

**Lemma J.4** (Empirical Bernstein's Inequality [Maurer and Pontil, 2009]). *Let $x_1, ..., x_n$ be i.i.d random variables such that $|x_i| \leq \xi$ with probability 1. Let $\bar{x} = \frac{1}{n}\sum_{i=1}^{n} x_i$ and $\widehat{V}_n = \frac{1}{n}\sum_{i=1}^{n}(x_i - \bar{x})^2$, then with probability $1 - \delta$ we have*

$$\left|\frac{1}{n}\sum_{i=1}^{n} x_i - \mathbb{E}[x]\right| \leq \sqrt{\frac{2\widehat{V}_n \cdot \log(2/\delta)}{n}} + \frac{7\xi}{3n}\log(2/\delta).$$

**Lemma J.5** (Freedman's inequality Tropp et al. [2011]). *Let $X$ be the martingale associated with a filter $\mathcal{F}$ (i.e. $X_i = \mathbb{E}[X|\mathcal{F}_i]$) satisfying $|X_i - X_{i-1}| \leq M$ for $i = 1, ..., n$. Denote $W := \sum_{i=1}^{n} \text{Var}(X_i|\mathcal{F}_{i-1})$ then we have*

$$\mathbb{P}(|X - \mathbb{E}[X]| \geq \epsilon, W \leq \sigma^2) \leq 2e^{-\frac{\epsilon^2}{2(\sigma^2 + M\epsilon/3)}}.$$

*Or in other words, with probability $1 - \delta$,*

$$|X - \mathbb{E}[X]| \leq \sqrt{8\sigma^2 \cdot \log(1/\delta)} + \frac{2M}{3} \cdot \log(1/\delta), \quad Or \quad W \geq \sigma^2.$$

**Lemma J.6** (Sum of Total Variance, Lemma 3.4 of Yin and Wang [2020]).

$$\text{Var}_\pi\left[\sum_{t=h}^{H} r_t\right]$$
$$= \sum_{t=h}^{H}\left(\mathbb{E}_\pi\left[\text{Var}\left[r_t + V_{t+1}^\pi(s_{t+1}) \mid s_t, a_t\right]\right] + \mathbb{E}_\pi\left[\text{Var}\left[\mathbb{E}\left[r_t + V_{t+1}^\pi(s_{t+1}) \mid s_t, a_t\right]\big|s_t\right]\right]\right)$$

*here $s_t, a_t, r_t, \ldots$ is a random trajectory.*

**Remark J.7.** *The infinite horizon discounted setting counterpart is $(I - \gamma P^\pi)^{-1}\sigma_{V^\pi} \leq (1 - \gamma)^{-3/2}$.*

**Lemma J.8** (Empirical Bernstein Inequality). *Let $n \geq 2$ and $V \in \mathbb{R}^S$ be any function with $||V||_\infty \leq H$, $P$ be any $S$-dimensional distribution and $\widehat{P}$ be its empirical version using $n$ samples. Then with probability $1 - \delta$,*

$$\left|\sqrt{\text{Var}_{\widehat{P}}(V)} - \sqrt{\frac{n-1}{n}\text{Var}_P(V)}\right| \leq 2H\sqrt{\frac{\log(2/\delta)}{n-1}}.$$

*Proof.* This is a directly application of Theorem 10 in Maurer and Pontil [2009]. Indeed, by direct translating Theorem 10 of Maurer and Pontil [2009],

$$V_n(V) = \frac{1}{n(n-1)}\sum_{1 \leq i < j \leq n}(V(s_i) - V(s_j))^2 = \frac{1}{n}\sum_{i=1}^{n}(V(s_i) - \overline{V})^2 = \text{Var}_{\widehat{P}}(V).$$

where $s_i \sim P$ are i.i.d random variables and

$$
\begin{aligned}
\mathbb{E}[V_n] =& \mathbb{E}\left[\mathrm{Var}_{\widehat{P}}(V)\right] = \mathbb{E}\left[\mathbb{E}_{\widehat{P}}[V^2] - \left(\mathbb{E}_{\widehat{P}}[V]\right)^2\right] \\
=& \mathbb{E}\left[\frac{1}{n}\sum_{i=1}^{n} V^2(s_i)\right] - \mathbb{E}\left[\left(\frac{1}{n}\sum_{i=1}^{n} V(s_i)\right)^2\right] \\
=& \mathbb{E}\left[V^2\right] - \frac{1}{n^2}\mathbb{E}\left[\sum_{i=1}^{n} V^2(s_i) + 2\sum_{1\le i < j\le n} V(s_i)V(s_j)\right] \\
=& \mathbb{E}\left[V^2\right] - \frac{1}{n}\mathbb{E}\left[V^2\right] - 2\frac{n(n-1)/2}{n^2}(\mathbb{E}[V])^2 \\
=& \frac{n-1}{n}\mathrm{Var}_P(V).
\end{aligned}
$$

Therefore by Theorem 10 of Maurer and Pontil [2009] we directly have the result. ∎

## J.1 Extend Value Difference

The extended value difference lemma helps characterize the difference between the estimated value $\widehat{V}_1$ and the true value $V_1^\pi$, which was first summarized in Cai et al. [2020] and also used in Jin et al. [2020].

**Lemma J.9** (Extended Value Difference (Section B.1 in Cai et al. [2020])). *Let $\pi = \{\pi_h\}_{h=1}^{H}$ and $\pi' = \{\pi_h'\}_{h=1}^{H}$ be two arbitrary policies and let $\{\widehat{Q}_h\}_{h=1}^{H}$ be any given Q-functions. Then define $\widehat{V}_h(s) := \langle \widehat{Q}_h(s,\cdot), \pi_h(\cdot \mid s)\rangle$ for all $s \in \mathcal{S}$. Then for all $s \in \mathcal{S}$,*

$$
\begin{aligned}
\widehat{V}_1(s) - V_1^{\pi'}(s) = & \sum_{h=1}^{H} \mathbb{E}_{\pi'}\left[\langle \widehat{Q}_h(s_h,\cdot), \pi_h(\cdot \mid s_h) - \pi_h'(\cdot \mid s_h)\rangle \mid s_1 = s\right] \\
& + \sum_{h=1}^{H} \mathbb{E}_{\pi'}\left[\widehat{Q}_h(s_h, a_h) - \left(\mathcal{T}_h\widehat{V}_{h+1}\right)(s_h, a_h) \mid s_1 = s\right]
\end{aligned}
\tag{45}
$$

*where $(\mathcal{T}_h V)(\cdot, \cdot) := r_h(\cdot, \cdot) + (P_h V)(\cdot, \cdot)$ for any $V \in \mathbb{R}^S$.*

*Proof.* Denote $\xi_h = \widehat{Q}_h - \mathcal{T}_h\widehat{V}_{h+1}$. For any $h \in [H]$, we have

$$
\begin{aligned}
\widehat{V}_h - V_h^{\pi'} &= \langle \widehat{Q}_h, \pi_h\rangle - \langle Q_h^{\pi'}, \pi_h'\rangle \\
&= \langle \widehat{Q}_h, \pi_h - \pi_h'\rangle + \langle \widehat{Q}_h - Q_h^{\pi'}, \pi_h'\rangle \\
&= \langle \widehat{Q}_h, \pi_h - \pi_h'\rangle + \langle P_h(\widehat{V}_{h+1} - V_{h+1}^{\pi'}) + \xi_h, \pi_h'\rangle \\
&= \langle \widehat{Q}_h, \pi_h - \pi_h'\rangle + \langle P_h(\widehat{V}_{h+1} - V_{h+1}^{\pi'}), \pi_h'\rangle + \langle \xi_h, \pi_h'\rangle
\end{aligned}
$$

recursively apply the above for $\widehat{V}_{h+1} - V_{h+1}^{\pi'}$ and use the $\mathbb{E}_{\pi'}$ notation (instead of the inner product of $P_h, \pi_h'$) we can finish the prove of this lemma. ∎

The following lemma helps to characterize the gap between any two policies.

**Lemma J.10.** *Let $\widehat{\pi} = \{\widehat{\pi}_h\}_{h=1}^{H}$ and $\widehat{Q}_h(\cdot, \cdot)$ be the arbitrary policy and Q-function and also $\widehat{V}_h(s) = \langle \widehat{Q}_h(s,\cdot), \widehat{\pi}_h(\cdot|s)\rangle \ \forall s \in \mathcal{S}$. and $\xi_h(s,a) = (\mathcal{T}_h\widehat{V}_{h+1})(s,a) - \widehat{Q}_h(s,a)$ element-wisely. Then for any arbitrary $\pi$, we have*

$$
\begin{aligned}
V_1^\pi(s) - V_1^{\widehat{\pi}}(s) = & \sum_{h=1}^{H} \mathbb{E}_\pi\left[\xi_h(s_h, a_h) \mid s_1 = s\right] - \sum_{h=1}^{H} \mathbb{E}_{\widehat{\pi}}\left[\xi_h(s_h, a_h) \mid s_1 = s\right] \\
& + \sum_{h=1}^{H} \mathbb{E}_\pi\left[\langle \widehat{Q}_h(s_h,\cdot), \pi_h(\cdot|s_h) - \widehat{\pi}_h(\cdot|s_h)\rangle \mid s_1 = x\right]
\end{aligned}
$$

*where the expectation are taken over $s_h, a_h$.*

*Proof.* Note the gap can be rewritten as

$$V_1^\pi(s) - V_1^{\widehat{\pi}}(s) = V_1^\pi(s) - \widehat{V}_1(s) + \widehat{V}_1(s) - V_1^{\widehat{\pi}}(s).$$

By Lemma J.9 with $\pi = \widehat{\pi}$, $\pi' = \pi$, we directly have

$$V_1^\pi(s) - \widehat{V}_1(s) = \sum_{h=1}^H \mathbb{E}_\pi \left[ \xi_h(s_h, a_h) \mid s_1 = s \right] + \sum_{h=1}^H \mathbb{E}_\pi \left[ \langle \widehat{Q}_h(s_h, \cdot), \pi_h(\cdot|s_h) - \widehat{\pi}_h(\cdot|s_h) \rangle \mid s_1 = s \right] \tag{46}$$

Next apply Lemma J.9 again with $\pi = \pi' = \widehat{\pi}$, we directly have

$$\widehat{V}_1(s) - V_1^{\widehat{\pi}}(s) = - \sum_{h=1}^H \mathbb{E}_{\widehat{\pi}} \left[ \xi_h(s_h, a_h) \mid s_1 = s \right]. \tag{47}$$

Combine the above two results we prove the stated result. ∎

## J.2 Hellinger Distance

**Definition J.11.** *Let $f, g$ are the two probability densities on the same probability space $\mathcal{X}$. Then the Hellinger distance between $f$ and $g$ is defined as:*

$$d_{Hel}^2(f, g) = \frac{1}{2} \int_{x \in \mathcal{X}} \left( \sqrt{f(x)} - \sqrt{g(x)} \right)^2 dx = 1 - \int_{x \in \mathcal{X}} \sqrt{f(x) g(x)} dx.$$

*In particular, it holds that*

$$\|f - g\|_{TV} \le \sqrt{2} \cdot d_{Hel}(f, g).$$

## J.3 Propoerty of Local Instance

**Lemma J.12.** *Recall the definition of $P'$ in (31)*

$$P_h'(s_{h+1}|s_h, a_h) = P_h(s_{h+1}|s_h, a_h) + \frac{P_h(s_{h+1}|s_h, a_h) \left( V_{h+1}^\star(s_{h+1}) - \mathbb{E}_{P_{s_h, a_h}}[V_{h+1}^\star] \right)}{8 \sqrt{\zeta \cdot n_{s_h, a_h} \cdot \mathrm{Var}_{P_{s_h, a_h}}(V_{h+1}^\star)}},$$

*where $\zeta = H/\bar{d}_m$ and $P$ is the original instance. Then :*

- *with high probability, when $n \ge C \cdot \sup_{h, s_{h+1}, s_h, a_h} \left( \frac{H}{1/P_h(s_{h+1}|s_h, a_h) - 1} \right)^2 \cdot \frac{1}{H \cdot \mathrm{Var}(V_{h+1}^\star)}$, $P'$ is a valid probability distribution;*

- $d_{Hel}(P_h'(\cdot|s_h, a_h), P_h(\cdot|s_h, a_h)) \le \frac{1}{\sqrt{2nH}}$ *for $n$ sufficiently large;*

- *Elementwisely, $(P_h' - P_h) V_{h+1}^\star \ge \mathbf{0}$ for all $h \in [H]$.*

**Remark J.13.** *WLOG, let us assume $n_{s_h, a_h} \cdot \mathrm{Var}_{P_{s_h, a_h}}(V_{h+1}^\star) > 0$ and this is valid since: 1. if $\mathrm{Var}_{P_{s_h, a_h}}(V_{h+1}^\star) = 0$ then these is no need to use multiple samples to test $P_h(\cdot|s_h, a_h)$ if $P_h(\cdot|s_h, a_h)$ is deterministic or if $V_{h+1}^\star \equiv 0$ then we can define $\frac{(V_{h+1}^\star(s_{h+1}) - \mathbb{E}_{P_{s_h, a_h}}[V_{h+1}^\star])}{2\sqrt{n_{s_h, a_h} \cdot \mathrm{Var}_{P_{s_h, a_h}}(V_{h+1}^\star)}} = 0$. 2. If $d_h^\mu(s_h, a_h) > 0$, then by Lemma J.1 $n_{s_h, a_h} > 0$ with high probability.*

*Proof of Lemma J.12.* **Proof of item 1.** First,

$$\sum_{s_{h+1}} P'_h(s_{h+1}|s_h,a_h) = \sum_{s_{h+1}} P_h(s_{h+1}|s_h,a_h)$$

$$+ \sum_{s_{h+1}} \frac{P_h(s_{h+1}|s_h,a_h)\left(V^\star_{h+1}(s_{h+1}) - \mathbb{E}_{P_{s_h,a_h}}[V^\star_{h+1}]\right)}{8\sqrt{\zeta \cdot n_{s_h,a_h} \cdot \mathrm{Var}_{P_{s_h,a_h}}(V^\star_{h+1})}}$$

$$= 1 + \frac{\mathbb{E}_{P_{s_h,a_h}}[V^\star_{h+1}] - \mathbb{E}_{P_{s_h,a_h}}[V^\star_{h+1}]}{8\sqrt{\zeta \cdot n_{s_h,a_h} \cdot \mathrm{Var}_{P_{s_h,a_h}}(V^\star_{h+1})}} = 1$$

Second, the non-negativity holds as long as

$$n_{s_h,a_h} \geq \left(\frac{H}{1/P_h(s_{h+1}|s_h,a_h) - 1}\right)^2 \cdot \frac{1}{\zeta \cdot \mathrm{Var}(V^\star_{h+1})}, \quad \forall s_h,a_h \quad s.t. \quad \mathrm{Var}(V^\star_{h+1}) > 0.$$

This is guaranteed by Lemma J.1 with high probability when

$$n \geq C \cdot \sup_{h,s_{h+1},s_h,a_h} \left(\frac{H}{1/P_h(s_{h+1}|s_h,a_h) - 1}\right)^2 \cdot \frac{1}{H \cdot \mathrm{Var}(V^\star_{h+1})},$$

where the sup is over all terms such that $\mathrm{Var}(V^\star_{h+1})d^\mu_h(s_h,a_h) > 0$.

**Proof of item 2.** Denote $\Delta_h := \frac{V^\star_{h+1}(s_{h+1}) - \mathbb{E}_{P_{s_h,a_h}}[V^\star_{h+1}]}{8\sqrt{\zeta \cdot n_{s_h,a_h} \cdot \mathrm{Var}_{P_{s_h,a_h}}(V^\star_{h+1})}}$ By the definition, we have

$$d^2_{Hel}(P'_h(\cdot|s_h,a_h), P_h(\cdot|s_h,a_h)) = \left|1 - \sum_{s_{h+1}} \sqrt{P_h(s_{h+1}|s_h,a_h) \cdot P'_h(s_{h+1}|s_h,a_h)}\right|$$

$$= \left|1 - \sum_{s_{h+1}} \sqrt{P^2_h(s_{h+1}|s_h,a_h)(1 + \Delta_h)}\right| = \left|1 - \sum_{s_{h+1}} P_h(s_{h+1}|s_h,a_h)\sqrt{(1 + \Delta_h)}\right|$$

$$= \left|\sum_{s_{h+1}} P_h(s_{h+1}|s_h,a_h) - \sum_{s_{h+1}} P_h(s_{h+1}|s_h,a_h)\sqrt{(1 + \Delta_h)}\right| = \left|\sum_{s_{h+1}} P_h(s_{h+1}|s_h,a_h)\left(1 - \sqrt{(1 + \Delta_h)}\right)\right|$$

$$= \left|\sum_{s_{h+1}} P_h(s_{h+1}|s_h,a_h)\left(1 - \left(1 + \frac{\Delta}{2} - \frac{\Delta^2}{8}\right)\right)\right| \quad (*)$$

$$= \left|\sum_{s_{h+1}} P_h(s_{h+1}|s_h,a_h)\left(-\frac{\Delta}{2} + \frac{\Delta^2}{8}\right)\right| = \left|\sum_{s_{h+1}} P_h(s_{h+1}|s_h,a_h) \cdot \frac{\Delta^2}{8}\right| = \frac{1}{8^3 \cdot \zeta \cdot n_{s_h,a_h}} \leq \frac{1}{2n \cdot H}$$

the step $(*)$ comes from second order Taylor expansion (where we omit the higher order since $n$ is sufficiently large already) and the next equation uses $\sum_{s_{h+1}} P_h(s_{h+1}|s_h,a_h) \cdot \Delta = 0$ and the last inequality uses Lemma J.1 $n_{s_h,a_h}$ such that $n_{s_h,a_h} \cdot \zeta \geq C \cdot n \cdot H$ with high probability and $d^2_{Hel}(P'_h(\cdot|s_h,a_h), P_h(\cdot|s_h,a_h)) \leq \frac{1}{2nH}$.

**Proof of item 3.** Note

$$[(P'_h - P_h)V^\star_{h+1}](s_h,a_h) = \sum_{s_{h+1}} (P'_h(s_{h+1}|s_h,a_h) - P_h(s_{h+1}|s_h,a_h))V^\star_{h+1}(s_{h+1})$$

$$= \sum_{s_{h+1}} (P'_h(s_{h+1}|s_h,a_h) - P_h(s_{h+1}|s_h,a_h))(V^\star_{h+1}(s_{h+1}) - \mathbb{E}_{P_{s_h,a_h}}[V^\star_{h+1}])$$

$$= \frac{1}{2}\sqrt{\frac{\mathrm{Var}_{P_{s_h,a_h}}(V^\star_{h+1})}{\zeta \cdot n_{s_h,a_h}}} \geq 0$$

∎