# OpenReview forum: "Towards Instance-Optimal Offline Reinforcement Learning with Pessimism"
_NeurIPS.cc/2021/Conference — NeurIPS 2021 Poster_

### Official Review · Reviewer_EcDg · 2021-07-14

**Rating:** 6
**Confidence:** 5

**Summary:**

This paper studies the capability and intrinsic hardness of pessimistic value iteration algorithm in the finite-horizon tabular MDP case when data is i.i.d. collected from a behavior policy. By a fine-grained analysis, the PEVI (reduced to the i.i.d. tabular case) is modified to incorporate empirical variance into the pessimistic penalty term. These empirical quantities converge to their population counterparts, which is also shown by a lower bound as the intrinsic hardness of learning the MDP with data collected by the policy. Extensive examples & special cases are analyzed to provide a comprehensive analysis of pessimistic offline RL in this setting.

**Limitations And Societal Impact:**

No. Maybe more discussions on limitations of the work can be an improvement.

**Main Review:**

Originality:
This work falls under the umbrella of the pessimistic value iterations framework [Jin et al., 2020], and provides a fine-grained and comprehensive analysis of a special case: tabular MDP with i.i.d. data from a behavior policy. The approach is not novel, but new insights and deeper understanding of offline RL with pessimism are established, which adds to the knowledge of the community.

Quality & Clarity:
The paper is technically sound and clearly written.

Significance:
The work adds to the knowledge of the community on offline RL by a fine-grained analysis. It might open possibilities of finer understanding of offline RL in more general settings, e.g., beyond tabular, beyond i.i.d.

More comments:
Strength:
1. The fine-grained analysis of penalty function provides more insights and understanding of the offline RL with pessimism.
2. The minimax optimal lower bound shows the intrinsic hardness of the problem.
Weakness / concerns:
1. The papers studies the special case of tabular MDP and i.i.d. trajectories, which might be a bit specific compared to the vanilla version [Jin et al., 2020].
2. The upper bound is not data-dependent, hence requires large sample size, while the work [Jin et al., 2020] poses no assumptions on the data except a minimal 'compliance' assumption. Would it be possible to give a data-dependent finite-sample version of the upper bound? Why do you prefer the current one?











Reference:
[Jin et al., 2020] Ying Jin, Zhuoran Yang, and Zhaoran Wang. Is pessimism provably efficient for offline rl? arXiv preprint arXiv:2012.15085, 2020.

**Time Spent Reviewing:**

4

---

> ### Author Response · Authors · 2021-08-10
> **Response for the reviewer EcDg**
>
> We appreciate the reviewer for the valuable comments and the precise understanding of our paper.
>
> ----- "The papers studies the special case of tabular MDP and i.i.d. trajectories, which might be a bit specific compared to the vanilla version" -----
>
> Thanks the reviewer for mentioning the tabular setting. Indeed, the tabular setting is not general since it assumes the discrete MDPs. However, even in this well-studied setting no previous theoretical work can provide strong adaptive guarantee and subsume minimax rates under various assumptions like this paper. We fill in this gap and we believe this paper is necessary for future research in terms of understanding instance-dependent offline RL in the general settings.
>
> The i.i.d.ness is commonly assumed in finite horizon offline RL, examples include [Xie et al. 2019, Yin and Wang, 2020, Duan and Wang, 2020, Ren et al. 2021]. It caters to the real-world offline applications like the game of GO and Robotics (you restart to the initial state after each trial for the task).
>
> [1] [Xie et al.] Towards Optimal Off-Policy Evaluation for Reinforcement Learning with Marginalized Importance Sampling, NeurIPS, 2019.
>
> [2] [Duan and Wang] Minimax-optimal off-policy evaluation with linear function approximation, ICML, 2020.
>
>
> ----- "The upper bound is not data-dependent, hence requires large sample size, ... a minimal 'compliance' assumption. Would it be possible to give a data-dependent finite-sample version of the upper bound?" -----
>
> Thank you very much for the insightful question!
>
> A data-dependent result is only the intermediate step of our study: the number of sample at each time step $n_{s_h,a_h}$ is distributed according to Binomial distribution with parameters $n$ and $d^\mu(s_h,a_h)$. Therefore, we can convert our result from the current expression to the version with data counts $n_{s_h,a_h}$ by Chernoff bound (Lemma H.1), which is actually line 695. Note in this case, we can actually get rid of the Assumption 2.3 by setting n'_{s_h,a_h}=\max(n_{s_h,a_h},1) or n'(s_h,a_h)=n(s_h,a_h)+1 (where n(s_h,a_h)+1 comes from their ridge regression with $\lambda=1$). That is to say, our AVPI also works under the compliance assumption. This is an excellent point and we will add this as an corollary in our revision.

---

### Official Review · Reviewer_HoUV · 2021-07-15

**Rating:** 8
**Confidence:** 2

**Summary:**

They propose a lower bound under the weak coverage assumption for offline RL. The lower bound is also reduced to several lower bounds in specific examples. They also show how this lower bound is achieved.


**Limitations And Societal Impact:**

Some minor comments:

* Line 304:  Var(r_1) is correct? Is it conditional variance?
*  Line 272:  the use of C^{*} makes the bound sightly problem independent: I guess in the time-invariant setting, the dependence on C^{*}  sounds very natural since we can get the upper-bound characterized by  C^{*}.  Does it mean using C^{*}  in the time-variant setting is problematic?
* Probably, worthwhile to note Liu et.al (2020) is assumption-free in a tabular case somewhere. They get an assumption-free result by setting the threshold for the offline data distribution. Then, they obtained an error based on the augmented MDP by this thresholding. Basically, I feel people can always get this assumption-free result by some type of thresholding on top of the original algorithm. Actually, Jin (2020) result is assumption-free (at least they claimed in this way) since there is a threshold parameter lambda.  So, might be not needed to emphasize too much.
* Is they any relation for the Cramer-Rao lower bound for OPE?


**Main Review:**

* Originality: Main results (lower bound and upper bound) are novel and very meaningful. This paper also unifies several scattered lower bound for offline RL. That’s also a nice contribution.

* Quality:  In the interest of time, I did not check the proof. But, according to a sketch of the analysis, it sounds correct.

* Clarity: Yes, it is clear

* Significance: Many theory offline RL people would be excited with this work. I enjoyed it a lot.



**Time Spent Reviewing:**

1.5

---

> ### Author Response · Authors · 2021-08-10
> **Response for the reviewer HoUV**
>
> We appreciate the reviewer for the comments and for understanding the value of this paper. Here are the detailed responses.
>
> ----- "Line 304: ... Is it conditional variance?" ------
>
> Thanks for pointing that out! Yes, it has to be the conditional variance given the state and action and we will edit this notation in a more precise way in our revision. Such a quantity actually tells an interesting story: even in the case where the transition dynamic $P_t$ is deterministic, the convergence rate still follows the standard statistical learning rate $\frac{1}{\sqrt{n}}$ without faster convergence (since $\mathrm{Var}(r_t)$ can be positive); such a perspective is lacking in the purely minimaxity literature (e.g. [Yin et al. 2021a]) since the transition $P_t$ dominates the complexity of the problem therefore the reward is commonly assumed to be deterministic.
>
>
> ----- " I guess in the time-invariant setting, the dependence on $C^
> {\pi}$ ($C^\star$) sounds very natural ..." --------
>
> Thanks. We agree with the reviewer and all we want to say is that our result is more general.
>
> ----- " note Liu et.al (2020) is assumption-free ... somewhere, they get an assumption-free result by setting the threshold for the offline data distribution. Jin et al. (2020)"
>
> Thanks for the excellent catch and this is a very insightful question! Indeed, we have already added this in our revision at hand and here is the difference: [Liu et al. 2020] considers the behavior policy with insufficient coverage probability $\epsilon_\zeta$ (see their Definition~1), but they end up with the suboptimality gap $\frac{V_{\max}\epsilon_\zeta}{1-\gamma}$ (their Theorem 1), when the insufficient coverage probability $\epsilon_\zeta\geq \frac{1}{2}$, this gap has order $(1-\gamma)^{-2}$, which is larger in order than the biggest possible suboptimality gap $(1-\gamma)^{-1}$ therefore unable to characterize the more essential statistical gap over the region that can never be visited by the behavior policy (and this happens similarly in [Kidambi et al. 2020], see their Theorem 1);
>
> [1] [Kidambi et al.] Morel: Model-based offline reinforcement learning. NeurIPS, 2020.
>
> Moreover, the regression penalty technique used in [Jin et al. 2020] essentially corresponds to the "plus one" technique when $n_{s,a}=0$ in the tabular setting and this is (in principle) very similar to the Laplace smoothing (in Naive Bayes algorithm) for dealing with zero count. Such a treatment is convenient but lacks the accuracy (for characterizing the gap) since: for example, at time $H-1$, if d^{\pi^\star}_{H-1}(s,a)=1 for some s_h,a_h but n_{s_h,a_h}=0, then this term will contribute the constant gap
>
> $$
> H\cdot d^{\pi^\star}_{H-1}(s,a) \cdot\frac{1}{\sqrt{n_{s_h,a_h}+1}}=H\cdot \frac{1}{\sqrt{0+1}}=H
> $$
>
> from their bound. In contrast, in this situation our assumption-free result will only sacrifice gap $1$ (since it corresponds to the term $d^{\dagger\pi^\star}_{H-1}(s^\dagger)\leq 1$). Suffering a $H$ gap in a single location is not very informative in understanding the challenging assumption-free regime.
>
> In conclusion, existing works Can provide assumption-free guarantee but our \sum_{h=2}^{H+1}d^{\dagger\pi^\star}_h(s^\dagger_h) quantity (with d^{\dagger\pi^\star}_h(s^\dagger_h)=\sum_{t=1}^{h-1}\sum_{(s_t,a_t)\in\mathcal{S}\times\mathcal{A}\backslash\mathcal{C}_t}d^{\dagger\pi^\star}_t(s_t,a_t)\leq 1) describes the ``must-suffer'' gap in a more precise way by absorbing all the agnostic probabilities into $s^\dagger$. In this sense, Theorem 5.1 provides a more precise understanding about what can offline RL can achieve when no assumption is made. This is a very valuable point from the reviewer and we will add careful discussions in our revision.
>
> ---- "Is they any relation for the Cramer-Rao lower bound for OPE?" ----
>
> We believe the reviewer is talking about C-R lower bound for the tabular OPE. They differ by a factor of $H$ from the asymptotic statistical perspective (comparing our work with [Yin and Wang, 2020], [Duan and Wang, ICML, 2020] and [Jiang and Li, 2016]). We will add a discussion in our revision.
>
> [1] [Jiang and Li] Doubly Robust Off-policy Value Evaluation for Reinforcement Learning, ICML, 2016

---

> > ### Comment · Reviewer_HoUV · 2021-08-25
> > **Several thoughts**
> >
> > Thank you for the rebuttal. I am convinced. I keep the score.
> > Regarding the discussion in the other thread, I can only follow partially since it is a too expert topic for me. But it is very insightful and provoking, here is my thought. I might be wrong.
> >
> > ——————
> >  In my understanding, [1] Xiao, C., they derive minimax optimal lower bound and instance dependent lower bound. They concluded pessimism does not help in both senses unless using tricky instance dependent lower bounds.
> >
> > Here,  in the current paper, the author provided a minimax optimal lower bound, which is characterized by important quantities d_mu, d^{*} etc. Pessimism is crucial to achieving this lower bound. So, it sounds this contradicts [1]… In my understanding, this is because quantities of interests are different provably. (Like d^{*} is included in this work, but not in [1]??)  Anyway, it should be definitely helpful for readers to add in the next version.
> >
> > Regarding instance-dependent lower bounds, the author claims that this is beyond their work in the last section. So, I think this is fine. The contribution of this paper would be providing adaptive minimax lower bound, and this is beyond the threshold of neurips submission.  But it would be also helpful to add more information regarding instance-dependent lower bounds.  The sentence in the last section might be too terse. (this might be because of space though) E.g. standard readers do not know anything about local minimax instance dependent lower bounds. So, citing several literature (and more explanation) would be helpful.

---

> > > ### Author Response · Authors · 2021-08-25
> > > **Further Response**
> > >
> > > Thank you for your nice suggestions, and yes, RL is in general more challenging than MAB and there are differences. As we promised to reviewer 8b9i, we will add the careful discussions w.r.t. [Xiao et al.] in the revision, especially the "weighted minimax" scheme. For those materials that couldn't fit into the main text, we will put them in the appendix.

---

### Official Review · Reviewer_8b9i · 2021-07-20

**Rating:** 6
**Confidence:** 4

**Summary:**

The paper proposes and analyzes the adaptive pessimistic value iteration algorithm for offline reinforcement learning. The authors study the sample complexity of the proposed algorithm and prove matching upper and lower bound. The paper also studies the regime where no assumption about the behaviour policy is made and provide a sub-optimality upper bound.



**Ethical Concerns:**

No ethical issue has been found.

**Limitations And Societal Impact:**

No societal issue has been found.

**Main Review:**

The main idea of the paper is to use Bernstein-type confidence interval in the algorithm and analysis, which gives the improved bound over the Hoeffding-type analysis (Jin et al. 2020). The authors provide thorough theoretical analysis including matching regret upper and lower bounds which is very nice. This paper is also well written and the problem setup is clearly defined.

However, I still have the following concerns and questions. Please address these in the rebuttal and correct me if I misunderstand some critical part of the work.

The advantage of using Bernstein-based analysis over the Hoeffding-based analysis is well established in the theoretical RL literatures. This makes the contribution of the paper somehow incremental and less interesting, although it seems that this is the first attempt to use such technique in offline RL.

I think to study the sample complexity in offline RL, we first need to understand how to define “the optimal” algorithm in this problem. The minimax optimality is too weak as a criterion in the offline setting as it only considers the worst case, which gives a dependency on the minimum data coverage. So it seems that any algorithm that makes prediction based on the confidence interval is minimax optimal. Also, it is recently known that in offline RL, no algorithm can be instance-dependent optimal. And in fact, any “reasonable” algorithm, including pessimism, plug-in or even optimism, can find some niche where it dominates all the other algorithms (see [1] for details). So no matter how the confidence interval is constructed in practice (either Hoeffding-type or Berstain-type), it’s still not possible to get an "optimal" algorithm in the instance level. This seems to be a closely related issue that should be discussed in the paper.

Since we cannot use minimax optimality or instance optimality to study the problem, one can instead define a “weighted minimax” with different weighting scheme, like the one used in the paper, the visitation under the optimal policy used in Jin et al. 2020, and the optimal value prediction error used in [1]. The choice of the weighting is kind of arbitrary as we have already known that any algorithm is good in some problem instance and we just need to find a weighting that is attached to the properties of good problems. I think this is the reason why both APVI and PVI can match the "corresponding" lower bound.

Regarding the extension to the no-assumption case, I think the result presented in Theorem 5.1 is not very interesting. The result just says that if $\pi^*$ is not covered the algorithm will suffer linear regret. I think the assumption that $\pi^*$ is covered by the behaviour policy is actually acceptable. Otherwise if there is no information can be collected in the data the best an algorithm can do is to make a random guess. Please correct me if I made a mistake on this.

[1] Xiao, C., Wu, Y., Mei, J., Dai, B., Lattimore, T., Li, L., Szepesvari, C. and Schuurmans, D., 2021, July. On the optimality of batch policy optimization algorithms. In International Conference on Machine Learning (pp. 11362-11371). PMLR.


**Time Spent Reviewing:**

4

---

> ### Author Response · Authors · 2021-08-10
> **Response for the reviewer 8b9i**
>
> We thank the reviewer for providing the valuable feedbacks and we really enjoy reading your expert comments! In particular, thanks for bringing up [1] and we have checked [1] very carefully. The followings are our detailed responses.
>
> ---- "The advantage of using Bernstein-based analysis ... well established in the theoretical RL ... less interesting" ----
>
> First of all, we agree Bernstein-based analysis is well studied and we did not claim it as our contribution in the paper (section 1.1). Indeed, what is important is that we derived strong problem-dependent result that can depict some generic structure of the offline RL instance and it subsumes (almost) all minimax results under the respective assumptions. We believe such an instance study has the merit of its own and it is new in the offline RL (where "RL" refers to the problem with horizon as least $2$, not bandits and the variance structure has only been derived in the online linear bandit setting [Zhang et al. 2021]).  In particular, our result directly generalizes over one new released work [Xie et al. 2021].
>
> [2] [Zhang et al. 2021] Variance-Aware Confidence Set: Variance-Dependent Bound for Linear Bandits and Horizon-Free Bound for Linear Mixture MDP, 2021
>
> [3] [Xie et al.] Policy Finetuning: Bridging Sample-Efficient Offline and Online Reinforcement Learning, 2021
>
> ----- "how to define “the optimal” algorithm in this problem", "no algorithm can be instance-dependent optimal", "can find some niche where it dominates all the other algorithms" " instead define a “weighted minimax” with different weighting scheme" -----
>
> Those are really outstanding comments with high quality. First of all, we want to say we have exactly the same goal as the reviewer since we wish to understand what is the more essential "optimality" in offline RL and as a result we go beyond the minimax optimality and study this problem at the instance level.
>
> The cited lower bound (Theorem 4 in [1]) does not contradict with our results and we do not claim instance-optimality in the same sense as in Xiao et al (2021) (see our line 358). In fact, we discovered a very similar phenomena when we initially analyzed the Plug-In approach (Greedy), which depends on both the optimal policy and the empirically optimal policy --- the same story from Xiao et al (2021) for “Greedy” in bandits. However, in our case pessimistic algorithm is usually a better choice than the optimistic one, due to the Assumption 2.3. and this corresponds to comments "The behavior policy plays a subset of the arms $S \subset [k]$ a large number of times and ignores the rest. If $S$ contains at least one good arm but no bad arm, LCB will select a good played arm" in [1] (see the Discussion after Remark 4 in [1]) since $\mu$ covers $\pi^\star$ by our Assumption 2.3. While we value [1] as an excellent literature, they seem to consider the case where $n_i>0$ for all arms, which corresponds to the regime where $\mu$ explore all the state-actions in offline RL. Our Assumption 2.3 is weaker and in those challenging MDPs (with only partial coverage w.r.t. $\pi^\star$), our pessimistic algorithm prevails over the optimistic ones. We do agree when $\mu$ explores more uniformly all of the methods can be relatively good and this teaches us for simpler tasks (sufficient coverage) all of the reasonable methods are quite good and instance-optimal consideration might only be relevant in the challenging tasks.
>
> In addition, we do agree the weighted notion of optimality is particularly interesting and we will add the detailed discussions of [1] and "weighted notion" in our paper.
>
> ---- "Regarding the extension to the no-assumption case" --------
>
> While 2.3 is a nice assumption, in practice considerations data often do not cover any optimal policy (see [Kidambi et al. 2020] for an study). We believe providing guarantee in this regime and also obtaining optimality in the reduced situation is helpful for the general offline RL. For more details, please refer to our third response to reviewer HoUV.
>
> [4] [Kidambi et al.] Morel: Model-based offline reinforcement learning. NeurIPS, 2020.
>
>
> We sincerely appreciate it if the reviewers could kindly consider improving the scores if we successfully addressed the concerns.

---

> > ### Comment · Reviewer_8b9i · 2021-08-23
> > **Response**
> >
> > Dear authors,
> >
> > Thank you for your detailed response.
> >
> > I agree that the results shown in this paper do not contradict with Xiao et al as the instance-optimality is not defined in the same way.
> > And this is why I strongly encourage the authors to specify the meaning of "instance-optimality" in the paper, as instance-dependent optimality has a specific meaning in theoretical analysis (see Chapter 16 in Lattimore and Szepesvári 2020).
> >
> > **While we value [1] as an excellent literature, they seem to consider the case where $n_i>0$ for all arms**
> >
> > I think assumption 2.1 and 2.3 do not make significant difference in the bandit case. These two assumptions do matter for the MDP case, where coverage of the state space can be very different and might have huge effect on the sample complexity.
> >
> > In the bandit case, if an arm has not been seen once in the data, we have no information about that arm (it could either be extremely good or bad), and thus there is no reason to argue the optimality of an algorithm and one should just remove this arm from the candidate arms. In some sense, this is quite similar as just assuming the optimal arm has been covered in the data set.
> >
> > Furthermore, even the authors do not agree on this, I think the results in [1] still hold under assumption 2.3.

---

> > > ### Author Response · Authors · 2021-08-24
> > > **Further response to reviewer 8b9i**
> > >
> > > We appreciate the reviewer for taking time responding to us.
> > >
> > > If we understand the reviewer correctly, the only question seems to be the definition of "instance-optimality" in offline RL. The following are our responses.
> > >
> > > First of all, we want to gently reiterate that we didn't claim we achieve the instance optimality (Line 355-356). The main contribution is that our result provides an instance-dependent bound that is fully described by $d^\mu,d^\star$ and $\mathrm{Var}_P(V^\star)$ with no explicit $H,S,A$ dependence. Such an adaptive guarantee is the first time and very strong as it subsumes basically all the existing minimax optimal results under various coverage assumptions.
> > >
> > > Second, since the reviewer asked, one appropriate definition of "instance optimality" (there could be alternative ways to characterize and we don't mean this is the only one) is via the non-asymptotic local minimax risk:
> > >
> > > $$
> > > \mathcal{R}_{n}(\mathcal{P}):=\sup _{\mathcal{P}^{\prime} \in \mathcal{G}} \inf _{\widehat{\pi}} \max _{\mathcal{Q} \in \\{ \mathcal{P}, \mathcal{P}^{\prime} \\}} \sqrt{n}\cdot E^{\mathcal{Q}}\left[v^{\star}(\mathcal{Q})-v^{\widehat{\pi}}\right]
> > > $$
> > >
> > >    where $\mathcal{P}=(\mu,M)$ is the instance at hand, $\widehat{\pi}$ is the output of any offline algorithm and $\mathcal{G}$ contains all the instances that satisfy Assumption 2.3. Such a definition naturally mirrors the very recent release [1] and is a standard alternative notion of instance optimality from John Duchi's thesis [2], and all of these non-asymptotic characterizations can be traced back to the original idea of Hajek-Le Cam local asymptotic result [3]. We hope those explanations can convince the reviewer this is one appropriate definition.
> > >
> > >    Lastly, we can actually establish a local minimax lower bound $\mathcal{R}_{n}(\mathcal{P})$ such that it nearly matches equation (1) (multiply by $\sqrt{n}$) by constructing a local instance $\mathcal{P}'$ that is similar to [1]. But even this does not mean our AVPI is optimal for all instances since for $P,r$ deterministic quantity (1) is $0$. We thank the reviewer for pointing this out and we will add this definition and our new finding in our revision.
> > >
> > > We kindly wish the reviewer to increase the score if your problems are resolved, and, if there are still questions, we are very happy to further explain them :)
> > >
> > > [1] [Koulik K., Eric X., Martin J W., Michael Jordan], Instance-optimality in optimal value estimation: Adaptivity via variance-reduced Q-learning, 2021.
> > >
> > > [2] [John Duchi] Multiple Optimality Guarantees in Statistical Learning, 2014.
> > >
> > > [3] [Theorem 10] https://web.stanford.edu/class/ee378a/lecture-notes/lecture12.pdf

---

> > > > ### Comment · Reviewer_8b9i · 2021-08-24
> > > > **Response**
> > > >
> > > > Thanks for your response and sorry for the confusion about my previous response.
> > > >
> > > > I was not suggesting the definition in the bandit book is the **only** instance-optimality definition. That definition seems to be the most obvious choice for policy optimization, as the hardness of an instance is measured by the performance of the best possible minimax algorithm on that instance, and it directly compares the algorithm's performance against that.
> > > >
> > > > I totally agree there are other instance-optimality definitions, like the one used for statistical estimation and local minimax as suggested by the reviewers. In some sense, the weighted regret also has some instance nature in the definition, where the hardness of an instance is measure by the "weighting". I was only suggesting the paper should give clear definition and comparison in the paper :)
> > > >
> > > > I will consider increasing the score after discussing with other reviewers.

---

### Decision · Program_Chairs · 2021-09-27

**Decision:**

Accept (Poster)

**Comment:**

This paper presents adaptive sample complexity bounds for offline RL. While the technique is not entirely novel, the results are new and interesting in the literature and thus the AC recommends acceptance. The authors are encouraged to incorporate comments from the reviewers.